# Principles for coding associative memories in a compact neural network

**Christian Pritz†, Eyal Itskovits, Eduard Bokman, Rotem Ruach, Vladimir Gritsenko, Tal Nelken, Mai Menasherof, Aharon Azulay, Alon Zaslaver***

Department of Genetics, Silberman Institute for Life Sciences, Edmond J. Safra Campus, The Hebrew University of Jerusalem, Jerusalem, Israel

**Abstract** A major goal in neuroscience is to elucidate the principles by which memories are stored in a neural network. Here, we have systematically studied how four types of associative memories (short- and long-term memories, each as positive and negative associations) are encoded within the compact neural network of *Caenorhabditis elegans* worms. Interestingly, sensory neurons were primarily involved in coding short-term, but not long-term, memories, and individual sensory neurons could be assigned to coding either the conditioned stimulus or the experience valence (or both). Moreover, when considering the collective activity of the sensory neurons, the specific training experiences could be decoded. Interneurons integrated the modulated sensory inputs and a simple linear combination model identified the experience-specific modulated communication routes. The widely distributed memory suggests that integrated network plasticity, rather than changes to individual neurons, underlies the fine behavioral plasticity. This comprehensive study reveals basic memory-coding principles and highlights the central roles of sensory neurons in memory formation.

## Editor's evaluation

In this study, the authors established paradigms for appetitive and aversive short-term and long-term olfactory learning. They then produced a large collection of activity recordings in a handful of sensory neurons and interneurons, produced a linear model to describe sensory-evoked interneuron activities, and observed changes in the activities caused by learning. Although more work is needed to explain how these activity patterns relate to behavior, the collection of data provides hypotheses for future studies on the function of the neurons implicated in the learning paradigms and provides useful references for similar studies in the field.

## Introduction

Learning and memory processes are presumably universal in the animal kingdom, forming the basis for adaptive behavior. An intriguing form of these behavioral adaptations is known as associative learning, where a link between two unrelated cues is formed. The famous pavlovian dogs set a classical example: These dogs were trained to associate a sound stimulus (the conditioned stimulus, CS) with food (unconditioned stimulus, US). Consequently, the mere auditory cue prompted the dogs to salivate in expectation of their meal (*Pavlov, 1910*).

To synthesize an adaptive associative memory that elicits an adaptive response upon future encounters with the CS, both the CS and the US must be encoded in the neural system. Moreover, their encoding needs to be logically integrated such that the behavioral response will match the expected valence that the CS predicts (*Josselyn and Tonegawa, 2020*). Whether the CS was associated with a positive or negative experience, this valence remains associated with the CS.

**\*For correspondence:**
alonzas@mail.huji.ac.il

**Present address:** †Neuroscience Institute Cavalieri Ottolenghi, Turin, Italy

**Competing interest:** The authors declare that no competing interests exist.

Animals have come up with different strategies for encoding associative memories. In flies, olfactory associative learning is centralized in the mushroom body, where it is distributed among various neurons and synapses to code both the US valence and the CS odorant (*Bilz et al., 2020*; *Roselli et al., 2021*; *Widmann et al., 2018*). Mammalian brains are thought to encode associative memories in a decentralized fashion where interconnected areas, distributed across various areas of the brain, link up to encode memory traces. For example, associative fear memories are thought to be distributed among the amygdala that encodes the valence, the hippocampus, which encodes the context, and the cortical neurons, which provide the specific sensory information (*Josselyn and Tonegawa, 2020*).

In that respect, sensory neurons also proved to play important roles in the formation of associative memories. Their learning-induced neuroplasticity was observed across various sensory modalities (*e.g.* olfactory, gustatory, auditory, and visual), and is thought to confer improved detection and enhanced attention towards important cues encountered in the past (*Åhs et al., 2013*; *McGann, 2015*).

Extracting the principles by which memories are formed within neural networks requires first to identify the brain regions, and preferably, the individual coding neurons. To this end, *Caenorhabditis elegans* worms offer an appealing research system. Their compact nervous system consists of 302 neurons, and a detailed blueprint of all the chemical and electrical connections is available (*Cook et al., 2019*; *White et al., 1986*; *Witvliet et al., 2021*). Moreover, the number and the identity of the neurons are invariant and individual neurons can be identified based on their position and anatomy across different individuals.

Though equipped with a small neural network, *C. elegans* can form both associative and non-associative memories (*Ardiel and Rankin, 2010*; *Loy et al., 2021*; *Sasakura and Mori, 2013*). To form associative memories, worms are typically conditioned with a chemical (e.g. an odorant, salt, or low pH) or a mechanical stimulus in the presence or the absence of food, to form positive or negative associations, respectively (*Adachi et al., 2010*; *Amano and Maruyama, 2011*; *Kauffman et al., 2010*; *Oda et al., 2011*; *Rankin, 2000*; *Torayama et al., 2007*; *Wen et al., 1997*). Following successful conditioning, attraction to the conditioned stimulus is enhanced or reduced depending on whether the training paradigm includes food (positive training) or not (negative training), respectively. Just like in higher organisms, these memories can be classified into short- and long-term memories, where the former last for a couple of hours and the latter may persist for days (*Amano and Maruyama, 2011*; *Kauffman et al., 2010*).

Whole-brain functional imaging can be used to extract the individual neurons whose activity was modulated following memory formation. In *C. elegans*, advanced microscopy techniques allow imaging neural dynamics of the entire network with cellular resolution in both restrained and freely behaving animals (*Hallinen et al., 2021*; *Kato et al., 2015*; *Nguyen et al., 2016*; *Schrödel et al., 2013*; *Toyoshima et al., 2020*; *Venkatachalam et al., 2016*; *Yemini et al., 2021*). In particular, studying neuroplasticity with cellular resolution allows addressing intriguing questions that were hitherto considered mainly based on theoretical grounds. For example, how do individual sensory neurons code and integrate both the stimulus and the valence? How many neural resources are required to form an associative memory, and more fundamentally, are there general organizing principles by which associative memories are encoded within a neural network?

Here, we have systematically studied how the four types of associative memories (positive/negative associations, each formed as a short- or long-term memory) are encoded within the compact neural network of *C. elegans*. We found that short-term, but not long-term, memories are evident in the sensory layer of the animal. Moreover, individual sensory neurons code the CS and/or the US components of the memory in a distributed manner. This information is integrated by the downstream interneurons, which code both the short- and long-term memories. Given the distributed nature of the memory code, it is the combined modulated activity of all the involved neurons that gives rise to the adaptive behavioral response.

## Results
### Establishing four memory-formation paradigms using the same CS

We begin by establishing training paradigms that form robust traces of associative memories (*Figure 1A*). Building on existing protocols (*Amano and Maruyama, 2011*; *Colbert and Bargmann,*

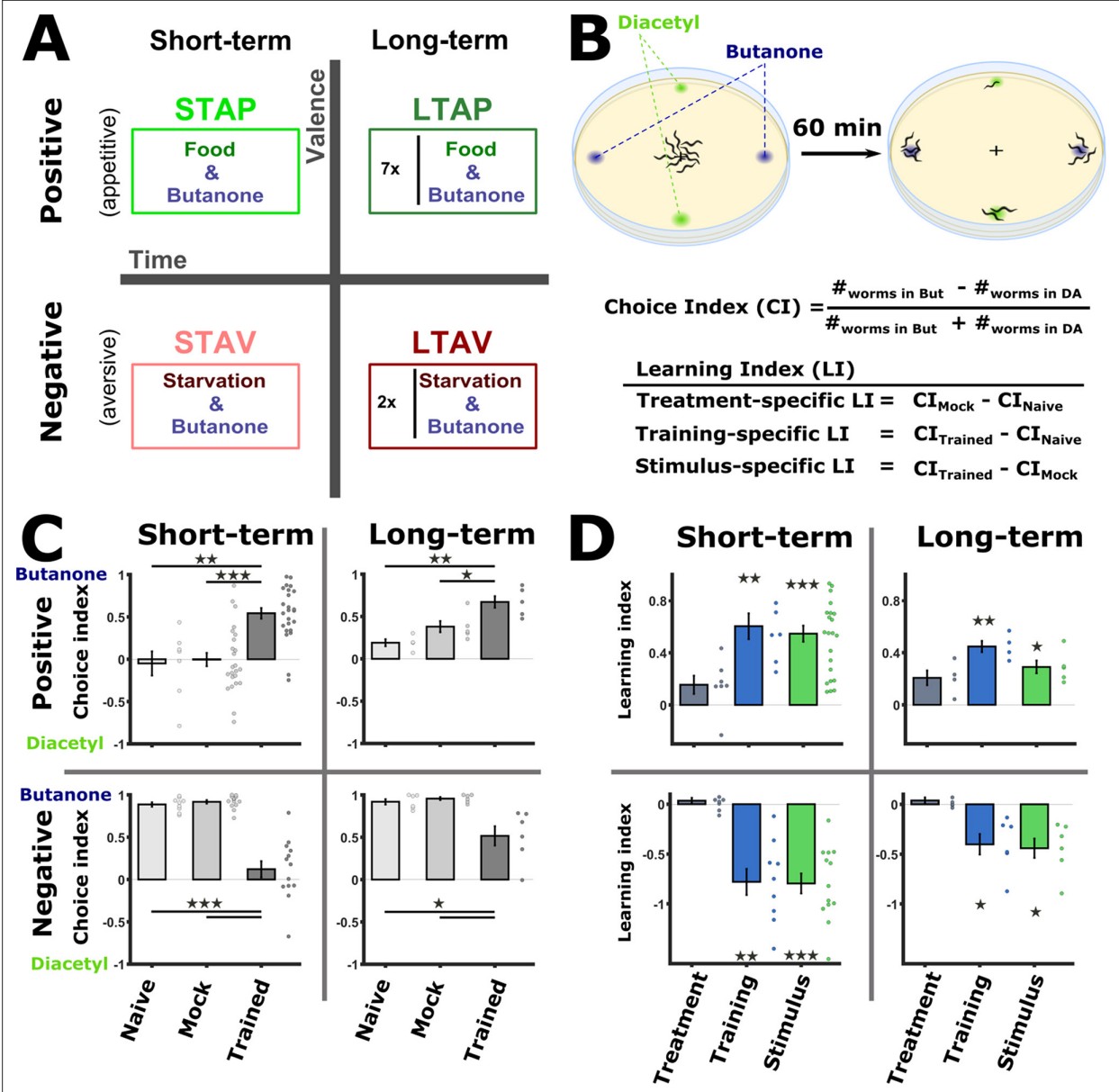

**Figure 1.** Training paradigms that form robust associative memories. (**A**) Worms were trained to form each of the four types of associative memories: Short- and long-term memories (denoted along the horizontal axis), each trained using a positive or a negative unconditioned stimulus (US, vertical axis). Notably, the same conditioned stimulus, butanone (BUT), was used for all types of memory. STAP, short-term appetitive; LTAP, long-term appetitive; STAV, short-term aversive; LTAV, long-term aversive. In the LTAP training, seven rounds of 30 min starvation (no CS) and 30 min on food (+CS) were used. For LTAV training, two rounds of pairing starvation with BUT (each for five hours) were required. (**B**) A two-choice assay was used to quantify animals' preference towards the conditioned stimulus BUT (against an alternative attractive choice, diacetyl). Scoring the number of worms reaching each of the choices provided the Choice Index (CI), which ranges from –1 (denoting complete aversion to the CS) to +1 (full attraction). Choice tests for positively- and negatively trained animals differed in concentrations and layout (**Figure 1—figure supplement 1A**) because of valence-specific effects on choice behavior (**Figure 1—figure supplement 2**). Learning indices (LIs), calculated based on these CIs, show the treatment-, stimulus-, and training-specific effects on the animals' choice (**Figure 1—figure supplement 3**). (**C**) CI values as scored following the behavioral choice assays. Positively trained animals increased attraction while negatively trained animals reduced attraction towards BUT. (**D**) LIs calculated according to the equations provided in (**B**) on the data shown in (**C**). Significant stimulus- and training-specific LIs in all paradigms indicate experience-dependent modulation of behavior that is based on stimulus and valence. LIs were calculated by comparing experiments performed on the same day only (**Figure 1—figure supplement 4** and Materials and methods). Experimental repeats (in C&D) were performed on different days and range between 4 and 21. Each experimental repeat is the average of three assay plates, each scoring 100-150 worms. Error bars indicate SEM. *p<0.05, **p<0.01, ***p<0.001 (one-sample t-test, FDR corrected; significant differences from the zero LI values).

The online version of this article includes the following figure supplement(s) for figure 1:

*Figure 1 continued on next page*

Figure 1 continued

**Figure supplement 1.** Behavioral choice assays using ethanol as the alternative choice reproduced the results obtained using diacetyl.

**Figure supplement 2.** Appetitive and aversive training paradigms shift the preference to butanone.

**Figure supplement 3.** Inferring memory components by comparing different experimental groups.

**Figure supplement 4.** The behavioral choice assays are subject to high day-to-day variability and a same-day difference analysis reduced this variation.

*1995*; *Kauffman et al., 2011*), we trained *C. elegans* worms to form each of the four types of associative memories: short-term aversive (STAV), short-term appetitive (STAP), long-term aversive (LTAV), and long-term appetitive (LTAP). Notably, we used the same CS (the odorant butanone, BUT) for all training paradigms. This allowed us to extract memory traces that are unique to the training paradigm and that are independent of the specific CS used.

To form positive (appetitive) or negative (aversive) associations, we exposed the worms to BUT in the presence or the absence of food, respectively. To form short-term memories, for which behavioral changes last for up to 2 hr (*Kauffman et al., 2010*), we trained the worms for 1 hr or 90 min for appetitive or aversive conditioning, respectively, and assayed the animals within 30 min following the training period. For long-term memories, which typically last for 1–2 days (~10% of the worms' lifespan), we used a repetitive-training protocol that lasted for ~12 hr and then assayed the animals 14 hr post the training period (*Figure 1A*, see also Materials and methods for details). In parallel to the trained animals, we always included in the analysis two important control groups: mock-trained animals (animals that underwent training in the absence of the CS BUT) and naive animals, which were left untreated.

To verify that these training paradigms form robust memory traces, we analyzed the attraction of trained animals to BUT, the CS (*Figure 1B*). For this, we used a standard two-choice assay, where worms were free to choose between the CS and an alternative attractant, diacetyl (DA). Based on the worms' distribution after 1 hr, we calculated the Choice Index (CI) which provides a quantitative measure for the animals' preference towards the CS over the alternative (*Figure 1B* and *Figure 1— figure supplement 1A*).

Following training, positively- and negatively trained animals shifted their preferences towards the CS in a concentration-dependent manner (*Figure 1—figure supplement 2*): Aversively trained animals decreased attraction towards the CS at lower concentrations, while the attraction of positively trained animals increased only at higher concentrations of the CS BUT (*Figure 1—figure supplement 2*). Thus, for chemotaxis assays, we used a $10^{-1}$ BUT concentration for positively trained animals, and a $10^{-3}$ concentration for the negatively trained animals (*Figure 1—figure supplement 1A*).

Worms trained to form positive associations were significantly more attracted to BUT than mock-trained or naive animals (*Figure 1C*). Similarly, worms trained to form negative associations were significantly less attracted to BUT when compared to mock-trained or naive animals (*Figure 1C*). These behavioral changes were evident in both short- and long-term training paradigms. Similar results were also obtained when assaying with ethanol, which was used to dilute BUT, instead of DA as the alternative choice (*Figure 1—figure supplement 1B*).

We next quantified the explicit effects of the CS (BUT) and the US valence (starvation/appetitive experiences) on the behavioral output. For this, we used the CI values to compute the Learning Index (LI), which reflects the difference between the CI of the different experimental groups within a training paradigm (*Figure 1B* and Methods). The treatment itself (US, $CI_{mock}$ - $CI_{naive}$) led to negligible changes in the choices (*Figure 1D*). By contrast, significant changes in the choice, of both training-specific ($CI_{trained}$ - $CI_{naive}$) and stimulus-specific (CS, $CI_{trained}$ - $CI_{mock}$), were observed across all training paradigms, indicating that the modulated behavior was due to the presence of the CS (BUT) (*Figure 1—figure supplements 2–4*). Furthermore, the change in the choice corresponded to the valence of the experience: appetitively trained animals increased their choice of BUT, while aversively trained animals decreased their choice (*Figure 1D*). Thus, the change in choice behavior was dependent on both the stimulus and the valence of the experience.

Taken together, using the same CS (BUT), the four types of associative memories could be robustly formed: positive (appetitive) experiences which increased attraction towards BUT, and negative (starvation) experiences which decreased this attraction. Both positive and negative associations were successfully formed for short- and long-term periods.

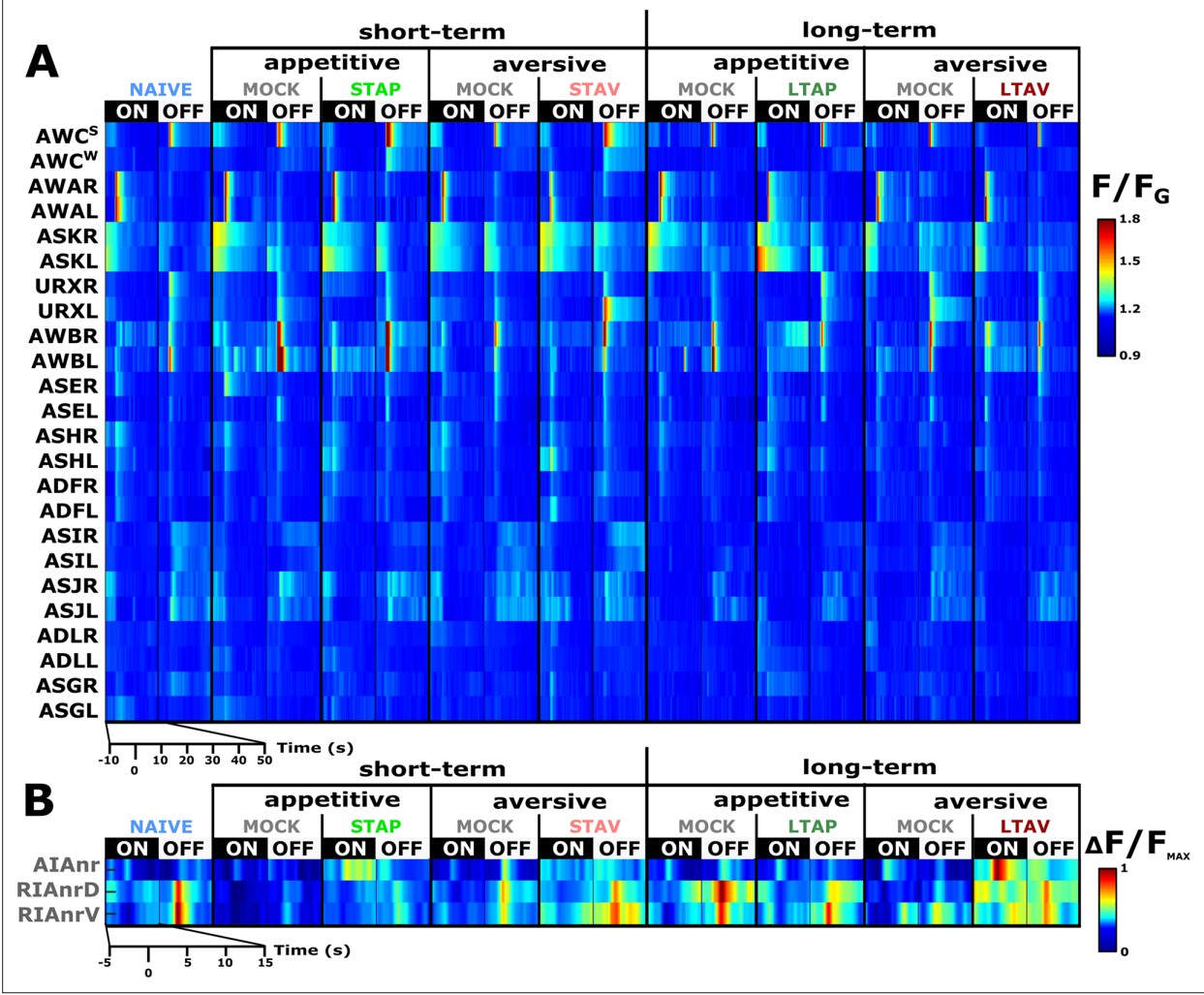

**Figure 2.** A comprehensive functional analysis of the chemosensory system including key interneurons. (**A**) A comprehensive systematic analysis of neural dynamics in naive, trained, and mock-trained animals across all four training paradigms (STAP, STAV, LTAP, LTAV). Neural dynamics were measured following exposure to (ON) or removal of (OFF) the conditioned stimulus butanone. Shown are the mean activities of the chemosensory neurons. The color bar indicates fluorescence normalized by the ground state of the neuron (see Methods for details). Statistical analysis suggested that at the population-level, activities of AWC$^W$ and AWC$^S$ correspond to AWC$^{ON}$ and AWC$^{OFF}$, respectively (see *Figure 2—figure supplement 3A–D* for a detailed analysis). (**B**) Mean activity of the RIA and AIA interneurons. Activities were extracted from the neurites. For RIA, we analyzed activity within the dorsal and the ventral regions of the neurite (see also *Figure 2—figure supplement 1C and D* and Materials and methods). Due to large amplitude differences between interneurons, the color bar indicates fluorescence normalized by the maximal fluorescence ($\Delta F/F_{max}$). nr, nerve ring; nrD/V, dorsal/ventral sides of the nerve ring; In both panels, presented are the means of 9–17 animals per each experimental group (column), resulting in a coverage of 2-17 traces per neuron (median=13). Only neurons with at least six traces were analyzed.

The online version of this article includes the following figure supplement(s) for figure 2:

**Figure supplement 1.** Identifying the individual chemosensory neurons in the pan-chemosensory reporter strain.

**Figure supplement 2.** Interneurons anatomy and the region of extracted activity.

**Figure supplement 3.** Discriminating between the two bilateral AWC neurons.

**Figure supplement 4.** Discriminating between the URX and CEPD neurons: Butanone removal elicits responses in the URX neurons, but not in the CEPD neurons.

**Figure supplement 5.** Neural activities of naive animals.

## A comprehensive functional analysis of the chemosensory system following memory formation

We systematically analyzed neuroplasticity of the sensory system, naturally focusing on the chemosensory neurons (*Figure 2—figure supplement 1*) as well as on two of the main downstream interneurons AIA and RIA (*Figure 2* and *Figure 2—figure supplement 2A and B*). The AIA neuron is directly and richly innervated by chemosensory neurons, and hence, forms a hub in processing sensory output (*Witvliet et al., 2021*). RIA integrates sensory signals with the head position, thereby affecting choice of locomotion directionality (*Hendricks et al., 2012*; *Hendricks and Zhang, 2013*; *Ouellette et al., 2018*). We measured neural activity for all the four training paradigms (*Figure 1A*), including the matched mock-trained control groups and naive animals. For this, we used several reporter strains, each expressing GCaMP in either a set or in individual types of neurons (**see Materials and methods**). We restrained the animals in a microfluidic device (*Chronis et al., 2007*) and allowed them to habituate to the imaging conditions before using a fast-scanning confocal system to image dynamics from individual neurons during exposure and removal of the CS butanone (*Figure 2*). Neural identities were unambiguously extracted using available anatomic maps (*Durbin, 1987*; *White et al., 1986*) and by comparing to reporter strains with known neural identities (*Figure 2—figure supplements 3 and 4*).

## Short-term memories are evident in the sensory- and the inter-neurons, while long-term memories are evident primarily in the interneurons

Previous studies showed that the chemosensory neuron AWC[ON] responds to BUT already at the naive state of the animals (*Kato et al., 2014*; *Larsch et al., 2013*). We found that additional chemosensory neurons participate in encoding BUT in naive animals, namely, AWA, ASH, AWB, ASJ, URX, as well as the interneurons AIA and RIA (*Figure 2* and *Figure 2—figure supplement 5*). However, neurons that participate in memory coding presumably show modulated response activities following training. To extract these neurons, we calculated the difference in the mean amplitude response between all possible pairwise comparisons of the various experimental groups, namely, naive, trained, and mock-trained animals, for each of the four training paradigms (*Figure 3A–B*).

For example, while the AWC[W] neuron (identified as AWC[ON], see *Figure 2—figure supplement 3*) shows mild innate responses to BUT in naive and mock-trained animals, its response activity was significantly heightened following short-term appetitive (STAP) and short-term aversive (STAV) training paradigms (*Figure 3C–F*). Since the sole difference between the trained and the corresponding mock-trained animals was the presence of the CS (BUT) during the training period, the differential activity suggests that the AWC[W] neuron may be coding the stimulus component of the memory. No significant differences were observed in the responses of the AWC[W] neuron when comparing STAP and STAV-trained animals, further strengthening the possibility that this neuron codes the stimulus rather than the valence component of the memory. A similar change in AWC[ON] neural activity was also observed by Cho and colleagues (*Cho et al., 2016*). Interestingly, this training-associated increased activity depended on intact neurotransmission (*Figure 3—figure supplement 1*), suggesting that network activity, rather than cell-autonomous processes, underlie the response plasticity in the AWC[ON] neuron.

In addition to AWC[ON], the neurons AWC[OFF], ASH, and ASK also showed significant modulated responses (*Figure 3A* and *Figure 3—figure supplements 2 and 3*). Most striking, the changes in neural responses were observed predominantly for the short-term training paradigms, while only minor negligible changes were detected following long-term training paradigms (compare the two rightmost sectors in *Figure 3A*).

In contrast to sensory neurons, interneurons showed marked changes in their responses following training in both short- and long-term paradigms (*Figure 3B* and *Figure 3—figure supplements 4 and 5*). For example, we observed significant increased activity in the ventral compartment of the RIA neurons as well as in the AIA neurites following training in the long-term negative paradigm (*Figure 3—figure supplement 4*).

Together, these results indicate that sensory responses to the CS were primarily modulated following formation of short-term, rather than long-term, memories. Moreover, this training-induced plasticity was distributed across various chemosensory- and inter- neurons.

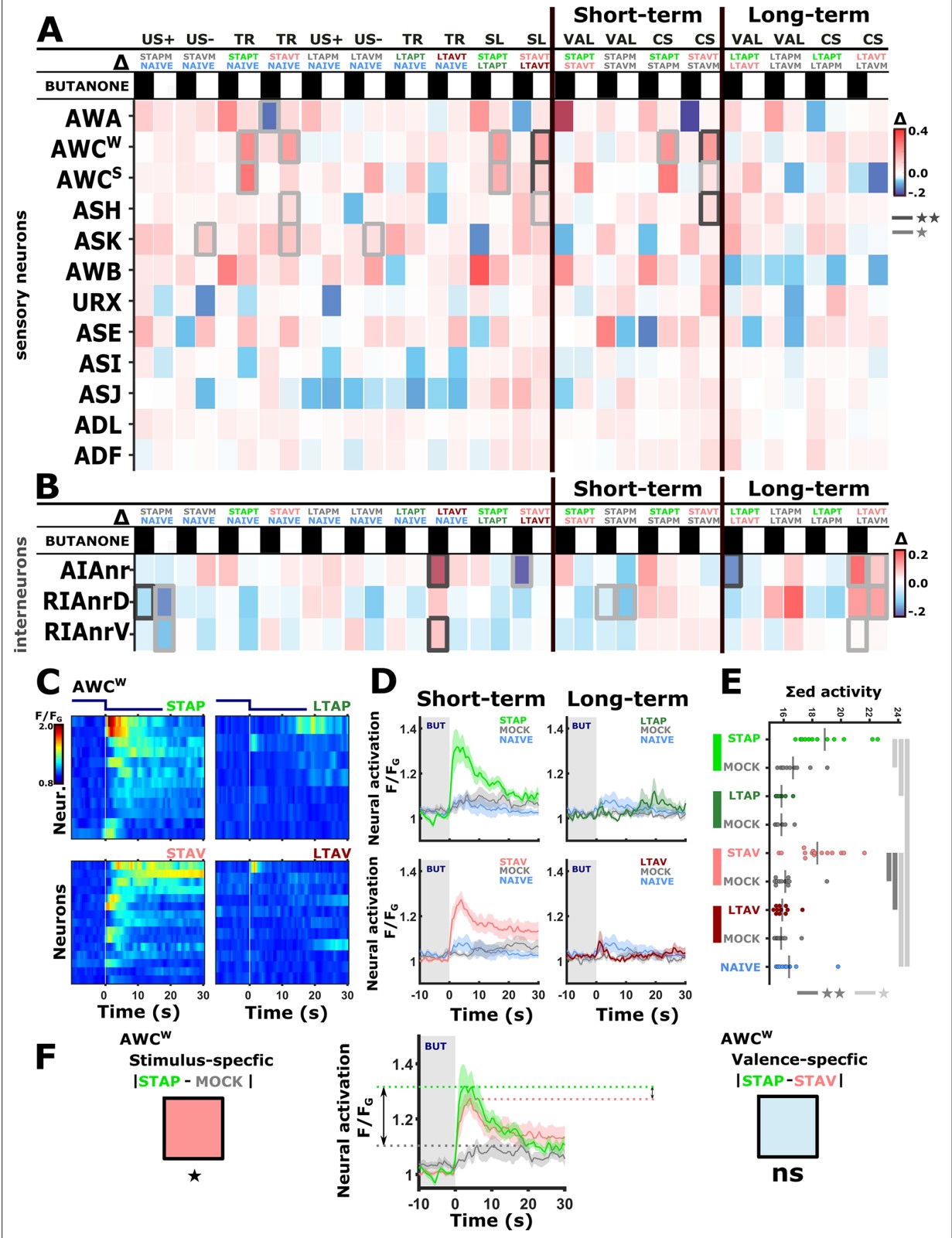

**Figure 3.** The activity of sensory neurons is predominantly modulated following short-term training paradigms, while the activity of interneurons is modulated following both short- and long-term training paradigms. (**A–B**) Differences in the mean maximal amplitudes of neural activities across the different experimental groups. Differences between pairwise groups are denoted by the Δ in the header. (**A**) Chemosensory neurons. (**B**) Interneurons with nr(D/V) meaning neurite (Dorsal/Ventral). The leftmost sector denoted all possible comparisons that provide the different coding measures. For

*Figure 3 continued on next page*

Figure 3 continued

example, US + compares the positively mock-trained animals to naive animals to yield the coding specific of the positive unconditioned stimulus. TR, treatment-specific coding; SL, short- vs long- term. The middle and the rightmost sectors denote short-term and long-term specific differences including the valence (VAL) and the conditioned stimulus (CS) specific coding or the short and long-term memory, respectively. Black or white rectangles denote exposure or removal of butanone (BUT), respectively. Note that for sensory neurons (**A**) higher amplitude differences are observed predominantly in short-term paradigms. In interneurons (**B**), high amplitude differences are observed for both short- and long-term training paradigms. Colorbar denotes the difference in the maximal amplitude of the two neural responses. Rectangles marked with a border line are significant differences: Gray * p<0.05, Black ** p<0.01. (**C–E**) The sensory neuron AWC^W shows a significant differential activity following formation of short-term memories. Analysis suggests that the AWC^W population activity corresponds to the one of AWC^ON (**Figure 2—figure supplement 3A–D**). (**C**) Heat maps of individual neurons (=animals) denoting neural activities in each of the four training paradigms. The vertical white line at t=0 denotes the time of BUT removal. (**D**) Mean activities, based on data from C. The different colors denote the experimental group in each of the four paradigms. The shaded gray background indicates BUT exposure, and shaded area around the mean activity indicates standard error of the mean. (**E**) Integrated activity during the first 10 s following BUT removal (based on the dynamics shown in C, n=7-15). Black vertical lines denote the mean of summed activity, and the dots present the individuals. *p<0.05, **p<0.01, ***p<0.001 (t-test, FDR corrected for multiple comparisons). (**F**) Two examples of mean AWC^W responses comparing different experimental groups. Green, Short-term appetitive (STAP); Red, Short-term aversive (STAV); Gray, appetitively mock-trained animals. Comparing response dynamics of STAP and the associated mock controls reveals a significant difference (denoted on the left side). This difference marks a stimulus-specific component of the memory. In contrast, the difference between STAP and STAV is negligible (shown as a blue box on the right), suggesting that the AWC^ON neuron does not code the valence component of the memory. The boxes' color code matches the different colors shown in panel A. The three dynamic curves shown here are taken from panel D to highlight the groups being compared.

The online version of this article includes the following figure supplement(s) for figure 3:

**Figure supplement 1.** The gained responses in AWC^ON following short-term appetitive training require intact synaptic transmission.

**Figure supplement 2.** The sensory neurons whose activity responses were modulated following short-term, but not long-term, training paradigms.

**Figure supplement 3.** Individual activity traces of chemosensory neurons in naive and mock-controlled animals.

**Figure supplement 4.** Interneurons exhibited modulated activities following both short- and long-term training paradigms.

**Figure supplement 5.** Individual activity traces of interneurons in naive and mock-controlled animals.

## Sensory neurons code both the stimulus and the unconditioned stimulus in short-term memories

The systematic functional analysis revealed small, though significant, differences in neural responses (**Figures 2–3**). This is in contrast to the differences observed in behavioral outputs that showed a strong experience-dependent shift in preference towards the CS when presented with an alternative choice (**Figure 1** and **Figure 1—figure supplement 1B**). Thus, to better resolve potential neural differences in encoding aversive and appetitive memories, we mimicked the conditions set during the behavioral assays and imaged neural responses using DA as the alternative choice to the CS BUT. We repeated the comprehensive functional imaging experiments, this time focusing on the short-term training paradigms in which the chemosensory neurons showed extensive modulated activities (**Figure 4A–B** and **Figure 4—figure supplement 1**). Strikingly, a systematic analysis of naive, trained, and mock-trained animals following positive and negative training paradigms revealed significant activity changes in most of the chemosensory neurons (**Figure 4A–B** and **Figure 4—figure supplements 2 and 3**). In particular, three neurons (AWA, ASER, and AWC^ON) stood out with marked differences in their activity (**Figure 4B–J** and **Figure 4—figure supplement 4A–F**).

In naive and mock-trained animals, the AWA neurons strongly responded upon switching from DA to BUT (**Figure 4B–D**). However, in trained animals (positive and negative), the AWA neurons showed no responses, suggesting that these changes code the stimulus component. When switching back from BUT to DA, the AWA neurons in trained animals were activated. This activation was significantly stronger in STAV-trained than in STAP-trained animals (**Figure 4—figure supplement 4D–F**), suggesting differences between aversive and appetitive encoding as well. Repeating these experiments in an additional AWA reporter strain similarly showed stimulus and valence specific modulations, although the response dynamics differed from the one observed in the pan-sensory reporter strain (**Figure 4—figure supplements 4 and 5**).

The ASER neuron displayed marked activity responses in naive animals upon the switch from DA to BUT. This response was completely lost in STAP-trained and all mock-trained animals, but not in STAV-trained animals (**Figure 4E–G**). This suggests that the ASER neuron may be coding the stimulus component of the memory as well as the positive (US+) and the negative (US-) experiences of the training paradigms. Activity changes in the AWC^ON neurons were hallmarked by a large increase in all

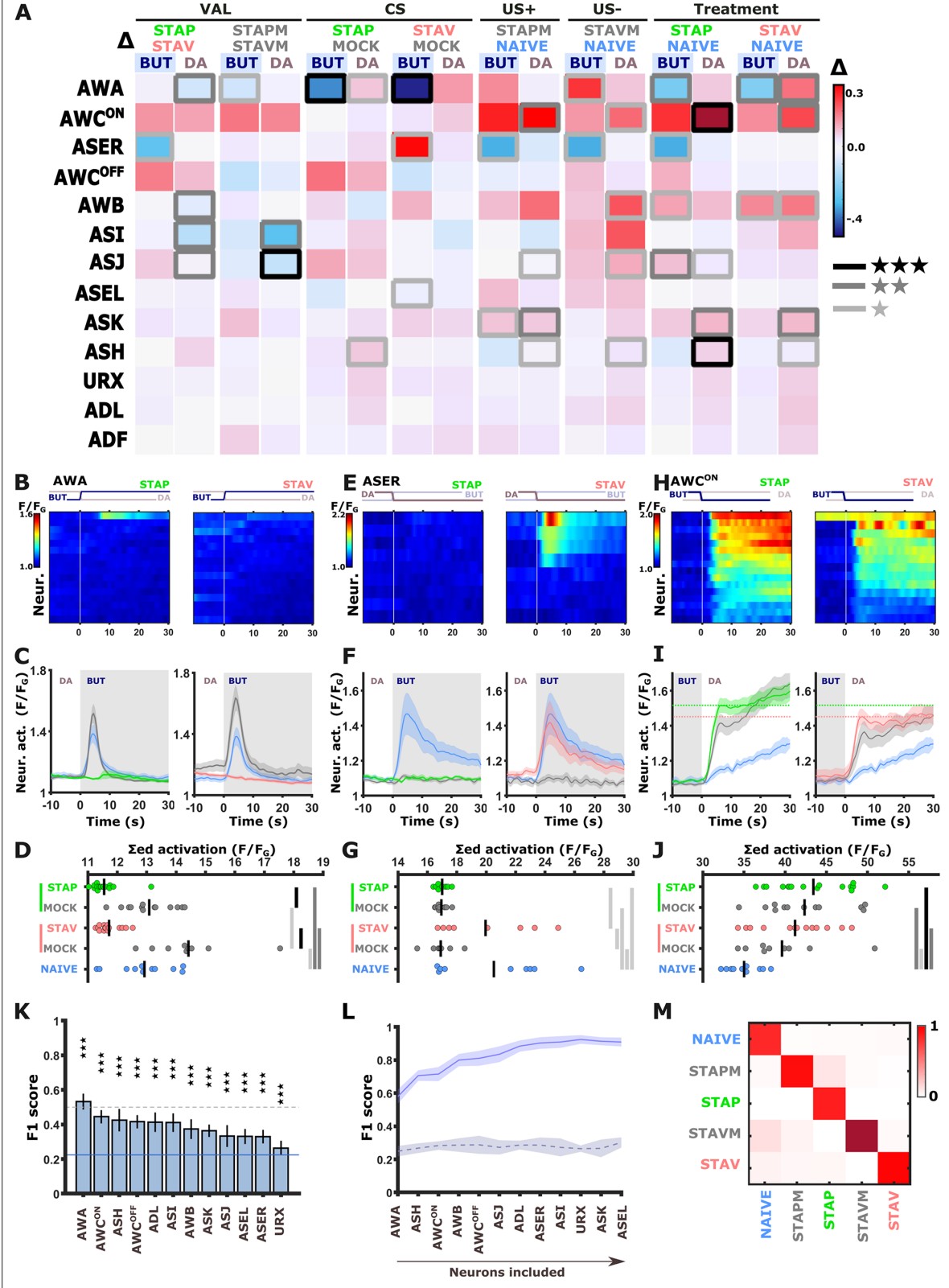

**Figure 4.** Short-term training broadly modulates activity of the sensory neurons. (**A**) Differences in the mean maximal amplitudes of neural activities across the different experimental groups. Differences between pairwise groups (denoted by the Δ in the header) denote the different coding-specific measures. Rectangles marked with a border line denote significant differences: Light gray * p<0.05, dark gray ** p<0.01, Black ***p<0.001. (**B–J**) The sensory neurons, AWA (B-D, n=9–16 animals), ASER (E-G, n=8-10 animals), AWC^ON (H-J, n=9-16 animals, see *Figure 2—figure supplement 3E–L*),

*Figure 4 continued on next page*

*Figure 4 continued*

show significant differential activities following the formation of short-term memories. Heat maps (**B, E, H**) denote activities of individual neurons in each of the training paradigms, and vertical white lines at t=0 denote the time of stimulus exchange BUT/DA. Line plots (**C, F, I**) show mean activity with SEM (shaded color). The different colors denote the trained animals in a given paradigm. Blue, naive animals; Gray, mock-trained animals. Shaded gray rectangles indicate butanone exposure. Dotted lines in panel I (AWC^ON) denote the maximal amplitude of aversively and appetitively trained animals. Dots in group-scatter graphs (**D, G, J**) represent the summed neuronal activity post the stimulus switch; black lines denote the population mean. *p<0.05, **p<0.01, ***p<0.001 (t-test, FDR corrected). (**K**) Classification accuracy of the different training conditions when considering response dynamics of individual neurons. Shown are the macro F1 scores following using a random forest classifier. Blue line denotes the score of a random classification, indicating that when considering single neurons, the accuracy is better than randomly expected. Dash line indicates an F1 score of 50%. Error bars indicate standard deviation from a cross validation. ***p<0.001 (t-test against random F1 score, FDR corrected). (**L**) Classification accuracy of the different training conditions when considering a growing number of neurons to be included in the model (indicated by the horizontal arrow). Continuous purple line denotes the real data; dashed black line is when using a scrambled data where neuronal responses were randomly assigned to various training paradigms. Shaded area around the lines indicates standard deviation from cross-validation. (**M**) Line-normalized confusion matrix for the data. When including activity from all sensory neurons, the classification efficiency exceeds 90%. Scores of true positives, positioned along the diagonal, range from 81% to 97%.

The online version of this article includes the following figure supplement(s) for figure 4:

**Figure supplement 1.** A comprehensive functional analysis of the chemosensory system and key interneurons following training in short-term paradigms.

**Figure supplement 2.** Chemosensory neurons showing modulated activities following short-term training paradigms.

**Figure supplement 3.** Individual activity traces of chemosensory neurons in naive and mock-controlled animals.

**Figure supplement 4.** The two AWA reporter lines differ in their response kinetics but carry the same memory-coding logic.

**Figure supplement 5.** The WT and the pan-chemosensory reporter strain show similar behavioral outputs following training.

**Figure supplement 6.** ASER and AWA neurons show high animal-to-animal variability.

**Figure supplement 7.** Activity of sensory neurons can be used to identify the training conditions by classification algorithms.

treated groups (when compared to the naive group), which is indicative of coding both the aversive and appetitive US (*Figure 4H–J*, *Figure 2—figure supplement 3E–H*). Activity changes of other sensory neurons were more subtle: for example, ASI, AWB, and ASJ neurons appeared to be related to the differences between aversive and appetitive experience (*Figure 4—figure supplements 2 and 3*).

In some paradigms, the AWA and ASER neurons exhibited marked variability in their responses (*Figure 4—figure supplement 6A–B*). For example, the ASER neuron responded in only ~50% of the naive or STAV-trained animals. This bi-modal distribution was observed when considering both trial-to-trial and animal-to-animal responses (in the following, the term 'trial' refers to individual BUT/DA exchanges, where animals underwent six cycles of such trials (exchanges), *Figure 4—figure supplement 6A*). Interestingly, this variability decreased following training (e.g. STAP and the associated mock group) as most animals (or trials) showed homogeneous responses. Other neurons did not show a marked animal-to-animal variability (*Figure 4—figure supplement 6D–G*).

As multiple chemosensory neurons changed activity following training in each of the paradigms, we next asked how many neurons need to be considered to accurately describe each of the training-associated states such that their combined activity can distinguish between these training paradigms. We trained several classifiers (including k-means, random-forest, and a neural net) on a fraction of the data, after which we tested the model accuracy on the remaining data (see Materials and methods). All classifiers provided similar results (*Figure 4L–M* and *Figure 4—figure supplement 7E and F*): When considering single neurons only, classification accuracy (measured as F1 scores) was rather low for all sensory neurons (up to 50%), though the scores were significantly higher than randomly expected (*Figure 4K* and *Figure 4—figure supplement 7*). When considering sets of sensory neurons, the decoding of the underlying training paradigm was better the more neurons were added to the model. Combining activities of all chemosensory neurons together resulted in 90% decoding efficiency, irrespective of the classification algorithm used (*Figure 4L–M* and *Figure 4—figure supplement 7E–F*).

These results demonstrate that the different training paradigms induce fine changes in a large fraction of the sensory neurons, suggesting that short-term memories are widely distributed across the chemosensory system, where each neuron may code the CS and/or the positive/negative experiences of the training paradigm.

## The modulated activity in the sensory neurons propagates to the interneurons

We next asked how these fine modulated activities are reflected in the downstream interneurons. The chemosensory neurons are presynaptic to several key interneurons, including AIY, AIA, and RIA (*Cook et al., 2019*; *White et al., 1986*; *Witvliet et al., 2021*). In particular, the AIY neurons are key immediate postsynaptic targets of the AWA, ASE, and AWC neurons that showed marked modulated activities following training. We therefore repeated the short-term training paradigms in strains expressing GCaMP in these interneurons (see Materials and methods). Activity profiles of these interneurons were extracted from their neurites since the soma activity remained largely static (*Figure 5—figure supplement 1*), consistent with previous reports (*Chalasani et al., 2010*; *Hendricks and Zhang,*

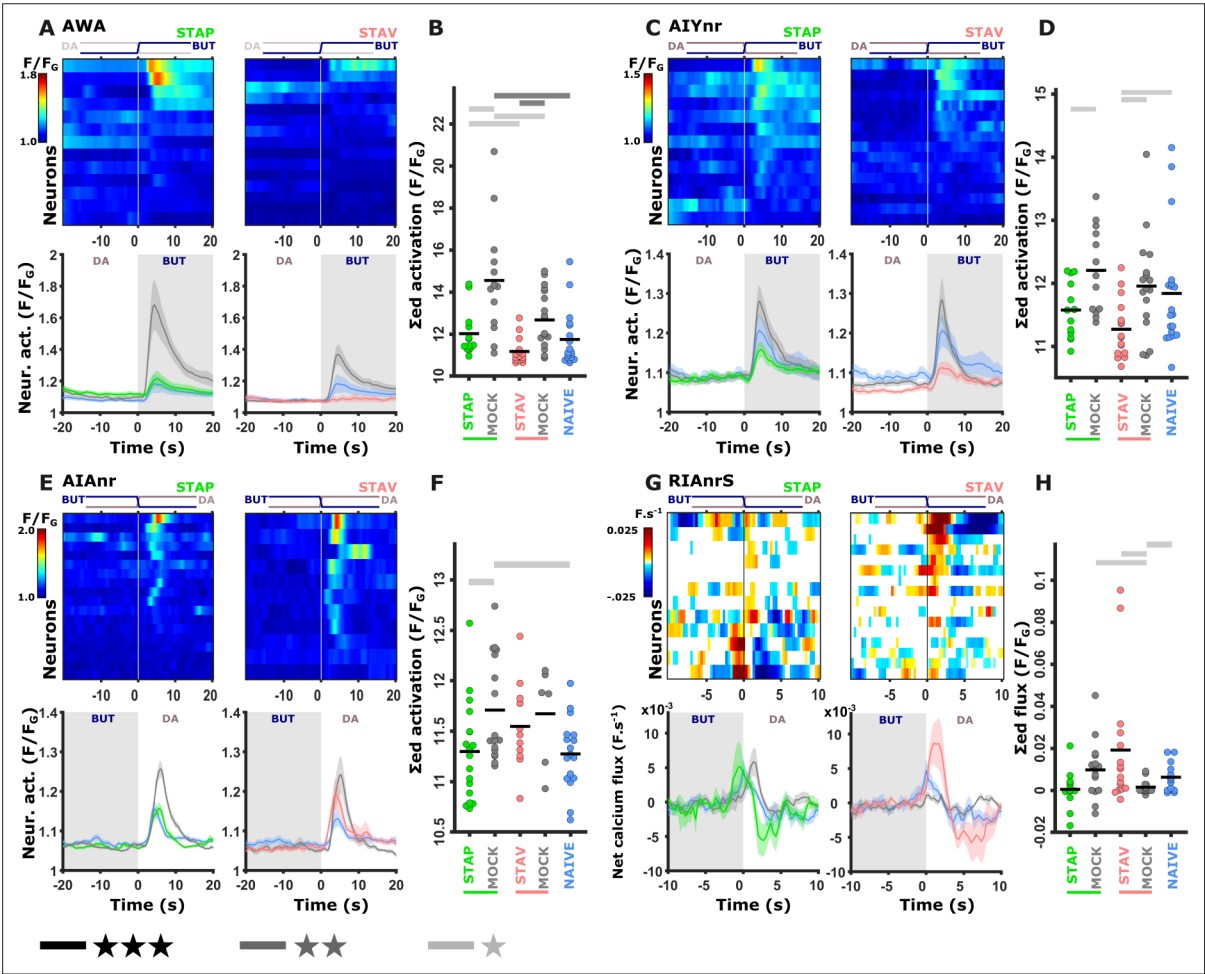

**Figure 5.** Training-induced modulated activity of the interneurons. (**A–D**) AWA and AIY activities measured simultaneously from the same animal. Note the similarity between the mean activities of the AWA (**A**) and AIY (**C**) neurons in trained animals (n = 13–19 animals). (**E–F**) AIA activities across all training paradigms. (**E**) Individual traces of trained animals (top) and mean activities (bottom). AIA Activity was significantly reduced following positive-associated training only (n = 6–18 animals). (**G–H**) Sensory-evoked component of the time-derivative of the neural activity *Hendricks et al., 2012*; *Jin et al., 2016* of RIA across all training paradigms. RIA Activity was significantly increased following negatively associated training only (n=12–16 animals). (**A, C, E, G**) Heatmaps show responses of single neurons. Line graphs show the mean activity. Colors denote the training paradigm. Shaded areas in the line plots indicate presence of the CS butanone. (**B, D, F, H**) Integrated activity during the first 10 s following stimulus exchange. Black horizontal lines denote the mean of summed activity, and the dots present the means of each individual trials. Significance is according to the bar color: *p<0.05, **p<0.01, ***p<0.001 (t-test, FDR corrected).

The online version of this article includes the following figure supplement(s) for figure 5:

**Figure supplement 1.** Stimulus-induced calcium dynamics is observed in the neurites of AIA and AIY, but not in the cell soma.

**Figure supplement 2.** The individual activity traces and the mean activities of RIA and AIY interneurons in all training paradigms.

**Figure supplement 3.** Activities in the AVA and AVE neurons were not modulated following training in the different paradigms.

*2013*; *Itskovits et al., 2018*). Of note, for the AIY interneuron, we used a line expressing GCaMP in both the AIY and the AWA neurons, allowing simultaneous recordings of these two neurons from the same animal (*Itskovits et al., 2018*).

Neural responses in the AIY neurons largely recapitulated the responses observed in the AWA neurons (*Figure 5A–D* and *Figure 5—figure supplement 2M–P*). This suggests that the AIY neurons, similarly to the AWA neurons, might encode the stimulus component of the memory. The activity in the AIA neurons was significantly increased in positive mock-control animals while naive and positively trained animals responded similarly, indicating that AIA neurons are probably sensitive to the treatment itself (US+) rather than to the stimulus (*Figure 5E–F* and *Figure 5—figure supplement 1E–H*). A similar tendency was observed in the aversive training paradigm.

To describe the sensory-evoked responses in the RIA neurons, we followed previous reports and considered the derivative form of the activity (*Hendricks et al., 2012*; *Jin et al., 2016*). This sensory-evoked component was significantly increased following aversive experiences only, where the trained animals gained neural responses whereas naive and mock-trained animals showed minimal activity (*Figure 5G–H* and *Figure 5—figure supplement 2I–L*). These results suggest that the RIA neurons show stimulus-specific activity modulation in aversive conditions.

We also studied activity changes in two major command neurons, AVE and AVA, that are positioned downstream to the aforementioned interneurons, and whose activity instructs a backward motion (*Gray et al., 2005*; *Piggott et al., 2011*). These neurons exhibited mostly baseline-level activity shifts that were unrelated to the switches between BUT and DA (*Figure 5—figure supplement 3*).

## The activity of interneurons can be explained by a linear combination of the sensory neuron's activities

Measuring dynamics of AWA and AIY neurons simultaneously from the same animal revealed a surprisingly low correlation between the two synaptic partners. In fact, only ~4–44% of the variance in AIY could be explained by the AWA activity across the different training conditions (*Figure 6—figure supplement 1D–E*). This suggests that additional neurons contribute to the overall dynamics of AIY, and possibly of other interneurons as well.

To study how the modulated activity of the sensory neurons impacts the activity of the interneurons, we considered a simple mathematical model where interneuron dynamics is dictated by a linear combination of the sensory neuron activities. For this, we averaged activities of each odor trial for each of the sensory neurons across all animals in the different paradigms (*Figure 6—figure supplement 1I*), and used a multivariate regression analysis to extract the weights that would best fit the activity of the AIY neurons (see Materials and methods). As expected, the more sensory neurons added to the model, the better was the overall prediction of the AIY activity (*Figure 6A and D*). When considering the combined activity of five sensory neurons types (namely, AWA, AWC, ASE, AWB, and ASG), up to 88% of the variance in the AIY neurons activity could be explained (*Figure 6A–D*). However, this improved accuracy was detected for the naive and the aversively trained animals (*Figure 6A*, pink arrows), while no improvement (compared to when considering AWA alone) was detected for the positively trained animals and the associated mock controls. This lack of improvement suggests that ASER, AWC, ASG, and AWB neurons contribute more to encoding the aversive experiences, while the AWA neurons were the prime contributors to the downstream AIY activity in appetitive experiences (*Figure 6C and D*). Overall, the highest portion of variance that explains AIY activity was obtained in the mock-trained animals, suggesting that the US experience alone may dominate the sensory-to-AIY input weights. This is particularly evident for the appetitive conditioning where the presence of food alone yielded a better fit to AIY activity than the training regime consisting of food and BUT combined. Model evaluation using F-statistics and cross validations indicated that different combinations of sensory neurons should be used to best explain the AIY activity in each of the training paradigms (*Figure 6—figure supplement 1H*). Together, these analyses indicate specific and distinct synaptic routes between the chemosensory neurons and the postsynaptic AIY interneuron are modulated in a paradigm-specific manner (*Figure 6E*).

## Sensory neurons and interneurons jointly code short-term memory

The analyses above indicated that neural activities were significantly modified in an experience-dependent manner. We next asked whether these combined changes provide a unique coding scheme

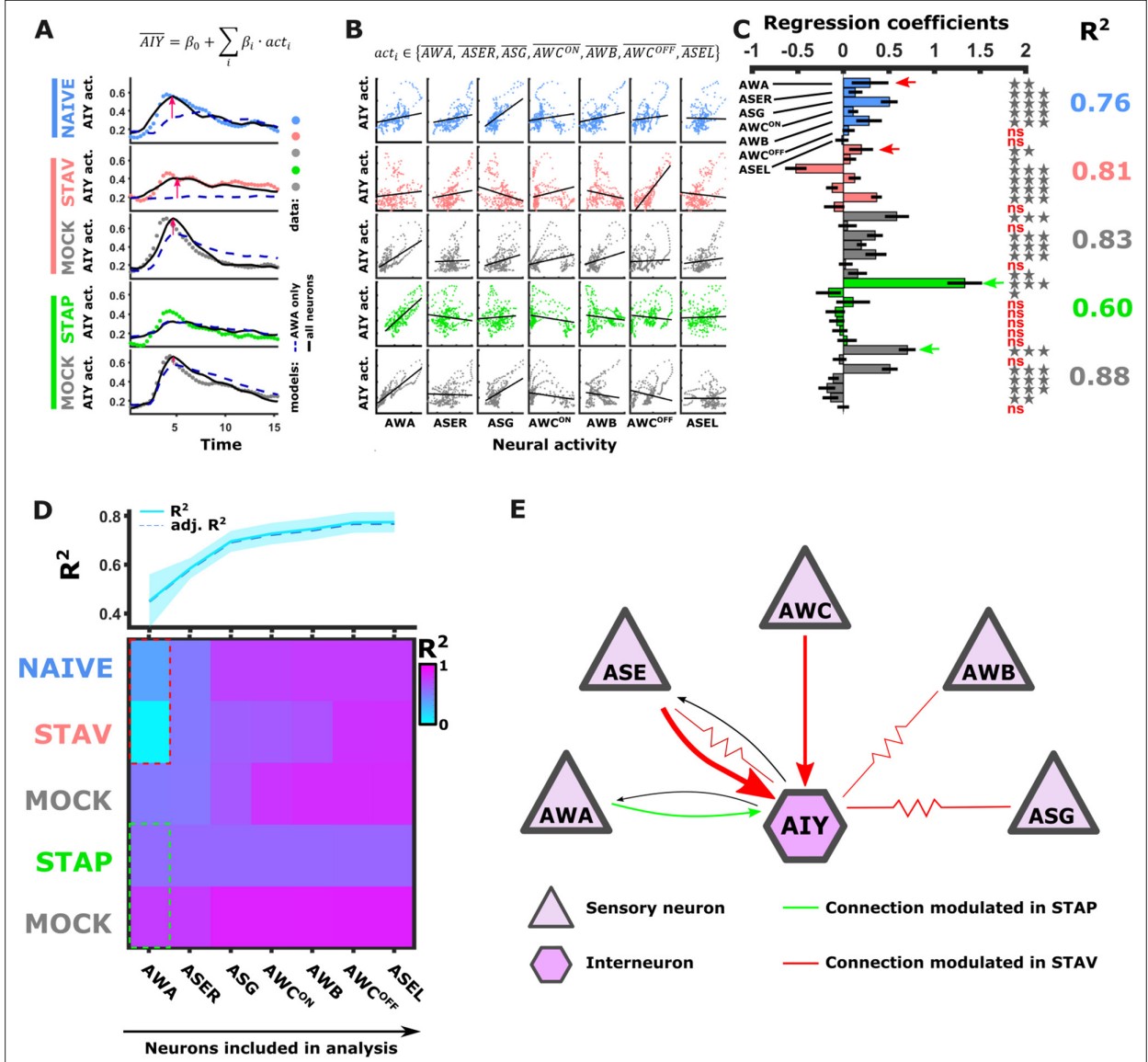

**Figure 6.** Activity of AIY interneurons can be described as a linear combination of the sensory neurons' activities. (**A–C**) A multivariate regression analysis was used to explain AIY activity based on a linear combination of the sensory neurons activity. (**A**) A multivariate regression model of average AIY activities (color coded by the training condition) based on activities of either AWA alone (broken blue line) or the combination of 5 neuron types (AWA, AWC, ASE, ASG and AWB, black line). Dynamics shown are during 15 s after the diacetyl-to-butanone switch. Pink arrows indicate where the addition of more neurons to the model improved the fit to the AIY activity. (**B**) Activity scatter plots of the different sensory neurons vs AIY across all training paradigms. (**C**) Regression coefficients of the sensory neurons. $R^2$ denotes the coefficient of determination. Asterisks denote the significance of regression coefficients in contributing to the linear combination. Note that the correlation coefficients for AWA strongly vary between conditions: green (pink) arrows indicate large (small) coefficients and hence strong (weak) effects on AIY activity. Error bars indicate confidence intervals. *p<0.05, **p<0.01, ***p<0.001 (t-test p-values for coefficients from regression statistics). (**D**) Gradual addition of sensory neurons to the linear combination model increased the variance explained in AIY Activity as reflected by the higher $R^2$ scores (line plot in top panel). In Naive and in STAV, AWA alone explained very little of the AIY activity variance (dotted red frame) and adding more neurons increased the $R^2$ scores. Appetitive conditions show a shallower increase in $R^2$ since AWA alone explains more than half of the activity variance observed in AIY (dotted cyan frame). Note that the overall adjusted $R^2$ does not deviate from $R^2$, indicating that overall the model excluded insignificant regressors. (**E**) Summary of the sensory-to-AIY communication routes (chemical and electrical *Choi et al., 2020*) that are modulated by experience. Colors indicate in which memory type they are modulated, and arrow's thickness indicates relative synaptic strength (*White et al., 1986*; *Witvliet et al., 2021*).

The online version of this article includes the following figure supplement(s) for figure 6:

**Figure supplement 1.** Synaptic communication routes between sensory neurons and the AIY interneurons change in an experience-dependent manner.

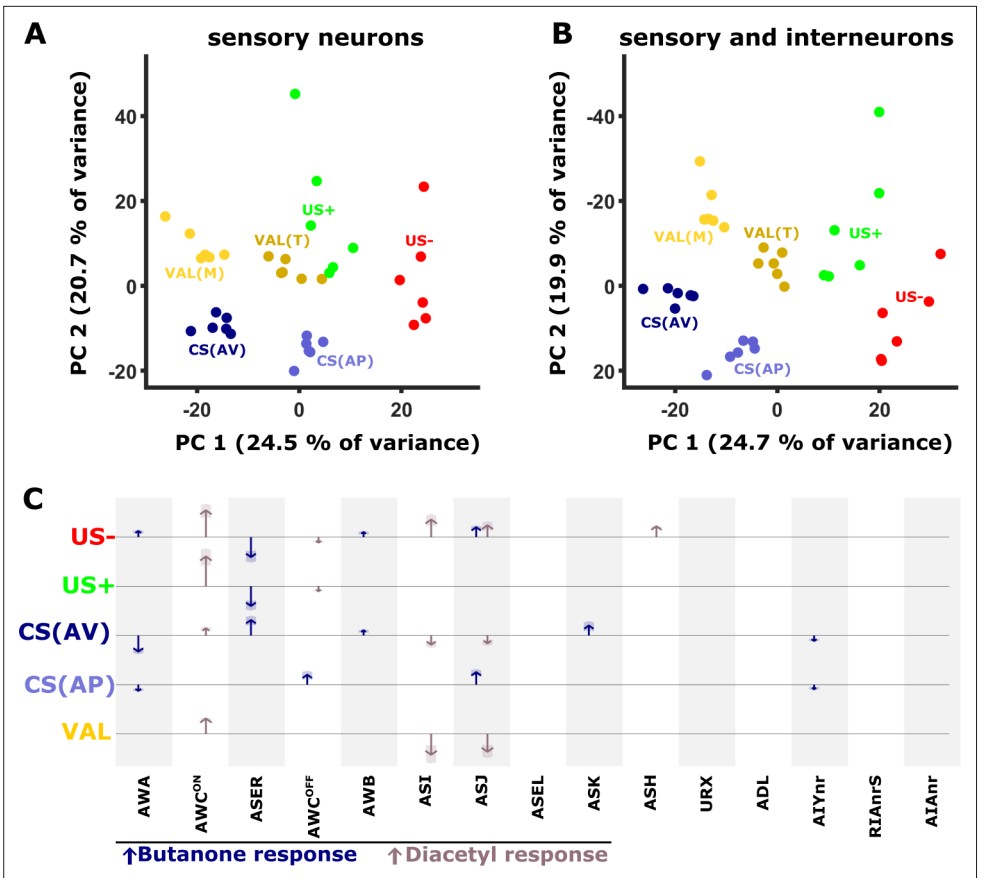

**Figure 7.** PC analysis reveals unique population codes for each of the training paradigms. (**A–B**) PCA scatter plot of the different experience components (*Figure 1—figure supplement 3*) as calculated based on the differences in neural activities: US+, appetitive unconditioned stimulus; US- aversive unconditioned stimulus; CS(AP) conditioned stimulus calculated from appetitive regime; CS(AV) conditioned stimulus calculated from the aversive regime. VAL(M) valence calculated from the mock-trained groups; VAL(T) valence calculated from the trained groups. (**A**) PCA when considering sensory neurons only. (**B**) PCA when combining sensory and interneurons. Note the better separation of clusters when interneurons are included in the analysis. (**C**) Map of activity changes associated with the various experience components. Blue arrows represent changes following butanone (BUT) exposure. Brown arrows reflect changes following exposure to diacetyl. Note some neurons (i.e. AWC) are OFF-type responders that react to stimulus withdrawal. Consequently, while the response change was recorded during DA presentation, the neuron is responding to BUT withdrawal. Shaded areas in the arrowhead indicate the standard deviation between trials.

The online version of this article includes the following figure supplement(s) for figure 7:

**Figure supplement 1.** PC analysis reveals the encoding neurons in each of the training paradigms.

for each of the memory components that jointly code the training paradigms. For this, we considered the changes in the activity dynamics of individual neurons across the different paradigms by calculating the difference in the neural activity between the training groups. For instance, subtracting the mean activity of aversively mock-trained animals from the mean activity of aversively trained animals, provides a measure for how much the CS changed the activity of that particular neuron (*Figure 1—figure supplement 3* and *Figure 7—figure supplement 1A*). We calculated these activity differences for each of the six trials across all neurons to generate an activity delta matrix (see Materials and methods and *Figure 7—figure supplement 1B*).

Next, we performed a principal component analysis (PCA) on this difference matrix. Interestingly, the first two components of the PCA already generated distinct clusters, each cluster representing an experience component (*Figure 7A*). For example, both CS components (bluish colors, denoted by the difference between trained and the corresponding mock-trained animals, *Figure 1—figure supplement 3*) are clearly distinct from the US components (green/red colors, denoted by the

activity difference between naive and mock-trained animals). Furthermore, even within the experience components, each condition is distinctly clustered. For example, the valence components, calculated by the differences within the corresponding trained and mock-trained groups (e.g. STAPT - STAVT and STAPM - STAVM), are grouped nearby, yet form two distinct clusters (yellow/brown colors, *Figure 7A*). When incorporating the data of the interneurons (AIY, AIA, and RIA), the separation between the clusters becomes even more prominent (*Figure 7B*). These findings suggest that individual experiences distinctly modulate the combined activity of the sensory and the interneurons.

The applied training regime is not the sole source for variation in neural activities. For example, differing network states (*Gordus et al., 2015*) and additional sensory inputs may significantly contribute to the overall modulated activity. To eliminate such plausible factors and to focus on the training-associated changes only, we reconstructed the original data based on the principal components 1, 2, 3 and 5, which best represented the experience components and their associated loads (see Materials and methods, *Figure 7B* and *Figure 7—figure supplement 1D*). We then plotted the reconstructed changes in the neural activities as arrows, where the length of the arrow represents the mean magnitude of the change associated with each of the experience components (*Figure 7C*). Notably, since this PCA-based method filters much of the variation in the data that is not associated with the experience components of the training paradigms, it provides a conservative, and presumably more accurate, representation for the role of neurons in each of the training paradigms.

It is evident that coding each of the memory components is distributed among several sensory neurons, where a few of them (e.g., AWC$^{ON}$) are broadly used to code most of these experience components. In addition, starvation itself, denoted by the US- experience component, involves activity modulation of multiple neurons. The changes in the activity of the interneurons showed marked variability, more than that observed in the sensory neurons (*Figure 5*). As a result, their activities were mostly filtered out following the PCA analysis and data reconstruction (see Methods and *Figure 7—figure supplement 1C–N*). Thus, the contribution of the interneurons to the overall neural representation is presumably underestimated. Together, these analyses show that experience components are distributed across various neurons that collectively form a unique population code for each of the training paradigms.

## Short-term training paradigms modulate the directionality of animals during chemotaxis towards the CS

The comprehensive analysis of neural activities showed that training modulates response dynamics of sensory- and inter- neurons (*Figures 2–7*), which in turn modulates animal preference towards the training stimulus (*Figure 1*). We therefore asked what are the fine training-induced locomotion changes that may underlie the modulated preference towards the CS (BUT). To understand the effects of training paradigms on specific locomotion parameters, animals were imaged when choosing between BUT and DA in conditions similar to the choice tests (*Figure 1*) and the neural imaging analyses (*Figures 3–6*). A multi-animal tracking system was used (*Itskovits et al., 2017*) to extract key locomotion parameters, namely, animal's speed, reversal frequency, and directionality (angle) towards the CS target (*Figure 8A–B*).

Positively trained animals were significantly more directed towards the CS target, while the negatively trained animals were the least directed towards the target (each paradigm compared to its naive and matched mock-trained animals, *Figure 8C–D*). Notably, animals that underwent aversive training with BUT showed a high deviation angle with low variance towards the alternative choice DA (*Figure 8C*, arrow), suggesting that the negative training increased aversion from BUT, and concomitantly, enhanced attraction to DA. Animal's speed and reversal frequency showed mild though significant changes (*Figure 8E–H*). To test the contribution of these changes to the overall behavior, we simulated animal chemotaxis based on experimental locomotion parameters (*Figure 8I–N*, and see *Figure 8—figure supplement 1* for explanation). These simulations indicated that animal directionality (the deviation angle from the target) accounted for most of the behavioral changes, while the contribution of the speed and the reversal frequency to the overall change in the locomotive behavior was negligible (*Figure 8—figure supplement 1*). These analyses suggest that training mainly affected animals' directionality features: more directed following positive training and less directed following an aversive training.

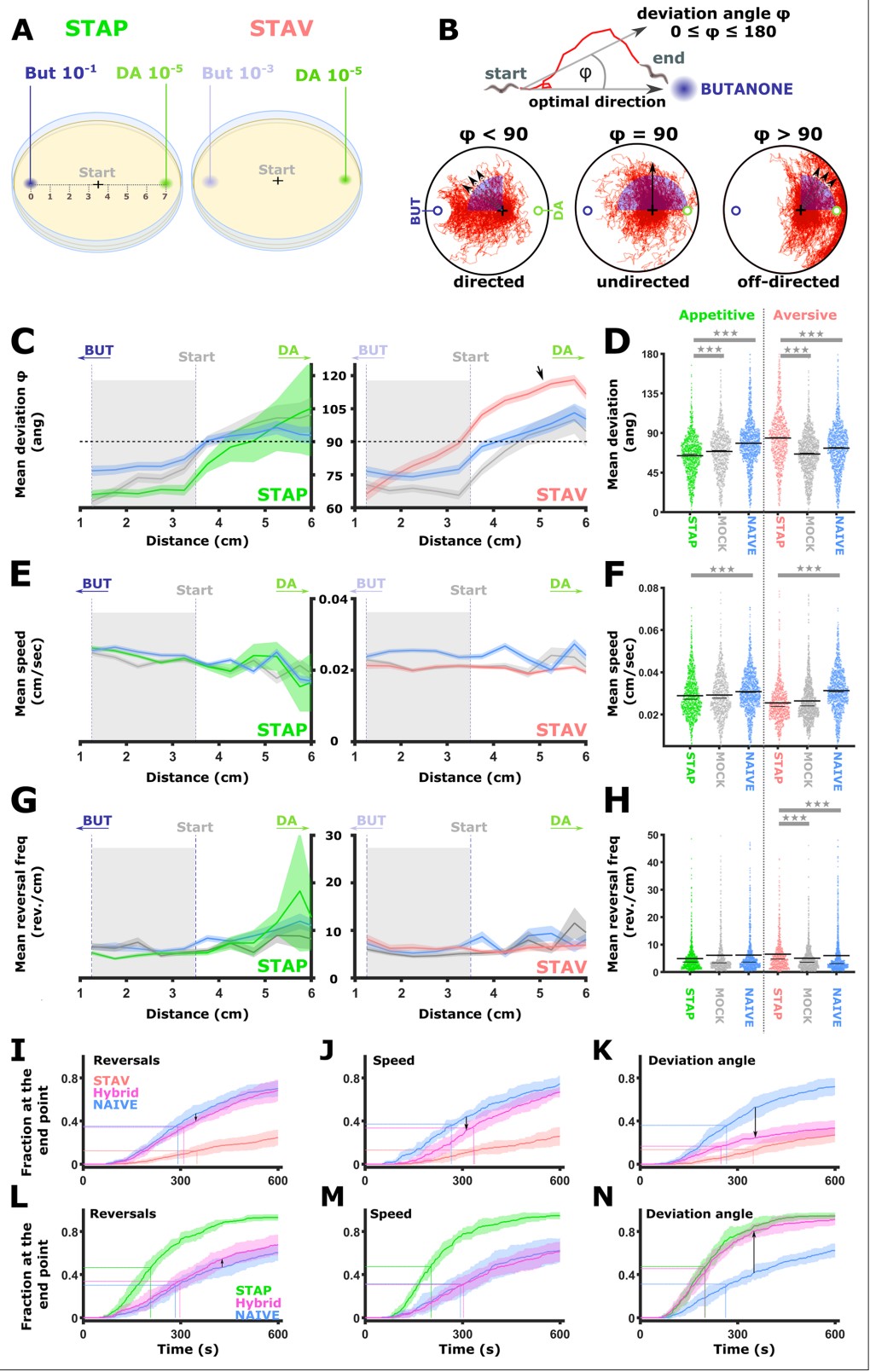

**Figure 8.** Short-term learning paradigms modulated the directionality towards the CS, but not the speed nor the reversal frequency. (**A**) A layout of the choice experiments for the appetitive and the aversive short-term training paradigms. Note the different concentrations of the CS butanone (BUT) used in each case. + sign marks the starting position of the worms at the beginning of the assay. Scale is in cm. (**B**) The directionality of the animal

*Figure 8 continued on next page*

*Figure 8 continued*

trajectory towards the BUT target is given by the deviation angle, the angle between the animal's directionality vector and the target. Deviation angles approaching zero mean a motion directed towards the target, and 180 degrees is directionality opposite from the target. (**C**) Plots of the deviation angle as a function of the distance from the target BUT (See panel A). STAP-trained animals make significantly smaller deviation angles than mock-trained and naive animals. In contrast, STAV-trained animals show significantly higher deviation angles than associated mock and naive controls. Note that the relative effect size is larger in aversive than appetitive animals and that animals migrate towards DA (arrow). The dotted horizontal line denotes 90 degrees. (**D**) Mean deviation angle in the proximal region (1.2-3.5 cm from the target, marked by gray area in C). (**E**) Plots of the speed as a function of the distance from the target BUT. (**F**) Mean speed angle in the proximal region (1.2-3.5 cm from the target, marked by gray area in E). (**G**) Plots of the reversal frequencies as a function of the distance from the target BUT. The units are given as reversals per centimeter worm track at the distance from the endpoint specified by the x-axis. (**H**) Mean reversal frequencies in the proximal region (1.2-3.5 cm from the target, marked by gray area in G). In C-H, shown are five independent experiments, each consisting of ~100 animals. Shaded areas around the plots indicate SEM. *p<0.05, **p<0.01, ***p<0.001 (rank-sum test, FDR corrected). (**I–N**) Simulations of choice behavior that test the contribution of each of the locomotion components to the behavioral output. Plots show the fraction of animals reaching the target over time. Each plot shows accumulation of simulated naive and trained animals by sampling locomotion parameters based on the measured data. Hybrid animals were simulated by sampling two of the parameters from the naive group and the relevant parameter from the trained group. Arrows indicate the magnitude of the effect between naive and hybrid simulated animals. See also *Figure 8—figure supplement 1*. (**I, L**) Sampling reversals from the STAV (**I**) or the STAP (**L**) trained group, while speed and directionality were sampled from the naive animals. Changes related to reversals are negligible (see arrows). (**J, M**) Sampling speed from the STAV (**J**) or the STAP (**M**) trained group, while reversals and directionality were sampled from the naive animals. Changes related to speed are negligible (see arrows). (**K, N**) Sampling deviation angle from the STAV (**K**) or the STAP (**N**) trained group, while speed and reversals were sampled from the naive animals. Changes related to directionality account for most of the difference between naive and trained groups (see arrows).

The online version of this article includes the following figure supplement(s) for figure 8:

**Figure supplement 1.** Simulations of choice behavior based on measured locomotion parameters.

## Discussion

Ample studies demonstrated associative (conditioning) learning in *C. elegans* (*Ardiel and Rankin, 2010*; *Cho et al., 2016*; *Kauffman et al., 2010*; *Loy et al., 2021*; *Oda et al., 2011*; *Sasakura and Mori, 2013*). In this study, we systematically mapped each of the four types of associative memories (short- and long-term memories, each encoded using positive and negative associations) onto the compact neural network of *C. elegans*. By using the same CS (BUT) in all training paradigms, we were able to extract the individual neurons that code either the CS, the US (positive or negative), or both.

Of note, short- and long-term memories can also be categorized based on the involvement of transcriptional and translational processes: While classical short-term memories do not depend on such processes, long-term memories do (*Asok et al., 2019*). Indeed, long-term memories were shown to depend on *crh-1* (a CREB homolog) and transcriptional changes (*Freytag et al., 2017*; *Kauffman et al., 2010*; *Lakhina et al., 2015*). However, short-term aversive conditioning with butanone was also shown to depend on RNAi and transcriptional changes (*Juang et al., 2013*). Thus, while the underlying molecular mechanisms may be paradigm or species specific (*Cho et al., 2016*), in this study, we classified the training paradigms into short and long-term memories based on the time period these memories were behaviorally persistent (e.g. modulated attraction to the conditioned stimulus, *Figure 1* and *Figure 8*).

### Sensory neurons' activities code short-term memories while interneurons code both short- and long-term memories

Our findings indicated that short-term, but not long-term, memories were mostly evident in changes of the sensory neurons' soma activities (*Figure 9A–B*). These short-term memories inflicted substantial changes in the stimulus response dynamics of multiple sensory neurons (*Figures 2–4* and *Figure 9C–D*). It is possible that the limited sensory neuroplasticity observed in long-term memories was due to the specific training conditions used herein, for example, the specific use of BUT as the CS and the training durations. Indeed, when using isoamyl alcohol as the CS, and coupling this odorant with a long-term aversive training, a mild modulated activity was observed in both the

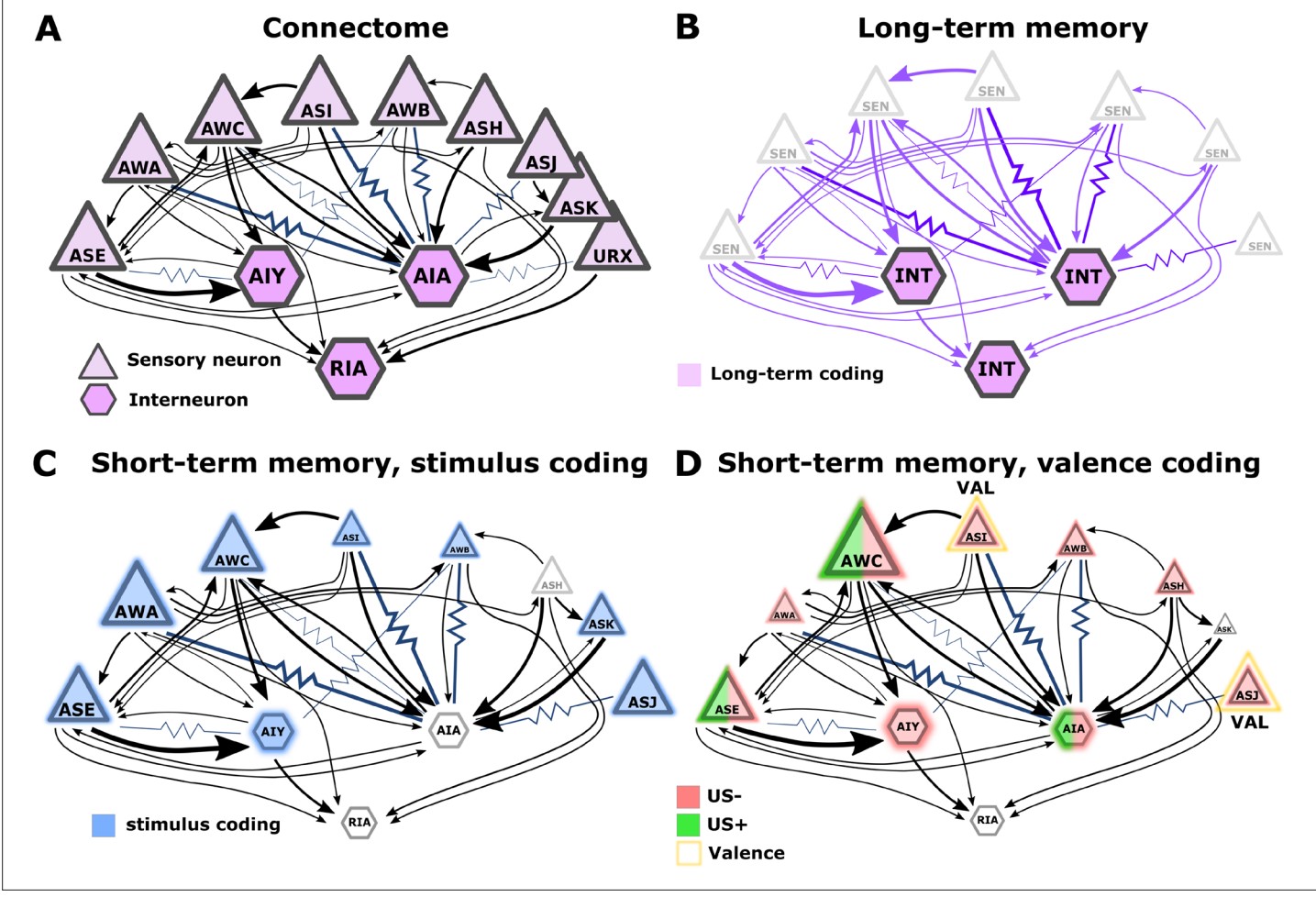

**Figure 9.** Illustration of the sensory- and the inter- neurons participating in coding the different memory types. (**A**) Multiple chemosensory neurons respond to butanone (BUT). All these chemosensory neurons innervate the AIA interneuron while only a few innervate the AIY interneurons. Arrows indicate chemical synapses and resistor symbols indicate gap junctions (*White et al., 1986*; *Witvliet et al., 2021*). (**B**) Long-term memory is evident in the interneurons and probably associated synapses (violet) rather than in the chemosensory neurons. (**C**) Highlighted in blue are the neurons participating in coding the stimulus component of short-term memories. (**D**) Highlighted in red/green are the neurons participating in coding the valence component of short-term memories. The fraction of the red/green color indicates the estimated impact of the memory on neural activity. Note that we denote AWC neurons as not coding valence because in our BUT-only experiments and in studies of others (*Cho et al., 2016*) AWC neurons showed no differences between aversive and appetitive conditions. In B-D, the size of the shapes (triangle or hexagon) indicates the estimated impact of the memory on neural activity.

AWC[OFF] and the ASH chemosensory neurons (*Eliezer et al., 2019*). Moreover, sensory neurons may still be coding long-term memories via modulation of the synaptic output and which is not reflected by soma calcium activity (*Oda et al., 2011*). Indeed, sleep is required for modulating the synatic structure between the AWC and the AIY neurons to sustain the long-term memory (*Chandra et al., 2022*).

The limited involvement of sensory neurons' activities in coding long-term associative memories may hint at an intriguing principle for coding memories in a compact neural network: Plasticity in the sensory neurons is likely to modulate sensory responses to various cues, possibly affecting innate behavioral outputs. For short-term memories, the modulated behavior will be brief, but for long-term memories, the impact on behavior will be long-lasting. Thus, for long-term memories, it may be advantageous to 'clear' information stored within sensory neurons and to relegate this information to the deeper layers of the network. This way, animals would quickly resume innate responses. The relegation of long-term memories to deeper layers can be viewed as analogous to the transfer process taking place in mammalian brains, where hippocampal short-term memories are moved for

long-term storage in cortical areas (*Rothschild, 2019*). In addition, resetting the sensory neuron's response dynamics may mitigate the limited sensory resources as more neurons may become available for coding new short-lived memories. Overall, the extensive sensory plasticity revealed herein is consistent with the idea that such plasticity may have evolved to increase animals' fitness by improving detection and enhancing attention towards salient stimuli (*McGann, 2015*).

Interneurons showed modulated activity following the formation of both short and long-term training paradigms, though activity changes following long-term experiences were more prominent (*Figure 3A, B*, and *Figure 9B–D*). Indeed, interneurons were previously shown to participate in coding both short- and long-term memories (*Jin et al., 2016*; *Oda et al., 2011*). Analogous roles of sensory and interneurons in coding short and long-term memories were observed in the salt conditioning paradigm. At least three chemosensory neuron types are required for the formation and retrieval of salt learning (*Jang et al., 2019*; *Watteyne et al., 2020*). While short aversive salt conditioning modulated responses of the salt-sensing neuron ASER (*Oda et al., 2011*), long-term positive salt conditioning did not (*Sato et al., 2021*). However, the activity of the downstream interneuron AIB was modulated following the positive long-term conditioning (*Sato et al., 2021*).

## Memory components are widely distributed across the sensory- and the inter-neurons

Activities of sensory neurons were broadly modulated by short-term experiences (*Figure 4A* and *Figure 7C*). The widely distributed changes in the chemosensory layer suggest that the sensory neurons encode the experience as a population code. Considering modulation of individual neurons only precludes accurate distinction between the training paradigms. But when considering all the chemosensory inputs, the underlying experience could be decoded with sufficient accuracy (*Figure 4K–M*), further highlighting the notion that the experience code is distributed among the sensory neurons. Within the distributed code of the experience components (*Figure 7A and B*), the PC-based analysis revealed how activity changes of individual neurons contributed to encoding the CS, the US, or both (*Figure 7C* and *Figure 9C–D*).

Assigning valence-coding neurons was somewhat limited due to the procedural differences in aversive and appetitive training paradigms. This was particularly evident when training for the long-term paradigms which involved several key procedural differences. Nevertheless, the procedural differences between positive and negative training in the short-term paradigms were minimal (see Materials and methods), so valence-coding neurons could be extracted with higher certainty. These analyses indicated that the ASJ and ASI neurons may be the strongest candidates for coding valence (*Figure 4A*, *Figure 7C*, *Figure 4—figure supplement 2A–C, M–R and V–W*). Indeed, the ASI neurons were shown to integrate information regarding food availability (*Gallagher et al., 2013*; *Hapiak et al., 2013*).

The observed activity changes were generally consistent with the experience logic (positive or negative). For instance, in aversively trained animals, switching from DA to BUT leads to ASER activation (*Figure 4E–G*), an activation that triggers a reversal behavior (*Appleby, 2012*; *Suzuki et al., 2008*) which prevents the animals from moving towards BUT. Changes in other neurons were harder to reconcile. For example, in both negatively and positively trained animals, activity in the AWA neurons was strongly reduced in response to BUT (*Figure 4B–D*). Thus, it is presumably the combined modulated activity of all sensory neurons that is being integrated into the downstream interneuron layer to express the adaptive behavioral responses.

The distribution of memory components among multiple neurons may evolve as an organizing principle in memory formation and storage. In flies, olfactory memories are encoded using a distributed code within the mushroom body (*Bilz et al., 2020*), and in mammalian brains, it is assumed that engram cells, the set of memory-storing neurons, are also widely distributed in the brain *Josselyn and Tonegawa, 2020*: the valence component in the amygdala (*Liu et al., 2012*), and the stimulus-specific information in sensory cortices (*Jones et al., 2008*; *Morris et al., 1998*; *Ohl and Scheich, 2005*; *Sacco and Sacchetti, 2010*).

## Changes in AWC<sup>ON</sup> responses are concentration-specific and depend on intact neural transmission

The AWC<sup>ON</sup> neuron shows training-induced plasticity that is strongly dependent on the specific concentrations of the CS (BUT). For example, when stimulating the trained animals with low concentrations of the CS, both the CS and the US were coded by the AWC<sup>ON</sup> neuron. Exposure to a ten-fold higher concentration of the CS suggested that the AWC<sup>ON</sup> neuron coded the CS only (compare *Figure 3D* and *Figure 4H*). The strong dependence of the AWC<sup>ON</sup> responses on the specific concentrations of the CS was also evident in a previous study where responses were observed only when stimulating the animals with the trained concentration, and regardless of the positive or the negative associating experience (*Cho et al., 2016*).

Furthermore, *Cho et al., 2016* demonstrated that in butanone conditioning, the AWC sensory neuron coded the sensory history of the animal, while the downstream AIA interneuron coded the associative component of the memory. This sensory history is analogous to our observation made with BUT only in which the AWC<sup>ON</sup> neuron codes strictly the CS, irrespective of the positive or the negative association.

The training-induced modulated activity of the AWC<sup>ON</sup> neuron appears to be non cell-autonomous, since this plasticity was not observed in mutant animals, defective in neural transmission (*Figure 2—figure supplement 3*). This suggests that a network activity is required for recruiting a population of neurons for storing the memory.

## The modulated activity of the AIY interneuron can be explained by the combined activity of the sensory neurons

Population coding requires integration of the coding sensory neurons into downstream neurons, such as the AIY interneuron. A regression analysis revealed that AIY activity can be explained via a simple linear combination of the sensory neurons (*Figure 6*). This analysis provided a quantitative measure for the relative contribution of the sensory neurons to the overall activity of the AIY neurons. These relative contributions uniquely changed in an experience-dependent manner, suggesting that the identity and the specific modulation of neural communication routes dictate each of the specific memory types. For example, the AWA neurons contributed the most to the modulated activity of AIY neurons in the appetitive paradigms, while AWC, ASER, ASG, and AWB neurons dominated in dictating AIY activity in naive animals and following aversive training (*Figure 6E*). In that respect, the AWA neurons are associated with attractive stimuli (*Bargmann et al., 1993*), while the sensory neurons required for aversive conditioning are mostly associated with reversals and avoidance (*Gray et al., 2005*; *Suzuki et al., 2008*; *Troemel et al., 1997*).

Notably, the aim of the linear combination model was to identify modulated synaptic communication routes. The computed regression coefficients, which denote a change in the communication routes, are not a direct measure of the underlying synaptic weights. This is due to the limitations in data acquisition (i.e. frame rate, kinetics of GCaMP, and number of available data points, *etc.*) and the fact that we measure calcium levels which are only correlates of neural potential. Furthermore, the modulated communication routes may include, in addition to the classical chemical synapses, also electrical gap junctions that were demonstrated to be imperative to form memories in *C. elegans* (*Choi et al., 2020*). Nonetheless, the model reliably revealed specific communication routes (chemical and electrical synapses combined) that changed in an experience-dependent manner.

## Short-term memories modulate the animals' directionality towards the conditioned stimulus

Behavioral assays following short-term training revealed that positively trained animals were more directed towards the CS, while negatively trained animals were significantly less directed (*Figure 8C–D*). Interestingly, aversive training also inflicted significant changes in sensory-specific responses of the RIA neurons (*Figure 5G and H*). As the RIA neurons dictate animals' head position (*Hendricks et al., 2012*; *Hendricks and Zhang, 2013*; *Ouellette et al., 2018*), their modulated dynamics may explain the changes observed in movement directionality. Indeed, RIA neurons were shown to participate in formation and forgetting of aversive experiences, suggesting that past experiences may be converged onto RIA interneurons to shape behavioral outputs (*Jin et al., 2016*; *Liu et al., 2022*). As the animals' reversal frequency and speed were not modulated (and no changes in AVA and AVE activity patterns

were observed either), RIA is a possible candidate that could underlie the observed change in directionality. However, it is impossible to tell whether animals changed directionality due to RIA-induced head swing bias, or alternatively, due to reorientation mediated by command neurons (e.g. AVA, AVE) since we tracked animals as center mass points. Also, activity changes in RIA neurons were significant though small, thus raising the question whether these changes are large enough to mediate the observed trained phenotype. Moreover, additional interneurons that control turning rates (e.g. AIB, AIZ, and RIM) and speed (e.g. RIB, SIA, and RMG) may also play roles in these memory-induced behavioral modulations (*Garrity et al., 2010*; *Iino and Yoshida, 2009*; *Lee et al., 2019*; *Li et al., 2014*; *Wakabayashi et al., 2004*). In fact, some of these neurons had been shown to participate in either memory formation (AIB & RIM) or memory retrieval (AIY & RIA) (*Jin et al., 2016*). Thus, additional experiments are required to support the functional role of RIA plasticity in shaping the learning-dependent behavioral outputs.

As memories are encoded in a distributed manner, it is presumably the integrative activity of many neurons that gives rise to the altered choice of directionality. As such, the impact of a single neurons' activity on locomotion might be limited. The need to consider the integrated response of all neurons was also evident from analyses of freely behaving animals, where a population code, rather than the activity of individual neurons, was shown to be a better descriptor of locomotion (*Hallinen et al., 2021*).

In summary, the systematic cellular-resolution analysis presented herein revealed basic principles for how associative memories are encoded in a compact neural network. These principles may extend to memory formation processes taking place in higher organisms with more complex nervous systems.

**Table 1.** Worm strains used in this study.

| Designation | Phenotype/Purpose | Genotype/Expression (Source/Ref) |
|---|---|---|
| N2 | WT | N2 wild-type (CGC) |
| P*gcy*-37::YX2.60 | URX reporter line | [P*gcy*-37::YX2.60] *Gross et al., 2014* |
| PS6250 | CEPD reporter line | Ex[P*dat*-1::GCaMP3] *Zaslaver et al., 2015* |
| PS6374 | AWC[ON] reporter line | Ex[P*str*-2::GCaMP3] *Zaslaver et al., 2015* |
| PS6253 | AWC[OFF] reporter line | Ex[P*srsx*-3::GCaMP3] *Zaslaver et al., 2015* |
| PS6498 | AWC[OFF] and AWC[ON] reporter line | Ex[P*str*-2::ChR2-cherry,P*str*-2::GCaMP3,*srsx*-3::GCaMP3; pha-1 rescue; lite-1 bkg] |
| ZAS96 | AWC[ON] reporter line in *unc-13* bkg | Ex[P*str*-2::GCaMP3] in *unc-13(e51)* - This study |
| ZAS76 | AWC[ON] reporter line in *unc-31* bkg | Ex[P*str*-2::GCaMP3] in *unc-31(e928)* - This study |
| ZAS280 | Sensory neurons reporter line | *azrIs347*[P*osm-6*::GCaMP3,P*osm-6*::NLS-mCherry-2xNLS + PHA-1] *Iwanir et al., 2019* |
| ZAS323 | Sensory and command neuron reporter line | *azrIs347*[P*osm-6*::GCaMP3,P*osm-6*::NLS-mCherry-2xNLS + PHA-1] x *goeIs5*[P*nmr-1*::SL1::GCaMP3.35::SL2::unc-54 3'UTR +unc-119(+)], crossing of ZAS280 x HBR191, *Schwarz and Bringmann, 2013* |
| PS6510 | RIA reporter line | Ex[P*glr-3*::GCaMP; *pha-1* rescue] - This study |
| ZAS256 | AIY reporter line | Ex[P*gpa-6*::GCaMP3, P*mod-1*::GCaMP3; P*pha-1*::PHA-1]; *pha-1*; *lite-1* *Itskovits et al., 2018* |
| CX16561 | AIA reporter line | [P*gcy28d*::GCaMP D381Y coel::dsRed, P*odr-7*::Chrimson::SL2::mCherry,P*elt-2*::mCherry 2] *Larsch et al., 2015* |
| HBR191 | command neurons reporter | *Int*[*nmr-1*p::SL1::GCaMP3.35::SL2::*unc-54* 3'UTR +UNC-119(+)] *Schwarz and Bringmann, 2013* |

## Materials and methods

### Worm cultures

Animals were grown at 20 °C on 9 cm nematode growth medium plates, seeded with 500 μL of confluent OP 50 bacterial suspension. For culturing and experiments, eggs were collected by dissolving the animals using standard bleaching protocols. The eggs were seeded at a density of 1000–1200 per plate. For training in short-term paradigms, bleaching and seeding were conducted 3 days before the experiment. Animals undergoing long-term training were seeded 48 hr before initiation of the training.

### Worm strains

For functional imaging, worm strains driving the expression of calcium reporters in neurons of interest were used (*Table 1*). N2 wild-type worms were used for behavioral assays.

### Training procedures

To induce olfactory associative memories, BUT was presented to the animals in combination with food (appetitive, positive conditioning) or in the absence of food (starvation, aversive conditioning). Associated mock-trained control groups underwent the same treatment without BUT presentation. Naive control animals were of the same age but not treated at all.

### Short-term appetitive training

Animals were washed three times in M9 and then starved for one hour in 1 mL of M9 in a 15 mL centrifuge tube with an open lid. Worms were then trained on high-food NGM plates (seeded with 500 μL of confluent OP 50 culture) in the presence of 20x5 μL droplets of 10% (v/v in DDW) BUT applied to the inside face of the plate lid. The mock-trained group received 20x5 μL droplets of DDW. Training duration was 1 h (modified after *Kauffman et al., 2010*).

### Short-term aversive training

As previously described (*Bargmann et al., 1993*; *Colbert and Bargmann, 1995*), worms were washed three times in an M9 buffer and transferred onto chemotaxis plates (1.7% (w/v) Agar, 25 mM $KH_2PO_4$, 1 mM $CaCl_2$, 1 mM $MgSO_4$, pH 6.0, no food). The trained group was incubated for 90 min with 20x5 μL droplets of 10% (v/v in DDW) BUT, while the mock-trained group was incubated with an equivalent amount of DDW. Note that short-term paradigms differed by overall incubation time, plates used for incubation and one transfer step.

### Long-term appetitive training

This training consisted of seven cycles in which BUT was paired with food, as described in *Kauffman et al., 2010*. Each repetition consisted of a 30-min starvation in M9 buffer and a 30-min food-BUT pairings, except for the first cycle, in which starvation lasted 1 hr. In contrast to all other training regimes, 5x2 μL droplets of 10% (v/v) were used in long-term appetitive training since initial calibration assays showed that higher levels of BUT led to an aversive choice behavior. Animals were imaged 14 hr post-training.

### Long-term aversive training

Worms were washed three times in an M9 buffer and transferred to chemotaxis plates. The trained group was starved with 20x5 μL droplets of 10% (v/v) BUT on the lid for 10 hr, with one exchange of the BUT droplets after 5 hr. The mock-trained groups were starved in the presence of DDW droplets. Animals were imaged 14 hr post-training.

### Behavioral assays

Worms were washed three times with a chemotaxis buffer, and 100–200 animals were transferred onto the center point of a chemotaxis plate, 3.5 cm from the target endpoints. Four-pole and 2-pole layouts were used as depicted in *Figure 1—figure supplement 1A*. Endpoints were loaded with BUT (BUT dissolved either in water or EtOH) or the alternative choice (DA in water or pure ethanol, see *Figure 1—figure supplement 1A*). Note that different concentrations of BUT were used for

animals with appetitive and aversive training because of the valence-specific shift in choice behavior (*Figure 1—figure supplement 2*). Positively trained animals were tested with 10-fold diluted BUT ($10^{-1}$) and negatively trained animals were tested with 1000-fold diluted BUT ($10^{-3}$). Worms were immobilized once reaching the endpoints by applying 1 µL of 1 M $NaN_3$ to those endpoint regions. Animals in each region were subsequently scored to provide the choice index. Learning index was calculated based on these choice indices as a measure of learning-induced behavioral changes (*Kauffman et al., 2011*).

To obtain locomotion parameters during chemotaxis, worms were imaged using a Micropublisher 5 RTV CCD camera (QImaging, Canada) equipped with a ZOOM 7000 Navitar macro objective (Navitar, New York, USA). Animal tracks were extracted using a multi-worm tracker (*Itskovits et al., 2017*), from which we quantified deviation angles, speed, and reversal frequencies. To control for BUT evaporation and to ensure behavioral consistency, only the first 10 min of the movies were analyzed. To provide higher accuracy of local deviation angles and speed, tracks were segmented into 24-frames segments. The deviation angle is defined as the angle between the vector pointing from the animal towards the endpoint and the average vector of the worm track segment (*Figure 9B*). Due to the use of the center-mass-tracking, reversals were defined as any perceptible form of backward movement (*Gray et al., 2005*).

## Calcium imaging and data analyses

In preparation for live imaging, animals were starved for 20 min on empty NGM plates. For imaging multiple neurons, worms were also paralyzed using 10 mM levamisole dissolved in chemotaxis medium. The worms were then loaded into the microfluidic 'olfactory chip' (*Chronis et al., 2007*) and allowed to habituate for 10 min.

Neural responses to BUT (*Figures 2 and 3*) were recorded for 90 s: 30 s after the initiation of imaging, animals were presented with 33.5 mM BUT (diluted in chemotaxis medium, $3 \times 10^{-3}$ dilution factor), and then imaged for an additional 60 s. After the animals were acclimated to the presence of BUT for 5 min, we re-initiated imaging, and after 30 s of imaging we switched the stimulus (BUT) off, and continued imaging for an additional 60 s. For imaging neurites of single interneurons, worms were not paralyzed. The stimulus exchange interval was 20 s and responses were recorded for 3 min without interruption (*Figure 2B*). Each BUT presentation/removal is referred to as a 'trial'.

To record responses to alternating BUT (3.35 mM, $3 \times 10^{-4}$ dilution factor) and DA (11.6 µM, $10^{-6}$ dilution factor) in *Figures 4 and 5*, animals were exposed to one minute of DA followed by six exchanges of BUT/DA, each step lasting 30 s. The exchange of BUT and DA was meant to mimic the conditions that the animals encountered during the two-choice assay. Note that in *Figures 4–7*, the BUT/DA exchanges are referred to as trials.

A Nikon A1*R*+confocal laser scanning microscope (Nikon, Japan) equipped with a 40x1.15 NA water immersion objective was used for fast live imaging. Z-series of the head region of the animal were recorded at 0.9–2 volumes per second. Individual z-stacks were scanned at 0.4–0.8 µm intervals (sampling rate 2–5 Hz) in the sensory-reporter lines (ZAS280 and ZAS323). Single sensory neurons were imaged with an IX 83 epifluorescence microscope (Olympus, Japan) and a 40x0.95 NA objective. The image acquisition was controlled by µManager (*Edelstein et al., 2010*).

To identify individual neurons in multi-neuron z-stack time series, neuronal somas were segmented using a Gaussian fitting and a tracking algorithm (*Toyoshima et al., 2016*) targeting nuclear mCherry tags. GCaMP intensities in target neurons were extracted from segmented neurons by a custom-built analysis pipeline (*Pritz, 2022*, GitHub) in Matlab (Mathworks, USA) reading voxels within a 70% radius of the initial segmentation radius. Image stacks from neurites were projected by summing all images using imageJ. Projected micrographs were analyzed using custom imageJ and Matlab scripts utilizing Fiji's trackMate plugin (*Tinevez et al., 2017*). Since data were acquired with varying frame rates, neuronal activation plots were linearly interpolated to the highest frame rate in the dataset (2 Hz sampling rate for neurons read at the soma and 3 Hz (*Figure 5*) and 5 Hz (*Figure 2B*) for the neurite datasets).

Neural activation levels in sensory neurons were normalized by their ground state ($F/F_G$), unless stated otherwise. In short recordings during BUT exposure (*Figures 2 and 3*), the ground state was extracted from the last 10 frames of the imaging after the neuron resumed its pre-exposure ground state. For interneurons, a 10-frame ground state was visually identified due to possible spontaneous

activity (*Figure 2B*). For longer recordings in BUT/DA exchange experiments (*Figures 4 and 5*), the ground state was determined as the mean of intensity values lower than the 10% percent-quantile after smoothing activation vector by a 20-frame kernel size. Automatic and visual determination of ground states were found to be in good agreement.

## Identification of individual neurons

Individual neurons within the pan-sensory reporter strain (P*osm-6*::GCaMP, ZAS280) were unambiguously identified using custom-built Matlab 3D visualization tools based on available anatomic maps (*Durbin, 1987*; *White et al., 1986*). Identification of AWC$^{ON}$, AWC$^{OFF}$, and URX neurons was verified by comparing to activity profiles of these neurons as imaged from reporter strains with known cell identities (see *Figure 2—figure supplements 3 and 4*). Signals from the AFD neurons were too dim to provide reliable measurements, and hence, were discarded from all analyses.

## Statistical analysis

Hypothesis testing of neuronal activation and behavior was carried out using MatLab and Python. For statistical comparisons, intensities following stimulus exchange were summed. Only neuronal intensity measurements within a 95% confidence interval were used for statistics. Integration times were neuron-specific since neuronal dynamics strongly varied among neurons. Data were tested for normal distribution by Shapiro-Wilk test (small sample number) or Kolmogorov-Smirnov test (larger sample number). ANOVA or ANOVA on ranks followed by pairwise comparisons based on t-tests or Wilcoxon rank-sum test/ signed rank test, depending on the underlying distribution, were used to test for differences. Multiple comparisons were adjusted using false discovery rates (pFDR *Storey, 2002*). For comparing neuronal activation between the different learning paradigms within the BUT-exposure dataset, neurons with reliable activity responses were included in the analysis (namely, AWA, ASH, AWC$^{ON}$, AWC$^{OFF}$, AWB, ASJ, ASI, ASK, RIAnrD, RIAnrV, RIAnrS, and AIA). Other neurons (ASE, ADL, ASG, and ADF) were excluded because of insufficient reads or cross-read artifacts. For each of these neurons, twelve pairwise comparisons of activity post-stimulus exchange were conducted as stated in *Table 2*, yielding 324 comparisons in total.

For comparing neuronal activation within the BUT/DA exchange dataset, 16 neuron classes were included (AWA, AWC$^{ON}$, AWC$^{OFF}$, AWB, ASER, ASEL, ASJ, ASK, URX, ASH, ASI, AIY, AIA, RIAnrD/nrV/ nrS, AVA, and AVE) applying the first eight comparisons in *Table 2* yielding 248 comparisons. Significant experience-dependent changes in choice behavior were detected by one-sample t-tests (paired test) against zero.

## Analyzing variability in sensory neuron responses

Neural activities in sensory neurons were categorized into 'responding' and 'non-responding' groups by applying a neuron-specific activity threshold. For the trial-to-trial variability, the neural activity had to cross the threshold once to be categorized as 'responding'. For animal-to-animal variability, the neural activity had to cross the threshold twice during the six consecutive trials (BUT/DA exchanges).

**Table 2.** pairwise comparisons of the different training conditions.

| Comparisons design | |
| --- | --- |
| Group 1 | Group 2 |
| STAP-T | STAV-T |
| STAPM | STAVM |
| STAP-T | STAP-M |
| STAV-T | STAV-M |
| STAPM | NAIVE |
| STAVM | NAIVE |
| STAP-T | NAIVE |
| STAV-T | NAIVE |
| | |
| STAPT | LTAP-T |
| STAVT | LTAV-T |
| LTAP-T | LTAV-T |
| LTAPM | LTAVM |
| LTAP-T | LTAP-M |
| LTAV-T | LTAV-M |
| LTAP-T | NAIVE |
| LTAV-T | NAIVE |
| LTAP-M | NAIVE |
| LTAV-M | NAIVE |

STAP = Short-term appetitive. STAV = Short-term aversive. LTAP = Long-term appetitive, LTAV, Long-term aversive. T = Trained. M = Mock.

## Classification of memory conditions

Single or concatenated neural activities were binned into a 20-frame kernel and subsequently z-normalized. To perform classification based on trials, activities of all neurons for each animal and trial were concatenated and 60% of the data was used to train the algorithms, while the remaining 40% of the data was used to test the classification algorithms. To perform classification based on splitting trials within individuals, trials of each neuron were random-sampled into test and training, and then averaged to obtain one vector per animal as shown in *Figure 4—figure supplement 7A*. When cross-validating the results after splitting the data into training and test data based on animal repeats, we obtained similar results, although with higher classification error. Thus, classification accuracy could benefit from adding more animals per training group (see 'between individuals', *Figure 4—figure supplement 7F*). Classifications were performed using the scikit-learn package in Python using, if not stated otherwise, k-means nearest-neighbor based on distance to the next two data points. Random forest classification consisted of 500 trees and a neural net (MLPC) consisted of 2 layers, with 50 and 100 neurons each. The average macro F1 score out of 10 rounds of cross-validation has been used to measure the performance of the classification procedures on the tested data.

## Regression analysis

AWA activities in a 15 s interval following stimulus exchange were regressed against AIY activity in the corresponding time interval using ordinary least squares (OLS) method to explain AIY activity by AWA activity within the same animal. As Generalized Least Squares with autoregressive models produced a less accurate fit, we used a simple OLS model to fit regression coefficients. To explain AIY activities by activities of sensory neurons, we used a multivariate regression model based on normalized activities within the 15 s following stimulus exchange. For the multivariate model, all neuronal activities were averaged across animals for each trial to make activities originating from different reporter strains comparable (*Figure 6—figure supplement 1I*). The resulting six averaged vectors for each neuron were used as inputs for the multivariate OLS models. Models were cross-validated for overfitting using a 50/50 data split and the variation of regression coefficient values across the different conditions was verified (see *Figure 6—figure supplement 1F and G*). Multivariate regression analysis could not be extended to AIA and RIA neurons because there were too many input neurons (regressors) opposed to too few data points.

## PCA-based filtering of activity deltas

Neural activities were averaged across individuals for each trial. Activity deltas of averaged activities (see *Figure 1—figure supplement 3*) were calculated for the CS, US, and valence differences. All activity deltas for each comparison and trial were aligned in a single vector and subjected to PCA. Using the PC scores and loads from principal components 1, 2, 3, and 5 (out of 35 principal components), we reconstructed the original activity deltas while filtering out 39.1% of the unrelated variance (*Figure 7C*, see *Figure 7—figure supplement 1A–C*). The filtered activity deltas were then summed and normalized by the mean amplitude of the neuron (contributions of neurons with higher response amplitudes are weighted more). The mean of the six trials for each neuron was used to indicate increase or decrease in the activity, and the standard deviation to provide an estimate of the variance (*Figure 7C*). To validate the method, we simulated the data and subjected it to PCA-based filtering (*Figure 7—figure supplement 1E–M*), allowing us to differentiate between significant and insignificant changes using k-means-based thresholding of the activity changes (*Figure 7—figure supplement 1N*).

## Acknowledgements

We thank the reviewers and the editors for the very helpful suggestions which eventually greatly improved the final version of this manuscript. We also thank Paul Sternberg in which lab we generated several of the strains used herein, and Cori Bargmann and Einav Gross for sharing strains. Some strains were provided by the CGC, which is funded by the NIH Office of Research Infrastructure Programs (P40 OD010440). Research in the AZ lab was supported by ERC (336803), ICORE (1902/12), and ISF (1300/17). COP postdoctoral fellowship was also supported by the David-Herzog-Funds at Styrian

Universities. EB, RR, and EI were supported by the Jerusalem Brain Center. AZ is the Greenfield chair in Neurobiology.

## Additional information

### Funding

| Funder | Grant reference number | Author |
|---|---|---|
| European Research Council | 336803 | Alon Zaslaver |
| Israel Science Foundation | 1300/17 | Alon Zaslaver |
| Israeli Centers for Research Excellence | 1902/12 | Alon Zaslaver |
| David-Herzog-Funds at Styrian Universities | | Christian Pritz |

The funders had no role in study design, data collection and interpretation, or the decision to submit the work for publication.

### Author contributions

Christian Pritz, Conceptualization, Resources, Data curation, Software, Formal analysis, Validation, Investigation, Visualization, Methodology, Writing – original draft, Writing – review and editing; Eyal Itskovits, Resources, Data curation, Software, Formal analysis, Validation, Investigation, Visualization, Methodology; Eduard Bokman, Resources, Data curation, Formal analysis, Investigation, Methodology, Writing – review and editing; Rotem Ruach, Resources, Data curation, Software, Formal analysis, Validation, Investigation, Methodology; Vladimir Gritsenko, Validation, Investigation, Methodology; Tal Nelken, Data curation, Formal analysis, Investigation; Mai Menasherof, Resources, Investigation, Methodology; Aharon Azulay, Conceptualization; Alon Zaslaver, Conceptualization, Resources, Data curation, Supervision, Funding acquisition, Investigation, Visualization, Methodology, Writing – original draft, Project administration, Writing – review and editing

### Author ORCIDs
Christian Pritz ⓘ http://orcid.org/0000-0003-4264-9468
Vladimir Gritsenko ⓘ http://orcid.org/0000-0003-2676-8256
Alon Zaslaver ⓘ http://orcid.org/0000-0002-7871-4420

### Decision letter and Author response
Decision letter https://doi.org/10.7554/eLife.74434.sa1
Author response https://doi.org/10.7554/eLife.74434.sa2

## Additional files

### Supplementary files
• Transparent reporting form

### Data availability
All neuronal activities and behavioral data sets together with the associated analyses scripts are available in https://osf.io/5v4qu/ and https://github.com/ChristianPritz/Principles-for-coding-associative-memories-in-a-compact-neural-network (copy archived at *Pritz, 2022*) respectively.

The following dataset was generated:

| Author(s) | Year | Dataset title | Dataset URL | Database and Identifier |
|---|---|---|---|---|
| Pritz CO, Itskovits E, Bokman E, Ruach R, Gritsenko V, Nelken T, Menasherof M, Azulay A, Zaslaver A | 2020 | Principles for coding associative memories in a compact neural network | https://osf.io/5v4qu/ | Open Science Framework, 5v4qu |

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
