## [Editor Report]

In this study, the authors established paradigms for appetitive and aversive short-term and long-term olfactory learning. They then produced a large collection of activity recordings in a handful of sensory neurons and interneurons, produced a linear model to describe sensory-evoked interneuron activities, and observed changes in the activities caused by learning. Although more work is needed to explain how these activity patterns relate to behavior, the collection of data provides hypotheses for future studies on the function of the neurons implicated in the learning paradigms and provides useful references for similar studies in the field.

---

## [Decision Letter]

**Decision letter after peer review:**

[Editors’ note: the authors submitted for reconsideration following the decision after peer review. What follows is the decision letter after the first round of review.]

Thank you for submitting your work entitled "Principles for coding associative memories in a compact neural network" for consideration by *eLife*. Your article has been reviewed by 2 peer reviewers, and the evaluation has been overseen by a Reviewing Editor and a Senior Editor. The reviewers have opted to remain anonymous.

Our decision has been reached after consultation between the reviewers. Based on these discussions and the individual reviews below, we regret to inform you that your work will not be considered further for publication in *eLife*.

We appreciate the newly developed paradigm of associative learning which can assess both appetitive and aversive learning as well as both short and long-term memory. Authors further went on to perform calcium imaging of multiple neurons simultaneously, and determined the odor response of these neurons before and after each of the four combinations of conditioning. This is a tremendous work, which is truly admirable. However, unfortunately, many of the results are not readily interpretable and there are multitude of apparent inconsistency in view of the authors' conclusions as described in the reviewers' comments. Although we cannot continue the review process further at this time so that authors can submit their results to another journal, if authors decide in the future that above concerns are eliminated with additional experiments, we will be happy to receive a new submission.

*Reviewer #1 (General assessment and major comments (Required)):*

In this study Pritz et al., expand the behavioral paradigm for of short- and long-term memory of appetitive and aversive odorant responses. This assay alone is a nice contribution to the field of learning and memory in *C. elegans*, as these four degrees of freedom can be specifically switched, while leaving other conditions constant. The authors then underwent a calcium imaging tour de force applying multi-neuron imaging to search for underlying memory traces in neuronal activity. They examine how the responses of neurons are modulated by the memory conditions, at different stages of sensory-motor flow, i.e. including the primary sensory neurons, interneurons and motor-command neurons. They find that the short-term memory conditions mainly modulate the response of the sensory neurons, while long term memory modulates the response of interneurons. A basic claim of this paper, therefore is that short- and long-term memory segregate at the circuit level, which is a very interesting result. Finally, the authors analyze how behavioral parameters such as turning, and reversal rates are modulated by the different memory paradigms.

Comments:

1. Unfortunately, while the authors indeed find memory traces in neuronal activity of some neurons, these results do not fall into place to allow for a clear comprehensive model how these changes in neuronal activity relate to the underlying changes in behavior.

I) AWC-ON neurons seem sensitized to odor-off stimulus both upon STAP and STAV, this is interesting but cannot explain any of the changes in behavior.

II) AWA activity to odor-on is not significantly different to the respective MOCK conditions, so I don't think one should conclude anything from the small effect sizes reported here only for STAP vs STAV and STAV vs naïve. I am not convinced of the authors' statement AWA would encode valence.

III) ASH is sensitized to odor-off specifically upon STAV. This is also interesting on its own. However, ASH is a well characterized nociceptive neuron, but odor-off upon STAV should be a favorable condition.

IV) AIA activity has been assigned to attractive odorant responses (Larsch, 2015; Dobosiewicz, 2019), which is inconsistent with enhanced responses upon STAV/LTAV. According to Figure S13 its baseline seems up-regulated rather than the response. I don't see reason for the authors' interpretation that AIA activity encodes valence.

V) Also RIA imaging seems not to reveal a clear picture what its role might be. The individual traces contain very variable interesting fluctuations making interpretation of mean traces difficult. To correctly interpret activity in RIA nrV or nrV one would need to know the head-bending position or head motor neuron activity (Hendricks, 2012 and Liu, 2018).

VI) Moreover, it is unclear from which neurons AIA and the RIAnrS domain receive their respective inputs under the various conditions. There is no consistent picture emerging, when comparting to the sensory neuron responses. Which sensory-to-interneuron communications change in the various conditions?

VII) Why were AIY and AIZ not imaged, which are major targets of AWC and AWA? Despite the comprehensive sensory neuron imaging, there are obvious gaps with respect to interneurons.

Altogether, this is very confusing, and it is questionable whether any of the significant activity changes allow interpretation of how stimulus, conditioning, valence, behavioral response LTM or STM are separately encoded. Although I highly appreciate the authors' efforts in comprehensive imaging, at the current state the study presents scattered results that don't fit into an intuitively understandable circuit model.

2. One possible source for the difficulty to assemble a coherent picture may stem from the different exposures to butanone used in the different assays used in this study. For example, the inability of the multiworm tracker assay (Figure 4) to recapitulate the appetitive memory shown in the chemotaxis assays (Figure 1) may stem from the absence of diacetyl as a competitor odor (or ethanol background), or from the difference in the concentration of the butanone. The authors acknowledge the different results from the two behavioral assays (Page 17), but they do not provide an explanation the different outcomes. Similarly, the calcium imaging experiments were also conducted without diacetyl, and with higher concentration of butanone. It is possible that the response of AWA to diacetyl is modulated by the butanone learning paradigm (perhaps as part of the requirement of AWA to the butanone memory, Figure S12). I therefore wonder whether calcium imaging experiments should have been performed for switches between butanone and diacetyl; same for the behavioral analyses in Figure 4, this would have been more informative had the authors included diacetyl.

3. Regarding the interpretation of the AWC-ablation results (Page 10 and Figure S12A-B). The AWC neurons (and specifically AWC-ON neurons) are required for chemotaxis towards butanone (Bargmann 1993, Wes 2001, Tsunozak, 2008) and are commonly cited as such in the literature. One therefore cannot easily untangle the role of the AWC neurons in butanone memory from their more basic role in butanone sensation (and chemotaxis). As also seen in Figure S12, the AWC ablated animals do not show chemotaxis towards butanone (while they do show some surprising aversion). Therefore, the claim that the AWC neurons are required for appetitive memory cannot be easily justified. In addition, Torayama et al., (2007, dio:doi.org/10.1523/JNEUROSCI.4312-06.2007) demonstrated that the AWC-OFF neuron is sufficient for chemotaxis towards butanone and that the AWC-ON neuron is also required for appetite memory of butanone. This work should be cited and the claims for the requirement for AWC neurons for memory should be clarified. The finding that the AWC-OFF neuron can also react to butanone, corroborates the behavioral data from Torayama, et al., J Neuroscience (2007) and can be discussed in text.

4. All interneuron imaging data lack the essential buffer-buffer switch controls.

5. The AVA activity (Figure S14), which can be assigned to fictive reversal behaviors, shows reduction upon both odor ON and OFF, in all conditions. Data in Figure 4E-F suggest that reversal modulation is implicated in STAV and LTAV. I therefore conclude that this cannot be recapitulated in chip imaging conditions. Did the authors perform enough repetitions? I fear that AVA just responds to switches in the microfluidic chip, but the essential buffer-buffer switch control is missing.

6. Regarding Figure 4E, the number of reversals per cm is calculated instead of the probability reversal per worm at this area. The former calculation should be made to normalize to the number of worms reaching this area in the agar plate.

*Reviewer #2 (General assessment and major comments (Required)):*

The manuscript reports the experiments and the results that aim to identify the principles used by *C. elegans* nervous system to encode associative memories. The study analyzes 4 types of associative learning, namely S/LTAV and S/LTAP, side-by-side in behavioral strategies and neural activities. The experimental design helps to identify unique behavioral and activity patterns associated with each type of memories. The results from this systematic analysis provide a large dataset that is informative for understanding sensory encoding and plasticity. Based on these results, the authors propose that sensory neurons have central roles in memory formation. The manuscript will benefit from addressing several questions in results, data analysis and interpretation.

1. In methods for imaging, the authors describe that "the animals were starved for 20 minutes on empty NGM plates. For imaging multiple neurons, worms were also paralyzed using 10 mM levamisole dissolved in chemotaxis medium. The worms were habituated to the restraint and paralysis for additional 10 minutes within the chip." The worms were starved 20-30 minutes before the imaging experiments. This treatment is important for interpreting imaging results because worms change reversal rates after being on an empty plate for 30 minutes, likely due to changes in the activity of the neurons that regulate reversals. The authors should describe this condition in "Results" and integrate this condition when interpreting and discussing the results.

2. The imaging experiments on sensory neurons and interneurons are done mostly using worms that express calcium indicators in many neurons. The validation of the cell identities needs to be shown.

3. The authors report increased activity in AWC for STAV and STAP. Because STAV and STAP generate opposite effects on chemotaxis to Butanone, the function of these AWC results on chemotaxis behavior need to be better clarified.

4. In addition, Cho et al. *ELife* 2016 reports that conditioning worms similarly as STAV in this paper decreases AWC activity. The difference in this result and the one reported in the manuscript warrants some discussion.

5. The author tested the function of AWC and AWA in learning using cell ablation. Because loss of sensory neurons can impair both sensing and learning, the interpretation of the results need to include these possibilities.

6. ASH shows increased activity after STAV. But ablating ASH does not change STAV learning ability. These results suggest that the increased ASH activity does not regulate STAV. The possibility needs to be included in the discussion of these results.

7. Figure 4 reports significant changes in speed and reversal rates after STAV and LTAV. However, none of the interneurons that are known to regulate speed or reversals shows any significant activity change. These results need to be carefully discussed.

8. Increased activity in AIA after LTAV predicts decreased reversals (Larsch et al. Cell Rep 2015). It does not explain increased reversal rate observed in chemotaxis observed in LTAV. In addition, increased RIAnrs predicts increased curving towards an attractant (Liu et al. Neuron 2018). It does not explain increased reversal rate and increased deviation in LTAV. One possibility is that S/LTAV generate changes in interneurons that regulate reversals (AIY, AIB, AVA, RIM etc) in a distributed manner. Although none of these interneurons shows a detectable change, their collective activity changes produce increased reversal rate. In addition, after LTAV, reversal rate increases for about 2 fold, but deviation only increases a fraction of the mock level. It is likely that increased reversal contributes to the increased deviation in chemotaxis. The author should include these possibilities in discussion.

9. Lemieux et al. Cell 2015 reports an increase in AVA activity after 1-2 hour starvation. The difference in this result and the results on AVA in this study warrants some discussion.

[Editors’ note: further revisions were suggested prior to acceptance, as described below.]

Thank you for submitting your article "Principles for coding associative memories in a compact neural network" for consideration by *eLife*. Your article has been reviewed by 3 peer reviewers, and the evaluation has been overseen by a Reviewing Editor and Ronald Calabrese as the Senior Editor. The reviewers have opted to remain anonymous.

The resubmitted manuscript was assessed by three reviewers, whose review comments follow. All reviewers appreciate the large amount of efforts put to obtain additional data. On the other hand, each reviewer points out interpretation and presentation problems. Considering this, I need to call for a revision of the manuscript, including additional data analysis and statistical tests. If authors decide to revise the manuscript as recommended, the revised manuscript will be re-assessed by the reviewers. Please note that reviewer#1 is identical to reviewer#2 in the previous submission, and reviewer #4 is identical to reviewer#1 in the previous submission, while reviwer#3 was newly recruited.

As described by authors in Figure 1 and Supplementary figures 3, behavioral and neuronal activity changes caused by learning paradigms could be categorized to stimulus-specific (caused by CS), treatment-specific (caused by US) and training-specific (caused by CS and US), and each of these changes could occur in either of appetitive training, aversive training or both, in the same of opposite directions. On the other hand, in describing neuronal activities, only "stimulus code" and "valence code" are evaluated. This is causing considerable confusion and need to be amended.

Also, presentation and evaluation of regression analysis and methods for evaluation of RIA activities need to be reconsidered. In these and other aspects pointed out by the reviewers, authors need to improve the presentation to efficiently and thoroughly convey the results, because the whole dataset is valuable to the readers in the field.

*Reviewer #1 (Recommendations for the authors):*

The revised manuscript has improved significantly.

First, the newly added calcium imaging experiments on sensory neurons and interneurons in response to BUT and DA helped to interrogate neural activities in a condition more comparable to the chemotaxis assay conditions. These analyses help to interpret potential function of the observed neural activities under the naive and training conditions.

Second, they found that a linear combination of presynaptic sensory activities explained the postsynaptic interneuron activities to various degrees. This analysis helped with a coherent understanding of a large amount of imaging results by showing potential flows of neural signaling.

Finally, they also performed detailed behavioral analysis on chemotaxis. The results examined multiple parameters (speed, reversal, direction) and the authors aimed to make a connection between the behavioral results and the imaging results. This is an informative addition. However, some clarifications are needed for their interpretation of the results "We did not observe behaviorally relevant differences in the animal's speed nor in t heir reversal frequency (though p values indicated significance, Figure 7E-H)." If they did not use the statistical test results to interpret the data, it will be useful to clarify how they drew the conclusions.

The revised manuscript has improved significantly.

First, the newly added calcium imaging experiments on sensory neurons and interneurons in response to BUT and DA helped to interrogate neural activities in a condition more comparable to the chemotaxis assay conditions. These analyses help to interpret potential function of the observed neural activities under the naive and training conditions.

Second, they found that a linear combination of presynaptic sensory activities explained the postsynaptic interneuron activities to various degrees. This analysis helped with a coherent understanding of a large amount of imaging results by showing potential flows of neural signaling.

Finally, they also performed detailed behavioral analysis on chemotaxis. The results examined multiple parameters (speed, reversal, direction) and the authors aimed to make a connection between the behavioral results and the imaging results. This is an informative addition.

However, some clarifications are needed for their interpretation of the results "We did not observe behaviorally relevant differences in the animal's speed nor in their reversal frequency (though p values indicated significance, Figure 7E-H)." If they did not use the statistical test results to interpret the data, it will be useful to clarify how they drew the conclusions.

Also, RIA synchronized calcium events were calculated as events in which both axon domains have time derivatives that were positive (influx) or negative (eflux) (Jin et al. 2016; Hendricks et al. 2012). In Jin et al., the authors used > 0.005 (% per second) or < – 0.005 (% per second) as a threshold. As the authors showed in Figure S6, it is not clear whether the shaded areas were synchronized events.

*Reviewer #3 (Recommendations for the authors):*

In this manuscript, authors established learning paradigm that realizes both appetitive and aversive learning and both short-term and long-term memory using the same CS, butanone. By using this type of learning, they made systematic analyses of learning-dependent changes in neuronal activity including many sensory neurons and several core interneurons. Upon revision, they added buffer control and added AIY interneurons. Also, addition of switching between butanone and another odorant diacetyl, as a control odorant, causes the results more interpretable. Now it has a total of 30 supplementary figures, which is remarkable. The relationship between the neuronal activity and behavior is still unclear, and authors claim that attraction and aversion behaviors may be regulated by polulation coding, which might be true. The major contribution of the study is a comprehensive analysis of sensory neuron responses to odorant which clearly changed by short-term training, and a set of interneurons, whose activities change by both short and long-term training. These data are informative for the community. Upon revision, authors also show that activity of interneurons can be explained by linear combination of sensory neuron activities.

However, in Figure 6 and other parts of the manuscript, authors use the word "trial-averaged activities". This is misleading because this word is commonly used for averaging across trials rather than averaging across animals for each trial (repeat of stimulus according to authors' definition of "trial"). It needs to be reworded. The averaging treatment is explained in Figure S27F, but this figure is not easy to understand, partly because only trial 1 is labelled even though two trials are shown in the drawings.

*Reviewer #4 (Recommendations for the authors):*

The authors made extensive efforts in addressing all my previous comments. The present manuscript is a tour de force through an impressive amount of non-trivial experiments. The new paper better highlights some very interesting aspects of their findings, like the distributed nature of memory components and of how sensory information is passed to the first layer of sensory interneurons. Still, I have some troubles in following the authors' heuristics in how they interpret the various experimental conditions. Further, the paper needs some improvements in the applied statistics.

1) Comparing trained animals with their respective matched MOCK conditions is a crucial control for a specific memory trace in a neuronal activity pattern. However, across other conditions the authors applied very different procedures to the worms where many experimental variables were changed, like repeated wash and transfer cycles, etc. In many cases these procedures caused strong effects on neuronal activity i.e. significant differences in MOCK-trained vs naïve (e.g., Figure 4F), hence the neuronal responses are modulated by starvation or procedure associated stress. When various MOCK conditions give different results, one cannot conclude from differences in the respective training conditions that the neuron is encoding valence. For example Figure 4I,J,K: the most obvious difference in AWCon between STAV and STAP is coming from the differences in MOCK vs MOCK, while STAV and STAP cause tiny and equal changes relative to their MOCK traces, hence AWCon is mainly modulated by the US (procedure) but does not represent valence.

Altogether, the resulting picture is still vastly confusing; the authors frequently conclude that certain neuronal activity changes "code" for various task parameters, but this is not really shown i.e., simply showing correlates of memory traces is not sufficient to make these statements. Moreover, It will be extremely challenging for the reader to navigate through the forest of > 30 Figures. I find it unsatisfactory that the authors main response to this is, in other words, – memory components are encoded in a distributed but in a somewhat unexplainable way-. The authors should make an effort to address this. I would suggest to revise Figure 4 (and Figure 3) to include all MOCK vs MOCK and MOCK vs naïve, also the sign of the δ values should be depicted, as well the significance levels of each matrix entry should be depicted on top of each square (not just labeling the neurons yellow). This would give a better overview. Then use this data matrix to train a classifier and test whether the task parameters, CS, US ( = the 2-4 different procedures), positive- and negative valence can be indeed decoded, and along which dimensions the major coding axes reside.

2) There are many instances where the authors applied repeated stimulus sequences leading to an inflation of datapoints, which are treated by the respective statistical tests as independent measures. This is incorrect and I fear many of the seemingly statistically significant results could fade-away when properly tested for. In all cases where the authors applied repeated stimuli mean responses per animal should be calculated so that each datapoint corresponds to one animal. In all figure captions n numbers must be indicated.

3) I find the results of the regression analysis quite interesting but unfortunately the authors did not control for overfitting, a frequent pitfall in regression procedures particularly when multi-parameter models are used. The authors should devise a cross-validation procedure and report R2 values for held-out test data.

4) AWCon and AWCoff in the multi-neuron recordings were classified based on high/low responses in MOCK, a classifier that was empirically inferred from imaging experiments performed in a marker line. This classifier seems unambiguous in STAP MOCK only, but how were AWCon and AWCoff classified in the multi-neuron experiments in all other conditions"?

5) AWC and AWA (Figure S21, S7E-L) show different results in different imaging lines. I think the authors do not provide a convincing explanation for this.

[Editors’ note: further revisions were suggested prior to acceptance, as described below.]

Thank you for resubmitting your work entitled "Principles for coding associative memories in a compact neural network" for further consideration by *eLife*. Your revised article has been evaluated by Timothy Behrens (Senior Editor) and a Reviewing Editor.

The manuscript has been improved and all reviewers are very enthusiastic about having the revised paper published in *eLife*, but there are some remaining issues that need to be addressed, as outlined below:

Please consider suggestions by Reviewer #1 on AIY activity and similar changes in AP and AV paradigms. Also, please consider suggestions (1) – (4) by Reviewer #3. We look forward to the revised version rather soon.

*Reviewer #1 (Recommendations for the authors):*

The revised manuscript has further improved by addressing concerns raised in previous reviews.

More specifically, the authors have further analyzed their data by focusing on paradigm-specific activity of sensory neurons in response to BUT – DA stimulation patterns. This focus allowed them to see more robust effects generated by different types of experience to better support the data interpretation. Their interpretation also improved and are now more supported by the results.

Their focus on using the linear combination of presynaptic sensory activities to explain AIY activity is appropriate given the sample size and variabilities of their imaging results across animals and across trials. It is informative to see that AIY activity can be explained by those of anatomically connected sensory neurons. It is worth noting that AIY activity is best explained by the activities of sensory neurons under mock conditions. This might suggest the role of these neurons in responding to the US experience, not necessarily the association of CS with US.

The principal component analysis in Figure 7 and supplementary figures is an improvement on data analysis and useful in dissecting complex neural responses in a large number of different imaging conditions implemented in this study.

Another clear feature of the neurons characterized here is that they seem to show similar patterns of activity changes in AP and AV paradigms (for example, as shown in Figure 5 for sensory neurons AWA, interneuron AIY and AIA; for AWC in Figure 3; for RIA in response to DA-BUT stimulation pattern). Although AP and AV paradigms showed different effects on RIA in response to control-BUT stimulation, it is not clear how the changes relate to chemotaxis behavior. The authors speculate this feature as one indicator for combinatorial effects. This is possible but remains to be tested in future studies.

When analyzing locomotion in chemotaxis, the revised study removed the results on long-term training and focused on short-term training. The results are easier to be understood than before. The short-term training paradigms modulated the directionality toward the CS, but not the speed and reversal frequency.

As a whole, the study produced a large collection of activity recordings in a handful of sensory neurons and a few interneurons with the goal to explain chemotaxis behaviors under 4 learning paradigms. Although more work is needed to explain how these activity patterns relate to behavior, the collection of data provides hypotheses for future studies on function of the neurons implicated in the learning paradigms based on calcium imaging results and provide useful references for similar studies in the field.

*Reviewer #3 (Recommendations for the authors):*

Upon revision, authors made a good job to improve the quality and readability of the manuscript. Additional evaluation of data was performed such as cross validation of linear regression and classification models, which adequately evaluates authors' assertions and strengthen them.

The concern by reviewer #3 was appropriately addressed, with the exception of the following.

This reviewer recommends publication of the paper because it is a comprehensive analysis which adds precious information to the research community, provided that the presentation flaws as listed below are corrected.

1) As previously pointed out, definition of "trials" needs to be explicitly provided. It is understandable that reflecting the revision history, there are several different protocols for the imaging experiments. For this reason, authors need to carefully try to avoid confusion for the readers. In the current version, the figure depicting "trials" appear in Figure 6—figure supplement 1I, while the word first appears in the text in line 255 where authors say "most animals (or trials) showed homogeneous responses", and here, readers may be puzzled. In Methods, there is also no such word, though it is described that repeated stimuli were applied in experiments in Figure 2B and Figures4-5.

2) p23, page 581: related to the previous comment by reviewer #1 (third in Major point), there is not enough basis provided in this paper that leads to the conclusion that head swing control (klinotaxis which RIA has been implicated) is the major mechanism that causes learning-dependent modulation of chemotaxis. The other major mechanism known, klinokinesis, involves changes in the turning frequency DEPENDING ON chemical concentration changes, where animals backup and turn more frequently when facing away from the odor source and odor concentration decreases upon its locomotion. It would also cause the deviation angle smaller.

3) p19, line466 and later: It is convincing that AIY changes its activity after LTAV training in DA-BUT imaging experiments. On the other hand, changes in RIA are not convincing. Figure 5G,H, RIA response is suggested to be increased, but it looks like the conclusion is based on two outlier animals. Also in Figure 3—figure supplement 4K/L, RIA activity looks increased even without BUT stimulus. Therefore, long-term changes in RIA activity needs to be toned down, at least in the discussion (where RIA is so heavily discussed in the current version). As authors depicted in Figure 9B, it is clear through the extensive data by the authors on sensory neurons that there are virtually no changes in the sensory neuron responses after long-term memory, but the behaviors change. As a logical consequence, interneuron activities need to be changing. However, observed changes are limited to AIY LTAV so far, and therefore hitherto unobserved changes may underlie the behavioral changes, for example that of AIB or any other neurons in the circuit.

4) p18 line 450: It is misleading to suggest sensory neurons are not involved in long-term memory. Authors only looked at calcium level in sensory neurons. However, there are multiple cases known in *C. elegans* and other animals where synaptic output machinery is modulated by learning, which is not reflected in calcium imaging of the sensory neurons; rather, it will be reflected in the response of downstream interneurons.

5) It is a bit of a concern that in cross validation in Figure 4—figure supplement 7A, authors used all individual animals for training the model, especially given that any single neuron can be a predictor, which suggests overfitting. However, revisions are not requested on this as per the principle of the journal.

---

## [Author Response]

[Editors’ note: the authors resubmitted a revised version of the paper for consideration. What follows is the authors’ response to the first round of review.]

We appreciate the newly developed paradigm of associative learning which can assess both appetitive and aversive learning as well as both short and long-term memory. Authors further went on to perform calcium imaging of multiple neurons simultaneously, and determined the odor response of these neurons before and after each of the four combinations of conditioning. This is a tremendous work, which is truly admirable. However, unfortunately, many of the results are not readily interpretable and there are multitude of apparent inconsistency in view of the authors' conclusions as described in the reviewers' comments. Although we cannot continue the review process further at this time so that authors can submit their results to another journal, if authors decide in the future that above concerns are eliminated with additional experiments, we will be happy to receive a new submission.

We have added experimental data for the AIY neuron which is a major interneuron downstream of the chemosensory neurons, and added the missing buffer controls for the inter- and command-neurons (per the reviewer’s request).The first version of the MS highlighted only ASI as a variable neuron. With the extended newly added dataset, we now provide a comprehensive analysis for all chemosensory neurons.The newly collected data also allowed us to introduce new analyses where we show that interneurons' activity can be explained using a linear combination of the distributed memory coded in the sensory neurons. Moreover, we show that the relative contribution of the sensory neurons is modulated in an experience-dependent manner.

Overall, we have now added 4 new figures to the main text and 15 supplementary figures. In the following, we address each of the points raised and specify the changes made in the manuscript.

Reviewer #1 (General assessment and major comments (Required)):In this study Pritz et al., expand the behavioral paradigm for of short- and long-term memory of appetitive and aversive odorant responses. This assay alone is a nice contribution to the field of learning and memory in *C. elegans*, as these four degrees of freedom can be specifically switched, while leaving other conditions constant. The authors then underwent a calcium imaging tour de force applying multi-neuron imaging to search for underlying memory traces in neuronal activity. They examine how the responses of neurons are modulated by the memory conditions, at different stages of sensory-motor flow, i.e. including the primary sensory neurons, interneurons and motor-command neurons. They find that the short-term memory conditions mainly modulate the response of the sensory neurons, while long term memory modulates the response of interneurons. A basic claim of this paper, therefore is that short- and long-term memory segregate at the circuit level, which is a very interesting result. Finally, the authors analyze how behavioral parameters such as turning, and reversal rates are modulated by the different memory paradigms.Comments:1. Unfortunately, while the authors indeed find memory traces in neuronal activity of some neurons, these results do not fall into place to allow for a clear comprehensive model how these changes in neuronal activity relate to the underlying changes in behavior.

We followed the suggestions provided below which greatly helped us to better understand the plasticity within individual neurons and their collective impact on behavioral changes. For this, we have now added a substantial amount of new data to match the behavioral assays with imaging conditions. These included re-imaging all neurons and repeating the behavioral assays (in short-term paradigms) while adding diacetyl as an alternative choice and using a higher concentration of butanone. Herein, we list the major changes to the manuscript which are relevant to all comments. We also provide a detailed answer for each of the points that follow.

Based on the original and the newly-added data we conclude that short-term memory components are distributed across the sensory neurons, and that it is virtually impossible to relate a modulated activity of a single sensory neuron to a concrete behavioral output. Short-term experiences change the sensitivity of many sensory neurons, and some neurons code both the experienced valence and the stimulus components of the memory together. It is therefore the integrative activity of the sensory neurons that expresses the adaptive response. Based on this strong experimental evidence, we now formulate these statements as part the new MS version conclusions.Regarding the activity of interneurons, we now show in detail how the modulated activity in sensory neurons can explain the activity of the downstream interneurons (AIY and AIA, RIA). In fact, interneuron activity is a linear combination of the sensory neurons activities, which intriguingly, change across the different training paradigms.Concerning locomotion parameters, we now observe behaviorally meaningful differences in the deviation angle (weathervaning behavior) which can be related to the changes observed in the RIA neurons. We no longer observe changes in reversals freq. and locomotion speed which were inconsistent with neuronal activities.Concerning long-term training paradigms, we refrain from making behavioral interpretations based on interneuron activities, as these, indeed, don't allow such inferences. We now only state how training-specific paradigms modulate the activity of individual interneurons, and that it is presumably their complex integrative activity that gives rise to the behavioral outputs.

Below we provide a detailed point-by-point response for each neuron in light of the new data.

I) AWC-ON neurons seem sensitized to odor-off stimulus both upon STAP and STAV, this is interesting but cannot explain any of the changes in behavior.

We agree. As explained in the general remarks above, behavior is difficult to explain when considering a single neuron (e.g., AWC ^ON)^ activity alone. Our newly acquired data show that the activity of many chemosensory neurons is modulated in an experience-dependent manner (see Figure 4 A, B), and changes in individual sensory neurons cannot readily explain the overall behavior. It is presumably the integration of multiple sensory neurons that expresses the adaptive behavior. This interpretation is supported by evidence from freely moving animals (Hallinen et al., 2021 *eLife*), in which a population coding of motor neurons was by far a superior predictor of locomotion behavior than any single command or motor neurons. Given that this paradigm is true for motor neurons (direct effectors of locomotion behavior), it is conceivably true also for individual sensory neurons which are positioned more distantly from the locomotion actuators and hence are presumably less reliable in predicting modulated behavior.

Moreover, our current understanding of the role of individual sensory neurons in features of locomotion behavior largely stems from experiments with non-physiological interventions (overstimulation by optogenetics, ablations etc.) that might overemphasize their contribution to natural behavioral metrics. We, therefore, refrain from trying to infer behavioral outputs directly based on the m odulated activity of the AWC ^ON^ neuron.

We added a new section to the discussion titled “Memory components are widely distributed across sensory- and interneurons”, that details that short-term memory components are distributed across the sensory neurons. This section can be found on pages 28-30 lines 592-626.

In the Discussion section dealing with the integration of sensory neurons in interneurons, we also note that the activity of many sensory neurons are required to properly describe the activity of AIA interneurons pages 31-32 lines 668-676:

“An emerging paradigm is that memory components are distributed in multiple individual neurons, suggesting that it is the integration of sensory neurons’ activities that underlies interneuron activity, and hence, downstream behavioral outputs. For example, to explain activity of the AIA interneurons, the most significant contributors were the AWA neurons, which explained only ~23% of the overall activity, followed by the AWC neurons which explained ~13% (Figure 6 H,J). Many additional sensory neurons, with lower percentage-contributions were required to explain up to 60 % of AIA activity. It is therefore challenging to predict behavioral outputs based on activities of a single (or few) sensory- or interneurons.”

Finally, we explicitly state that is hard to infer behavior from a single neurons’ activity when discussing the behavior in relation to neural activity, page 34 line 726-729 :

“Moreover, as memories were encoded in a distributed manner, encompassing several chemosensory and at least three interneurons, it is presumably the integration of all their activities that gives rise to the modulated behavioral responses. As such, it may be very difficult to relate a modulated activity of a single neuron to a specific locomotion feature.“

II) AWA activity to odor-on is not significantly different to the respective MOCK conditions, so I don't think one should conclude anything from the small effect sizes reported here only for STAP vs STAV and STAV vs naïve. I am not convinced of the authors' statement AWA would encode valence.

We agree. According to the original data the changes in AWA neurons are small and may not seem convincing. We, therefore, imaged the activity of the AWA neurons in two new and independent experiments using two different stains (now appear in figure 4C-E and supplemental Figure S21). These assays included many more repeats which revealed substantial changes in AWA activity, particularly between STAV and STAP and their corresponding mock-trained cohorts. Thus, we believe the changes in the AWA neurons are genuine. Nevertheless, we explicitly state that it is the combination of several small changes within several neurons that are likely to give rise to the observed behavioral changes. We consequently refrained from any statement that AWA alone is responsible for the observed behavior.

We now note that the AWA neurons are only one type of neuron and that there are additional ones that encode the valence component (see Figure 8 D). This is also highlighted in our conclusion statements as reflected in the summary to figure 8, and we a lso explicitly stated on page 28, lines 594 to 599:

“For example, activities of the neurons AWB, ASJ, and ASI were modulated only when comparing positive and negative experiences, indicating that they code the valence component of the memory. Activities of AWA, ASER neurons, and AWC neurons (at a lower concentration of the CS) were modulated both in a stimulus- and a valence- specific manner, suggesting that these neurons code for both components.”

And we go on to conclude on pages 2 8-29 lines 600 to 602:

“These findings support the notion that valence and stimulus components of the memory are distributed across multiple individual neurons, rather than being stored in a single or a few neurons.”

III) ASH is sensitized to odor-off specifically upon STAV. This is also interesting on its own. However, ASH is a well characterized nociceptive neuron, but odor-off upon STAV should be a favorable condition.

We have now explicitly re-assayed this point. The new data is consistent with the original one showing small amplitude changes in ASH. Again, we conclude from these extensive experiments that it is difficult to infer behavioral outputs based on the small changes observed in the ASH activity (~10%). Indeed, the relative contribution of ASH to the overall activity of the interneuron AIA was ~1.6 (±0.66)% (figure 6H).

In addition, in our newly-added experiments, we assayed a strain in which both ASH and AVA were expressing an integrated GCaMP reporter. While we observed minute changes in ASH activity following training (supplementary figure S18A-F), this change was not followed by a significant change in the AVA neurons after any stimulus exchange at all (Supplementary Figure S26).

Our overall conclusion is, therefore, that these small changes in the ASH cannot explain the observable behavioral changes. This is again reflected as a main premise of the revised version of the manuscript.

IV) AIA activity has been assigned to attractive odorant responses (Larsch, 2015; Dobosiewicz, 2019), which is inconsistent with enhanced responses upon STAV/LTAV. According to Figure S13 its baseline seems up-regulated rather than the response. I don't see reason for the authors' interpretation that AIA activity encodes valence.

We agree, and our newly-added data indeed supports the notion that AIA activity does not code valence in the short-term paradigms (Figure 5 E,F). In particular, these new data consist of many more experimental repeats using a 10- fold lower CS concentration and the addition of an alternative choice (diacetyl). Thus, we believe these data better reflect the choice paradigm that we observe in the initial behavioral experiments (Figure 1). According to these data AIA codes the stimulus and not the valence (Figure 5E,F).

In the new version of the manuscript, we omitted any claims regarding AIA coding valence in the short-term paradigms.

Regarding the long-term paradigms, we re-analyzed the data and we still observe significant changes in AIA activity between LTAV and LTAP paradigms (Suppl. Figure S14B).

When comparing the appetitive (upper right panel) with the aversive (lower right panel) there is a clear significant difference in AIA activity.

We have now combined the two curves into a single graph to show this (Author response image 1, right):

**Author response image 1. sa2fig1:** 

We also considered the interesting base shift and noticed that after correcting for this shift (Author response image 1, left panel), we do not observe a difference between LTAV and the associated mock group.In the revised version, we therefore only note this finding in the Results section (page 11, lines 234 to 236).

“…while the AIA neurons showed significantly elevated responses following the long-term aversive training paradigm (supplementary Figure S14A-C).”

We also highlighted the fact that the change in long-term aversively trained animals might stem from the baseline shift (see legend to supplementary figure S14 page 21 lines 334 to 337):

“The activity of AIA following LTAV training is higher compared to its activity following LTAP training or naive controls. Note that the difference between LTAV and the associated mock controls might be due to a baseline shift.”

V) Also RIA imaging seems not to reveal a clear picture what its role might be. The individual traces contain very variable interesting fluctuations making interpretation of mean traces difficult. To correctly interpret activity in RIA nrV or nrV one would need to know the head-bending position or head motor neuron activity (Hendricks, 2012 and Liu, 2018).

We agree. The new data, which is much less noisy, showed a clearer picture. We observed a stimulus-specific change in the aversive experience (figure 5 G,H). Interestingly, aversively trained worms also showed a significant increased deviation angles (weathervaning locomotion) during chemotaxis. Thus, this correlation between RIA activity and its associated behavioral output may explain how a change in RIA activity affects behavior. However, more experiments showing a direct causal effect will be needed for that.

We now state all these findings on page 33, lines 700 to 710:

“Behavioral assays following short-term training revealed that positively-trained animals were more directed towards the CS, while negatively-trained animals were significantly less directed (Figure 7 C-D). Interestingly, aversive training also inflicted significant changes in sensory-specific responses of the RIA neurons (Figure 5G,H, supplementary Figure S29). As the RIA neurons dictate animals’ head position (Hendricks et al., 2012; Hendricks and Zhang, 2013; Ouellette et al., 2018), their modulated dynamics may explain the changes observed in movement directionality. Moreover, RIA neurons were previously shown to participate in the formation of aversive imprinting (Jin et al., 2016), further highlighting how past experiences may be converged onto RIA interneurons to shape behavioral outputs. However, while correlated changes in behavior and RIA activity may suggest a functional link, additional experiments are needed to establish a causal role.”

In addition, since we cannot account for the head position in our microfluidic chamber, we have now removed any interpretation based on the dorsal or the ventral sides of the neurites. We present the nrV and nrD activities in supplementary figures since these dynamics are important to deconstruct RIAnrS signal. Moreover, while precise correlation of head timing and neuronal activity is not possible, calcium dynamics from dorsal and ventral compartments in restrained and paralyzed animals have been used to infer implications in learning in memory previously by Jin et al., 2016, Cell. Some readers will find those dynamics informative.

VI) Moreover, it is unclear from which neurons AIA and the RIAnrS domain receive their respective inputs under the various conditions. There is no consistent picture emerging, when comparting to the sensory neuron responses. Which sensory-to-interneuron communications change in the various conditions?

We have now comprehensively addressed this point to provide a clear understanding of which sensory-to-interneurons communication routes are at play following each of the training paradigms.

The activity of AIY neurons, in most conditions, mirrors the activity of AWA neurons (compare Figure 5 A to C), with the exception that in aversively trained animals there is an additional activity that cannot be explained by AWA alone. This suggests that different sets of sensory neurons modulate AIY activity i n an experience-dependent manner.

To further investigate this intriguing possibility, we constructed a mathematical model where the activity of each of the interneurons may be represented by a linear combination of all the upstream sensory neurons. This analysis resulted in a new figure that we now added to the main text (figure 6) and three new supplementary figures (S27-S29).

Specifically:

When considering AIY activity, a linear combination of the AWA and ASER sensory neurons alone could explain up to ~80% of the AIY activity (Figure 6A-C). Interestingly, the relative contribution of each of these sensory neurons significantly differed across the different paradigms, including a switch from positive to negative values for ASER. These analyses indicate that sensory-to-interneurons communication routes are modified in an experience-dependent manner (figure 6A-C, supplementary figure S27).

When considering AIY activity, a linear combination of the AWA and ASER sensory neurons alone could explain up to ~80% of the AIY activity (Figure 6A-C). Interestingly, the relative contribution of each of these sensory neurons significantly differed across the different paradigms, including a switch from positive to negative values for ASER. These analyses indicate that sensory-to-interneurons communication routes are modified in an experience-dependent manner (figure 6A-C, supplementary figure S27).

A similar analysis for the AIA neurons revealed that many more neurons are needed to be considered to explain AIA activity. In particular, activities of AWA, ASK, and ASI neurons were differently weighted depending on the training paradigm, again, indicating an experience-dependent modulation of the relative weights (figure 6D-F, supplementary figure S28).

1. For the sensory-evoked signal in RIA neurons, a similar analysis resulted in poor overall fit, indicating that additional neurons, probably ones that we did not measure herein, are required to faithfully explain RIA activity. Indeed, according to the wiring diagram (supplementary figure S29G) RIA receives multiple inputs including other motor neurons which presumably dominate its overall activity.

For the sensory-evoked signal in RIA neurons, a similar analysis resulted in poor overall fit, indicating that additional neurons, probably ones that we did not measure herein, are required to faithfully explain RIA activity. Indeed, according to the wiring diagram (supplementary figure S29G) RIA receives multiple inputs including other motor neurons which presumably dominate its overall activity.

Nevertheless, despite the overall low fit, we could explain ~65% of RIA activity following aversive conditioning. This indicates an interesting possibility where the aversive training shifts the weights modulating RIA activity towards the sensory neurons that we studied herein, particularly, the URX, AIY, ASH and AWC ^ON^ neurons.

All these new analyses are now provided in a dedicated whole section in the results: ‘The activity of interneurons can be explained by a linear combination of the sensory neuron’s activities. ’, pages 19-21, 403-450:

“AWA and ASER neurons, which showed the greatest modulated responses following the short-term experiences (Figure 4C-H and supplementary Figure S19A), directly innervate the AIY and AIA interneurons (Cook et al., 2019; White et al., 1986; Witvliet et al., 2021). However, measuring dynamics of AWA and AIY neurons simultaneously from the same animal revealed a surprisingly low correlation between the two synaptic partners (Supplementary Figure S27A-C, Pearson correlations range between ~0.25-0.5 for the different training conditions), suggesting that additional neurons, and possibly spontaneous activities, contribute to the overall AIY dynamics.

To study how the modulated activity of the sensory neurons impacts the activity of the interneurons, we considered a simple mathematical model where interneuron dynamics is dictated by a linear combination of the sensory neuron activities. When considering AWA activity alone, only 4-44% (depending on the specific training paradigm) of the activity observed in the AIY neurons could be explained, even if AWA and AIY activities were simultaneously extracted from the same animal (supplementary Figure S27D,E).

We therefore added to the linear combination model activities of other sensory neurons, obtained by the pan-chemosensory reporter strain (P *osm-6* ::GCaMP3, supplementary Figure S27F), by using trial-averaged activities of the neurons in the model. Interestingly, a linear combination of two neurons only, AWA and ASER, sufficed to explain 55-83% of AIY activities across the different training paradigms (Figure 6A-C). Adding more sensory neurons to the linear model only marginally improved these predictions (Figure 6G,I and supplementary Figure S27G-L).

The weights (denoted by the regression coefficients) of the AWA and ASER neurons considerably varied across the different training paradigms and spanned negative and positive values (Figure 6C). This suggests that synaptic outputs, reflected by the relative weights in the linear model (**β**_i)_, were differentially modulated in a training-specific manner.

To explain activity of the AIA interneurons, additional sensory neurons needed to be considered (Figure 6D-F,H,J and supplementary Figure S28). In fact, the two neurons that mostly contributed to AIA activity (AWC ^ON^ and AWA) explained only ~30% of its activity, and considering five neurons (AWC ^ON,^ AWA, URX, ASK, and ASI) explained 48-78% of the AIA activity (across the different training paradigms). Similar to AIY, several sensory neurons switched from positive to negative weight values in a paradigm-specific manner, indicating training-specific synaptic modulations (Figure 6E-F).

In contrast, characterizing sensory-evoked signals in the RIA neurites using a linear combination of the sensory neurons was generally poor, explaining only 20-39% of the activity variance (supplementary Figure S29). The fact that RIA neurons receive many more inputs, many of which are not covered by our model, may explain the low fit when relying on signals from a few sensory neurons only (supplementary Figure S29G). Nevertheless, in aversively-trained animals, 65 % of RIA activity could be explained by a linear combination of nine sensory neurons together with the activity of the AIY interneuron. In this paradigm, the sensory neurons AWC ^ON,^ URX, ASK and ASI as well as the AIY interneurons were the major contributors to RIA activity, suggesting that in aversive conditions, RIA neurons are increasingly tuned to these neurons.”

And in the discussion, p ages 31-33, 648-697:

“A detailed quantitative analysis of the downstream interneurons, AIY, AIA, and RIA, revealed that their activity can be explained via a simple linear combination of the sensory neurons. It is striking that the relative contribution of each sensory neuron greatly varied across the different paradigms. For example, the ASER neuron coded both the valence and the stimulus components of the memory, and its activity positively, or negatively, correlated with AIY activity following aversive or appetitive experiences, respectively (Figure 6 A-C). Likewise, ASER activity significantly affected AIY activity in aversively-trained animals while its activity did not affect AIY activity in the associated mock-trained animal. These results demonstrate how learning-specific paradigms finely modulate the communication routes between neurons within the network.

While two sensory neurons sufficed to explain most of the dynamics in the AIY neurons, at least five neurons were required to describe the activity of the AIA neurons. For the RIA neurons, even 9 neurons only poorly explained their activity. The reason for these differences may be explained once considering the connectome (Cook et al., 2019; White et al., 1986; Witvliet et al., 2021). AIY neurons are postsynaptic primarily to chemosensory neurons, while AIA neurons are postsynaptic to many more neurons. The RIA neurons also receive multiple inputs, including several interneuron inputs, not accounted for in our model. Thus, the more complex the presynaptic input, the more neurons are required to faithfully describe the activity of the postsynaptic neuron.

An emerging paradigm is that memory components are distributed in multiple individual neurons, suggesting that it is the integration of sensory neurons’ activities that underlies interneuron activity, and hence, downstream behavioral outputs. For example, to explain activity of the AIA interneurons, the most significant contributors were the AWA neurons, which explained only ~23% of the overall activity, followed by the AWC neurons which explained ~13% (Figure 6H,J). Many additional sensory neurons, with lower percentage-contributions were required to explain up to 60 % of AIA activity. It is therefore challenging to predict behavioral outputs based on activities of a single (or few) sensory- or interneurons.

The prime purpose of the regression analysis was to reveal if and how sensory activities are modulated in an experience-dependent manner to control the activity of the downstream interneurons, rather than to provide an accurate fit prediction for interneuron activity. The modulated sensory contributions to interneuron activity, denoted by the regression coefficients (Figure 6 C,F and supplementary Figures S27-S29), may be viewed as a proxy to synaptic strengths. As such, it is compelling to speculate that our analyses may point to specific synaptic modulations that underlie the memory storage, though additional synaptic in-situ evidence for that is needed. Moreover, in addition to modulating responses to the CS butanone, the plasticity of individual neurons (and possibly synapses) also modulated the responses to the alternative choice diacetyl (Figure 4C-E), and presumably to other stimuli as well. This global plasticity highlights how a distributed code broadly affects network dynamics culminating in modulated chemosensory preference.

Of note, we considered GCaMP signals when analyzing the relative contribution of the sensory neurons to the overall activity of the interneurons. However, the slow GCaMP dynamics (compared to the rapid intracellular Calcium/voltage changes) may introduce correlative biases that increase the dependencies between the neurons. Nevertheless, we observed large differences between the regression coefficients across the various training paradigms (Figure 6C-F and supplementary Figures S27I, S28B, S29C), including a switch from positive to negative values. This demonstrates that the synaptic weights were significantly modulated, well above the possible correlative biases, thereby reflecting genuine experience-dependent changes.“

VII) Why were AIY and AIZ not imaged, which are major targets of AWC and AWA? Despite the comprehensive sensory neuron imaging, there are obvious gaps with respect to interneurons.

This is a very good suggestion. We have now analyzed activity of the AIY neurons across all the different experiments and conditions performed for the other neurons. We chose to focus on this neuron since it is not as densely connected in the network as AIZ, thus increasing the possibility to infer meaningful insights into how an interneuron processes sensory information in an experience-dependent manner.

Moreover, in all experiments, we coupled AIY imaging with AWA neurons, which is its key upstream sensory neuron. This positioned AWA as a cross-reference neuron since we could compare its activity in the pan-sensory reporter strain (*osm-6)* to its activity in the AIY/AWA reporter strain, and by this, relate the activity of all sensory neurons to AIY.

As stated above, these experiments demonstrated that AIY activity can be explained, for the most part, by the activities of two sensory neurons only: AWA and ASER.

A detailed description of AIY activity across the different training paradigms is provided in the previous comment (VI). We have now added to the main text all the data (figure 6 A-C and supplementary Figure 27). In Results section, we state on page 17 lines 370 to 377:

“Neural responses in the AIY neurons largely recapitulated the responses observed in the AWA neurons (Figure 5A-D and supplementary Figure S25M-P). This suggests that the AIY neurons, similarly to AWA neurons, encode both the stimulus and the valence components of the memory. In contrast, the AIA neurons showed differential responses between STAP-trained and the associated mock-trained animals, suggesting that this neuron type codes the stimulus component of the memory. STAP- and STAV- trained animals had similar responses, indicating that the AIA neurons do not encode the experience (valence) component of the memory (Figure 5E-F and supplementary Figure S24E-H).”

Altogether, this is very confusing, and it is questionable whether any of the significant activity changes allow interpretation of how stimulus, conditioning, valence, behavioral response LTM or STM are separately encoded. Although I highly appreciate the authors' efforts in comprehensive imaging, at the current state the study presents scattered results that don't fit into an intuitively understandable circuit model.

We appreciate that the original data provided an incomplete, and in part, inconsistent picture of the experience-dependent activity changes. But after repeating all imaging experiments and automated behavioral experiments for the short-term paradigms, with the modified conditions as suggested, we could provide much more complete, consistent, and definite results. We also significantly increased the number of experimental repeats per neuron, which ultimately allowed us to make stronger statements with higher confidence. Also, adding the analyses of the AIY neurons helped us to fill a crucial gap in our previous observations. From these new data a clearer picture of the circuit plasticity emerges:

Short-term experience broadly affects sensitivity of sensory neurons to the CS (butanone) and diacetyl (see points 1 I-III and 2).Short-term memory exists as distributed code in sensory neurons, in which the valence component and the stimulus component of the memory are distributed over many neurons (see points 1 I-III).Immediate downstream neurons (AIY, AIA) receive modulated input from sensory neurons (see point 1 IV, point 1 VIII).Activity of the downstream interneurons can be described as a linear combination of sensory activities. Experience modulates how sensory inputs are weighted.Ultimately all outputs converge onto RIA neurons that show differential activation in aversive conditions (point 1 VIII).Consistently with RIA observations, the directionality (weathervaning) during choice behavior in aversively trained animals shows large differences (see point 1 VI).

2. One possible source for the difficulty to assemble a coherent picture may stem from the different exposures to butanone used in the different assays used in this study. For example, the inability of the multiworm tracker assay (Figure 4) to recapitulate the appetitive memory shown in the chemotaxis assays (Figure 1) may stem from the absence of diacetyl as a competitor odor (or ethanol background), or from the difference in the concentration of the butanone. The authors acknowledge the different results from the two behavioral assays (Page 17), but they do not provide an explanation the different outcomes. Similarly, the calcium imaging experiments were also conducted without diacetyl, and with higher concentration of butanone. It is possible that the response of AWA to diacetyl is modulated by the butanone learning paradigm (perhaps as part of the requirement of AWA to the butanone memory, Figure S12). I therefore wonder whether calcium imaging experiments should have been performed for switches between butanone and diacetyl; same for the behavioral analyses in Figure 4, this would have been more informative had the authors included diacetyl.

Indeed, and we have now followed these important suggestions.

(I) We have now used a 10-fold lower concentration of butanone in the imaging experiments. This concentration is closer to the concentration used in the behavioral assays (though it is very difficult to match between the sensed concentration in airborne gradients to the exact concentration flown in the microfluidic device). Initial attempts to use a 100-fold lower concentration yielded poor sensory responses. Of note, in the original experiments, we tried to match butanone concentrations to the concentrations used during training. Thus, we have now covered a wide range of conditions which together provide a much more revealing picture (Figures3-5).

(II) We also included diacetyl as a second choice in both the imaging and the behavioral experiments (Figure 7).

Applying both of these changes provided a clearer view of the data with a better match between the behavioral results shown (figure 1), the imaging data (Figures2-5) and the locomotion parameters (Figure 7). Moreover, these new conditions now portrayed a clear difference between positive and negative experiences in both neural imaging and locomotion experiments.

Specifically, concerning the AWA responses, adding diacetyl as an alternative choice and reducing butanone concentration yielded clear differences across the different training paradigms. These experiments also indicated that AWA responses were modulated to both butanone and diacetyl (see Figure 4 C-E and supplemental figure S19A-C). These findings suggested that olfactory learning not only modulated responses to the CS (butanone) but also to diacetyl. Thus, olfactory experiences might broadly shift the animals’ chemosensory preference towards the CS depending on the valence of the preceding experience, such that the sensitivity to other stimuli are affected as well. We now state this in the discussion, page 32 lines 684 to 687:

“Moreover, in addition to modulating responses to the CS butanone, the plasticity of individual neurons (and possibly synapses) also modulated the responses to the alternative choice diacetyl (Figure 4C-E), and presumably to other stimuli as well.”

3. Regarding the interpretation of the AWC-ablation results (Page 10 and Figure S12A-B). The AWC neurons (and specifically AWC-ON neurons) are required for chemotaxis towards butanone (Bargmann 1993, Wes 2001, Tsunozak, 2008) and are commonly cited as such in the literature. One therefore cannot easily untangle the role of the AWC neurons in butanone memory from their more basic role in butanone sensation (and chemotaxis). As also seen in Figure S12, the AWC ablated animals do not show chemotaxis towards butanone (while they do show some surprising aversion). Therefore, the claim that the AWC neurons are required for appetitive memory cannot be easily justified. In addition, Torayama et al., (2007, dio:doi.org/10.1523/JNEUROSCI.4312-06.2007) demonstrated that the AWC-OFF neuron is sufficient for chemotaxis towards butanone and that the AWC-ON neuron is also required for appetite memory of butanone. This work should be cited and the claims for the requirement for AWC neurons for memory should be clarified. The finding that the AWC-OFF neuron can also react to butanone, corroborates the behavioral data from Torayama, et al., J Neuroscience (2007) and can be discussed in text.

We agree, and the same comment was raised by reviewer #2 (point num 5). For convenience, we include that response here as well.

It is absolutely true that by using mutants, we cannot differentiate whether the missing neurons are responsible for sensing or learning, or both. Since use of mutants is not really indicative, we removed all the data concerning the mutants and refrained from making any conclusion based on mutants use.

We also made sure to cite the indicated paper Torayama, et al., 2007 J Neuroscience, now appears as a citation in the supplementary discussion of naive neuronal responses, in the caption of supplementary Figure S9, page 15 line 241-244:

“AWC ^OFF^ neurons responded upon butanone removal in naive animals (see supplementary Figure S11 and supplementary Figure S7), thereby corroborating their roles in chemotaxis as previously described (Torayama et al., 2007).”

4. All interneuron imaging data lack the essential buffer-buffer switch controls.

We have now performed all these important controls and added the data to the manuscript (shown in supplementary Figure S9). It is evident that the activity due to flow changes between the buffers alone are significantly lower than those observed in response to the CS.

5. The AVA activity (Figure S14), which can be assigned to fictive reversal behaviors, shows reduction upon both odor ON and OFF, in all conditions. Data in Figure 4E-F suggest that reversal modulation is implicated in STAV and LTAV. I therefore conclude that this cannot be recapitulated in chip imaging conditions. Did the authors perform enough repetitions? I fear that AVA just responds to switches in the microfluidic chip, but the essential buffer-buffer switch control is missing.

We agree. We have now added the missing buffer-buffer controls showing that the command neurons did not respond to buffer-buffer switches (Supplementary figure S9). In the original data, we could perform only one stimulus presentation and removal, while in the newly added data we performed six stimulus exchanges. When retesting AVA in the new experimental conditions, AVA neurons did not show strong responses to stimulus exchanges (neither to butanone-to-diacetyl nor vice versa). Nor did we observe experience-dependent changes in AVA activity, except for baseline shifts. Likewise, in the new behavioral data, we did not observe changes in reversals. The effects of memory on AVA activities are stated in the Results section, page 18 lines 383 to 387:

“Downstream to the aforementioned interneurons are the command neurons that directly regulate animal behavior. We studied activity changes in two major command neurons, AVE and AVA, whose activity instructs a backward motion (Gray et al., 2005; Piggott et al., 2011). These neurons exhibited mostly baseline-level activity shifts that were unrelated to the switches between butanone and diacetyl (Supplementary Figure S26).”

Furthermore, we assayed AVA activity with and w/o levamisole and did not observe fundamental differences (Supplementary figures S26 C-H vs I-N). This suggests that the mechanical chip restraint, rather than the levamisole-induced paralysis, may underlie the fictive reversal behavior.

Together, we believe that interpretation of these results are yet difficult due to possible fictive behavior in the chip. In the new version of the MS, we only state that the behavior and imaging results are consistent, and acknowledge the limitation in drawing strong conclusions regarding AVA. P ages 33-34 lines 711 to 721:

“Quantifying reversal frequencies did not reveal a significant difference between the trained and the associated control animals (Figure 7G,H). The stable reversals frequency was also supported by the lack of activity responses in the backward-inducing command neurons (AVA and AVE, supplementary Figure S26). However, calcium imaging of command neurons in restrained animals is prone to artificial activity that may arise from fictive behavior (Hallinen et al., 2021), possibly rendering these neurons unresponsive to stimulation. Nevertheless, when using a higher concentration of butanone, similar to the concentration used during training (and in the absence of diacetyl), then the aversively-trained animals exhibited a significant increase in reversal frequency (Supplementary Figure S30F-G), which was also accompanied by stimulus-evoked calcium transients in the AVA and AVE neurons (Supplementary Figure S16).”

6. Regarding Figure 4E, the number of reversals per cm is calculated instead of the probability reversal per worm at this area. The former calculation should be made to normalize to the number of worms reaching this area in the agar plate.

Indeed, it was our mistake in labeling the units which actually denote reversals per cm wormtrack at this area (and not the non-normalize reversal frequency at this area). We changed the axis caption and explicitly explained it in the figure legend (Figure 7G-H for the new data and supplementary Figure S30F which is related to the original data), in the manuscript page 26, lines 533 to 535:

“(G) Plots of the reversal frequencies as a function of the distance from the target butanone. The units are given as reversals per centimeter worm track at the distance from the endpoint specified by the x-axis.”

As well as in the supplementary Figure S30, page 50, lines 718-720:

“(F) Mean reversal rates as a function of the distance from the target point for each of the four training paradigms. The units are given as reversals per centimeter worm track at the distance from the endpoint specified by the x-axis”

Reviewer #2 (General assessment and major comments (Required)):The manuscript reports the experiments and the results that aim to identify the principles used by *C. elegans* nervous system to encode associative memories. The study analyzes 4 types of associative learning, namely S/LTAV and S/LTAP, side-by-side in behavioral strategies and neural activities. The experimental design helps to identify unique behavioral and activity patterns associated with each type of memories. The results from this systematic analysis provide a large dataset that is informative for understanding sensory encoding and plasticity. Based on these results, the authors propose that sensory neurons have central roles in memory formation. The manuscript will benefit from addressing several questions in results, data analysis and interpretation.1. In methods for imaging, the authors describe that "the animals were starved for 20 minutes on empty NGM plates. For imaging multiple neurons, worms were also paralyzed using 10 mM levamisole dissolved in chemotaxis medium. The worms were habituated to the restraint and paralysis for additional 10 minutes within the chip." The worms were starved 20-30 minutes before the imaging experiments. This treatment is important for interpreting imaging results because worms change reversal rates after being on an empty plate for 30 minutes, likely due to changes in the activity of the neurons that regulate reversals. The authors should describe this condition in "Results" and integrate this condition when interpreting and discussing the results.

We agree, and we now state in the ‘Methods’ section that the worms are allowed to habituate in the microfluidic device with a reference to the fact that ~30 minutes of starvation can modulate AVA activity. I n the Material and Methods section, page 39, line 827-836:

“In preparation for live imaging and prior to loading the animals onto the microfluidic chips (Chronis et al., 2007), the animals were starved for 20 minutes on empty NGM plates. For imaging multiple neurons, worms were also paralyzed using 10 mM levamisole dissolved in chemotaxis medium. The worms were then allowed to habituate to chamber restraint and paralysis for an additional 10 minutes within the chip. The overall ~30 minutes starvation could in principle modulate activity of neurons associated with local search behavior, like for example AVA and others (Gray et al., 2005; Lemieux et al., 2015; Skora et al., 2018). We therefore may capture such starvation-induced changes, however, since we always compare trained and mock trained group animals which undergo exactly the same preparatory treatments, we essentially filter out these physiological effects.”

We also explain this when presenting the data regarding AVA, in supplementary Figure S16 page 23, lines 372-374:

“In general, preparations for imaging involved a ~30 minute starvation period. However, since we always compare trained and mock-trained group animals, which undergo exactly the same preparatory t reatments, we essentially filter out these physiological effects.”

And also in the legend o f supplementary Figure S26, page 41, lines 591-593:

“A general remark to all panels, imaging procedures required a 30 minute starvation period. However, since we always compare trained and mock trained group animals, which undergo exactly the same preparatory treatments, we essentially filter out these physiological effects.”

In the discussion, we discuss all these points together. Pa ge 33 lines 711 to 716:

“Quantifying reversal frequencies did not reveal a significant difference between the trained and the associated control animals (Figure 7G,H). The stable reversals frequency was also supported by the lack of activity responses in the backward-inducing command neurons (AVA and AVE, supplementary Figure S26). However, calcium imaging of command neurons in restrained animals is prone to artificial activity that may arise from fictive behavior (Hallinen et al., 2021), possibly rendering these neurons unresponsive to stimulation.”

2. The imaging experiments on sensory neurons and interneurons are done mostly using worms that express calcium indicators in many neurons. The validation of the cell identities needs to be shown.

We now provide a validation for the cell identities. A new supplementary figure S5 shows how we unequivocally identified each of the neurons. The caption to the legend explains the identification process. Briefly, we identified the neurons primarily based on anatomy, but also by functional activity that matched previous reports and knowledge. In cases where we could not verify a neuron identity (e.g URX, supplementary figure S8), we used neural-specific calcium-reporter lines. To resolve between AWC ^ON^ and AWC ^OFF,^ we used an additional line that differentially tags them (appears in a supplementary information and Supplementary figure S7 and detailed in Methods).

3. The authors report increased activity in AWC for STAV and STAP. Because STAV and STAP generate opposite effects on chemotaxis to Butanone, the function of these AWC results on chemotaxis behavior need to be better clarified.

Indeed, and this comment was also raised by reviewer #1. Our findings show that the modulated activity in the AWC neurons alone cannot explain the overall behavioral change. It is presumably the integration with other sensory neurons (and interneurons) that collectively affect behavior, and this is now one of the manuscript premises.

In the Discussion section dealing with the integration of sensory neurons in interneurons, we, therefore, note that the activity of many sensory neurons are required to properly describe the activity of AIA interneurons pages 31-32 lines 668-676:

“An emerging paradigm is that memory components are distributed in multiple individual neurons, suggesting that it is the integration of sensory neurons’ activities that underlies interneuron activity, and hence, downstream behavioral outputs. For example, to explain activity of the AIA interneurons, the most significant contributors were the AWA neurons, which explained only ~23% of the overall activity, followed by the AWC neurons which explained ~13% (Figure 6H,J). Many additional sensory neurons, with lower percentage-contributions were required to explain up to 60% of AIA activity. It is therefore challenging to predict behavioral outputs based on activities of a single (or few) sensory- or interneurons.”

We also explicitly state that is hard to infer behavior from a single neurons’ activity when discussing the behavior in relation to neural activity, page 34 line 726-729 :

“Moreover, as memories were encoded in a distributed manner, encompassing several chemosensory and at least three interneurons, it is presumably the integration of all their activities that gives rise to the modulated behavioral responses. As such, it may be very difficult to relate a modulated activity of a single neuron to a specific locomotion feature.“

Concerning the peculiar AWC activity modulations, our results suggest that AWC neurons code both STAV and STAP, thus the stimulus component of the memory (figure 3C-E). We find that this modulated activity is concentration-specific: In the original dataset, we observed changes in AWC activity when using a high butanone concentration, which was more similar to the training concentration. In the newly collected data, we used a 10-fold lower concentration of butanone (and added diacetyl as a choice). Still, we find the stimulus-specific differences in AWC activity attenuated (differences only between naive and trained groups figure 4 I-K for). This is also in line with results reported by the Bargmann group (Cho et al., 2016 *eLife*), where the AWC ^ON^ neuron displayed sensitization to stimulus removal at concentrations similar to the training concentration (1 mM) but not to lower concentrations.

We now explicitly discuss this premise in the Discussion, page 28-29, lines 593-602:

“We identified individual sensory neurons that code either the experience valence, or the conditioned stimulus, or both, components of the memory (Figure 8 C-D). For example, activities of the neurons AWB, ASJ, and ASI were modulated only when comparing positive and negative experiences, indicating that they code the valence component of the memory. Activities of AWA, ASER neurons, and AWC neurons (at a lower concentration of the CS) were modulated both in a stimulus- and a valence- specific manner, suggesting that these neurons code for both components. Interneurons generally coded both memory components (Figure 8C-D and supplementary Figures S24-S25). These findings support the notion that valence and stimulus components of the memory are distributed across multiple individual neurons, rather than being stored in a single or a few neurons.”

4. In addition, Cho et al. ELife 2016 reports that conditioning worms similarly as STAV in this paper decreases AWC activity. The difference in this result and the one reported in the manuscript warrants some discussion.

Indeed, Cho et al. demonstrated that AWC activity is decreased following training. However, this was observed at the lower concentration of butanone (e.g. 111 nM) as shown in that paper in figure 4 panel B (for aversive) and figure 5G (for appetitive). We note that for a higher concentration (1 mM), which was also used as the conditioning concentration, AWC actually increased (rightmost dynamics in these panels). A higher activity was also noted in the 10-fold lower concentration 111 μM. Thus, there is a full agreement between our results and the results reported by Cho et al.

We now added these comparisons to the Discussion section, page 29 lines 603 to 616:

“Of note, these coding schemes are sensitive to the concentrations and conditions used during memory formation and retrieval. For example, when using the same CS concentrations for training and for memory retrieval, then the AWA neurons exhibited valence-specific responses only (Supplementary Figure S11A-C). However, using lower CS concentrations during retrieval reveals that the AWA neurons show stimulus- as well as valence-specific responses (Figure 3C-E and supplementary Figure S19A-C). Similarly, the activity of the AWC ^ON^ neuron was modulated in a stimulus-specific manner only when evoking the memory using concentrations similar to the concentrations used during training (compare Figure 3C-E and Figure 4I-K). The high dependence of the AWC ^ON^ responses on the specific concentrations of the CS was also evident in a previous study where responses were observed only when stimulating the animals with the trained concentration, and regardless of the positive or the negative associating experience (Cho et al., 2016). Thus, the expression of the valence or the stimulus components of the memory greatly depends on the match between CS concentrations used during training and memory retrieval.

5. The author tested the function of AWC and AWA in learning using cell ablation. Because loss of sensory neurons can impair both sensing and learning, the interpretation of the results need to include these possibilities.

We totally agree, and the same comment was raised by reviewer #1 (point num 3). For convenience, we provide the same reply in both places:

It is absolutely true that by using mutants, we cannot differentiate whether the missing neurons are responsible for sensing or learning, or both. Since use of mutants is not really indicative, we removed all the data concerning the mutants and refrained from making any conclusion based on mutants use.

6. ASH shows increased activity after STAV. But ablating ASH does not change STAV learning ability. These results suggest that the increased ASH activity does not regulate STAV. The possibility needs to be included in the discussion of these results.

We agree. Since the ablation experiments are obsolete, we do not raise this possibility anymore. Moreover, we now added a mathematical model that analyzed the relative contribution of each of the sensory neurons and indeed found that ASH neurons play a minor role. We observed only small changes in ASH activity (~10%), and the relative contribution of ASH to the overall activity of the interneuron AIA was ~1.6(±0.66)% (figure 6 H-J and supplementary figure S28).

7. Figure 4 reports significant changes in speed and reversal rates after STAV and LTAV. However, none of the interneurons that are known to regulate speed or reversals shows any significant activity change. These results need to be carefully discussed.

Indeed. In the original data, we observed changes in the locomotion speed, while in the newly-added data, which better mimics the conditions in the imaging and chemotaxis assays (Figure 1, due to addition of diacetyl as a choice ad lower butanone concentration), the reduction in locomotion speed was minute. In fact there was no difference between trained and mock-trained animals. Hence, we no longer state that reduction in locomotion speed is a result of training essential to explain the choice behavior displayed in figure 1. The new data now appears in the new Figure 7, and the original data is now provided in supplementary figure S30. We therefore conclude that the speed feature is sensitive to the CS concentration and the specific chemotaxis conditions, on pages 33-34 line 711 to 725:

“Quantifying reversal frequencies did not reveal a significant difference between the trained and the associated control animals (Figure 7 G,H). The stable reversals frequency was also supported by the lack of activity responses in the backward-inducing command neurons (AVA and AVE, supplementary Figure S26). However, calcium imaging of command neurons in restrained animals is prone to artificial activity that may arise from fictive behavior (Hallinen et al., 2021), possibly rendering these neurons unresponsive to stimulation. Nevertheless, when using a higher concentration of butanone, similar to the concentration used during training (and in the absence of diacetyl), then the aversively-trained animals exhibited a significant increase in reversal frequency (Supplementary Figure S30F-G), which was also accompanied by stimulus-evoked calcium transients in the AVA and AVE neurons (Supplementary Figure S16). Similarly, while the speed of trained animals was not behaviorally significantly different from control animals, a significantly reduced speed was observed when using a higher CS concentration and in the absence of the alternative choice, diacetyl (Supplementary Figure S30D-E). These findings indicate that training induces a fine-tuned neural and behavioral plasticity that is sensitive to the exact experimental conditions.”

8. Increased activity in AIA after LTAV predicts decreased reversals (Larsch et al. Cell Rep 2015). It does not explain increased reversal rate observed in chemotaxis observed in LTAV. In addition, increased RIAnrs predicts increased curving towards an attractant (Liu et al. Neuron 2018). It does not explain increased reversal rate and increased deviation in LTAV. One possibility is that S/LTAV generate changes in interneurons that regulate reversals (AIY, AIB, AVA, RIM etc) in a distributed manner. Although none of these interneurons shows a detectable change, their collective activity changes produce increased reversal rate. In addition, after LTAV, reversal rate increases for about 2 fold, but deviation only increases a fraction of the mock level. It is likely that increased reversal contributes to the increased deviation in chemotaxis. The author should include these possibilities in discussion.

We agree. In our newly-added experiments, we imaged neural activity (including that of AIA neurons) and assayed locomotion parameters while presenting the animal with butanone together with diacetyl. This condition was more comparable to the conditions in which we characterized the choice behavior (figure 1). In these new settings, we did not observe a behaviorally meaningful change in the reversals (see figures 7 G-H), nor did we observe an increase in AIA activity in STAV (see figure 5 E,F). We therefore speculate that these differences arise due to the specific test conditions which included an alternative choice. Thus, we now refrained from relating AIA activity to behavioral outputs and limited our conclusion to the observation that interneuron activity is modulated following short and long term training paradigms. On page 28 lines 581 to 583 of the Discussion section:

“Interneurons showed activity changes following the formation of both short and long-term training paradigms, though, activity changes following long-term experiences were more prominent (Figure 8B-D, and supplementary Figure S14).”

9. Lemieux et al. Cell 2015 reports an increase in AVA activity after 1-2 hour starvation. The difference in this result and the results on AVA in this study warrants some discussion.

We actually observed the increase in AVA basal activity in our data as well.

In Lemieux et al. 2015, starvation alone (without stimulating the animals) increased AVA spontaneous activity. Interestingly, we observed a heightened basal AVA activity (and most prominently when imaging in the absence of levamisole, supplementary Figure S26 J and M), analogous to what Lemieux et al. reported. In fact, both trained- and mock-trained animals (in both positive and negative paradigms) showed consistently higher baseline activities than naive animals (Supplementary Figure S26).

However, in our study, we aimed to reveal whether the encounter with (or removal of) the conditioned stimulus butanone affects AVA activity following training, but we did not observe prominent state transition or activity changes between trained and mock trained animals (Supplementary Figure S26). Thus, the short starvation prior to imaging may have increased AVA basal activity, but the training itself did not affect AVA responsiveness to the conditioned stimulus.

We now state these observations in the legend of supplementary Figure S26 page 41, lines 566-571:

To exclude the possibility that levamisole affects activity, AVA neurons were imaged in the absence of levamisole. Again, no differences were observed in the response dynamics following the stimulus switch. The only significant changes were the shifts in the baseline activity (see mean activity graphs J and M, n=12-16 animals). These baseline shifts in aversively and appetitively trained animals could be due to starvation-induced increase in AVA activity as previously reported (Lemieux et al., 2015).

[Editors’ note: further revisions were suggested prior to acceptance, as described below.]

The reviewers have discussed their reviews with one another, and the Reviewing Editor has drafted this to help you prepare a revised submission.The resubmitted manuscript was assessed by three reviewers, whose review comments follow. All reviewers appreciate the large amount of efforts put to obtain additional data. On the other hand, each reviewer points out interpretation and presentation problems. Considering this, I need to call for a revision of the manuscript, including additional data analysis and statistical tests. If authors decide to revise the manuscript as recommended, the revised manuscript will be re-assessed by the reviewers. Please note that reviewer#1 is identical to reviewer#2 in the previous submission, and reviewer #4 is identical to reviewer#1 in the previous submission, while reviwer#3 was newly recruited.As described by authors in Figure 1 and Supplementary figures 3, behavioral and neuronal activity changes caused by learning paradigms could be categorized to stimulus-specific (caused by CS), treatment-specific (caused by US) and training-specific (caused by CS and US), and each of these changes could occur in either of appetitive training, aversive training or both, in the same of opposite directions. On the other hand, in describing neuronal activities, only "stimulus code" and "valence code" are evaluated. This is causing considerable confusion and need to be amended.

We have now extensively revised the manuscript and added comprehensive analyses of CS and US neural encoding. Specifically, we have identified the neurons that code for the US, and denoted in Figure 3A,B and 4A the impact of the US on neural activities. In the newly added Figure 7, we also identify the neurons that specifically encode the US+ and US-. Moreover, we extended Figure 1-supplement 3 to fully denote how the CS and the US can be inferred from the experimental groups together with terminology used throughout the manuscript. Finally, all coding neurons are summarized in Figure 9. Together, these additions provide a much more accurate analysis of the encoding logic. It also allows to better relate differences found in behavioral assays with those stemming from neural functional imaging.

Also, presentation and evaluation of regression analysis and methods for evaluation of RIA activities need to be reconsidered. In these and other aspects pointed out by the reviewers, authors need to improve the presentation to efficiently and thoroughly convey the results, because the whole dataset is valuable to the readers in the field.

We have now revised the analysis of RIA activities according to the reviewer’s comments. We present and analyze the RIA’s synchronous neurite signal as the time derivative of the activity rather than the raw activities. As for the regression analysis regarding RIA and AIA neurons, our new analysis indicated that the model consists of too many regressors and a limited number of data points. We therefore present the full valuable data without the regression analysis of these neurons.

Concerning the AIY neurons, our new analyses indicated that the data is well suited to reliably infer the experience-dependent changes in the regression coefficients, and by this, to support our conclusions regarding the changes in the synaptic communication routes (this analysis was added to Figure 6). We have now improved and simplified the data presentation of the results and made them more accessible to the reader (see Figure 3 AandB, Figure 4 A, Figure 6E). We also clearly stated the limitations of the regression method as to avoid any confusion in the interpretation of the results.

Reviewer #1 (Recommendations for the authors):The revised manuscript has improved significantly.First, the newly added calcium imaging experiments on sensory neurons and interneurons in response to BUT and DA helped to interrogate neural activities in a condition more comparable to the chemotaxis assay conditions. These analyses help to interpret potential function of the observed neural activities under the naive and training conditions.Second, they found that a linear combination of presynaptic sensory activities explained the postsynaptic interneuron activities to various degrees. This analysis helped with a coherent understanding of a large amount of imaging results by showing potential flows of neural signaling.Finally, they also performed detailed behavioral analysis on chemotaxis. The results examined multiple parameters (speed, reversal, direction) and the authors aimed to make a connection between the behavioral results and the imaging results. This is an informative addition. However, some clarifications are needed for their interpretation of the results "We did not observe behaviorally relevant differences in the animal's speed nor in t heir reversal frequency (though p values indicated significance, Figure 7E-H)." If they did not use the statistical test results to interpret the data, it will be useful to clarify how they drew the conclusions.

Indeed, we find that animal directionality (deviation angle) is the significant behavioral feature that is being modulated. As for speed and reversal frequencies, while statistical analyses show significant p values, their magnitude was low, close to borderline. We believe that these p values become significant due to the large sample size (e.g. thousands of track segments with speed values). Thus, while statistically significant, the relevant biological/behavioral significance is not obvious.

We now demonstrate this point by performing a detailed analysis, in which we simulated chemotaxis behavior by randomly choosing locomotion parameters (speed, reversal frequency, and deviation angles) drawn from a distribution constructed based on the experimental data (shown in the new panels of Figure 8 I-N). These simulations allow us to test the extent by which each locomotion feature shifts naive animal behavior towards the trained behavioral phenotype. For this, we sampled for two features from the naive group and one parameter was sampled from the trained group. By measuring the differences between locomotion this hybrid-type simulated behavior and that of real trained animals, we can infer the significance of the parameters that were sampled from the trained animals. These simulations indicated that the contribution of speed and reversal frequency to the observed changes in chemotaxis is negligible. In contrast, deviation angles significantly contributed to the experimentally observed behavioral changes (See figure 8 legend, Figure 8-supplement 1, and Methods for exact details of the simulation). This indeed corroborated our initial estimations.

We now provide a detailed explanation of these analyses and the considerations that led us to the above conclusions (pages 16-17, lines 407 to 421):

“Positively-trained animals were significantly more directed towards the CS target, while the negatively-trained animals were the least directed towards the target (each paradigm compared to its naive and matched mock-trained animals, Figure 8C-D). Notably, animals that underwent aversive training with BUT showed a high deviation angle with low variance towards the alternative choice DA (Figure 8C, arrow), suggesting that the negative training increased aversion from BUT, and concomitantly, enhanced attraction to DA. Animal’s speed and reversal frequency showed mild though significant changes (Figure 8E-H). To test the contribution of these changes to the overall behavior, we simulated animal chemotaxis based on experimental locomotion parameters (Figure 8I-N, and see Figure 8—figure supplement 1 for explanation). These simulations indicated that animal directionality (the deviation angle from the target) accounted for most of the behavioral changes, while the contribution of the speed and the reversal frequency to the overall change in the locomotive behavior was negligible (Figure 8—figure supplement 1). These analyses suggest that training mainly affected animals’ directionality features: more directed following positive training and less directed following an aversive training.”

Also, RIA synchronized calcium events were calculated as events in which both axon domains have time derivatives that were positive (influx) or negative (eflux) (Jin et al. 2016; Hendricks et al. 2012). In Jin et al., the authors used > 0.005 (% per second) or < – 0.005 (% per second) as a threshold. As the authors showed in Figure S6, it is not clear whether the shaded areas were synchronized events.

We have implemented this suggestion and used the coinciding time derivative of the dorsal and ventral compartment to describe RIA sensory-evoked signals (as described by Hendricks et al. and Jin et al). Since the accuracy of the derivative is noise sensitive, and noise levels markedly vary across experimental setups (e.g., different noise from the detector), we used different thresholds (±0.015) that we estimated from pressure controls inside the microfluidic device. We now modified Figure 5 and Figure 3-supplement 4 and 5, and Figure 5—figure supplement 2.

Reviewer #3 (Recommendations for the authors):In this manuscript, authors established learning paradigm that realizes both appetitive and aversive learning and both short-term and long-term memory using the same CS, butanone. By using this type of learning, they made systematic analyses of learning-dependent changes in neuronal activity including many sensory neurons and several core interneurons. Upon revision, they added buffer control and added AIY interneurons. Also, addition of switching between butanone and another odorant diacetyl, as a control odorant, causes the results more interpretable. Now it has a total of 30 supplementary figures, which is remarkable. The relationship between the neuronal activity and behavior is still unclear, and authors claim that attraction and aversion behaviors may be regulated by polulation coding, which might be true. The major contribution of the study is a comprehensive analysis of sensory neuron responses to odorant which clearly changed by short-term training, and a set of interneurons, whose activities change by both short and long-term training. These data are informative for the community. Upon revision, authors also show that activity of interneurons can be explained by linear combination of sensory neuron activities.However, in Figure 6 and other parts of the manuscript, authors use the word "trial-averaged activities". This is misleading because this word is commonly used for averaging across trials rather than averaging across animals for each trial (repeat of stimulus according to authors' definition of "trial"). It needs to be reworded. The averaging treatment is explained in Figure S27F, but this figure is not easy to understand, partly because only trial 1 is labelled even though two trials are shown in the drawings.

Indeed, we have incorrectly used the term 'trial-averaged’. We have now corrected all instances where this term appears and replaced it by ‘averaged each trial across all animals’. Moreover, we modified the explanatory scheme (now Figure 6—figure supplement 1I) to properly explain the repeated stimulus presentations, where each repeat was averaged across the different animals. This is now explicitly explained in supplementary material page 72, lines 2077 to 2084:

“(I) Our approach for relating activities from the pan-sensory reporter strain (osm-6::GCaMP) with activities extracted from a reporter strain expressing GCaMP in both the AWA and the AIY neurons. Data points from simultaneous recordings of AWA and AIY activities within the same animal (blue field) are synched. To accurately relate the neural activity of AIY neurons to activities of sensory neurons measured in different animals (i.e. ASER activities originating from the pan-chemosensory strain, red field), we averaged each trial across all animals (gray arrows). This minimized animal-to-animal variation and allowed the use of activities originating from different animals within the same regression model (black arrow).”

Reviewer #4 (Recommendations for the authors):The authors made extensive efforts in addressing all my previous comments. The present manuscript is a tour de force through an impressive amount of non-trivial experiments. The new paper better highlights some very interesting aspects of their findings, like the distributed nature of memory components and of how sensory information is passed to the first layer of sensory interneurons. Still, I have some troubles in following the authors' heuristics in how they interpret the various experimental conditions. Further, the paper needs some improvements in the applied statistics.1) Comparing trained animals with their respective matched MOCK conditions is a crucial control for a specific memory trace in a neuronal activity pattern. However, across other conditions the authors applied very different procedures to the worms where many experimental variables were changed, like repeated wash and transfer cycles, etc. In many cases these procedures caused strong effects on neuronal activity i.e. significant differences in MOCK-trained vs naïve (e.g., Figure 4F), hence the neuronal responses are modulated by starvation or procedure associated stress. When various MOCK conditions give different results, one cannot conclude from differences in the respective training conditions that the neuron is encoding valence. For example Figure 4I,J,K: the most obvious difference in AWCon between STAV and STAP is coming from the differences in MOCK vs MOCK, while STAV and STAP cause tiny and equal changes relative to their MOCK traces, hence AWCon is mainly modulated by the US (procedure) but does not represent valence.

Indeed, there are some differences in the training procedures. However, when training for the short-term paradigms, there is a small difference in the training procedures where the appetitive paradigm consists of only one extra transfer of the worms onto plates with food. While this extra handling may have some limited effects on the neural activity, we believe that the salient features are the presence or the absence of food, and as such, comparing the mock-trained animals to naive animals truly reflects the US.

Nevertheless, we now tone down our conclusions regarding the valence, acknowledging the above concern. We now mainly discuss such differences as US coding, that is, that the difference may be attributed to the entire training procedure (e.g., +/-food and differences in experimental procedures). For example, in the case of the ASER, we originally stated that it codes the valence, but we have now toned it down to say that there are differences depending on the presence or absence of food in the training paradigm (page 10, line 239 to 243):

“The ASER neuron displayed marked activity responses in naive animals upon the switch from DA to BUT. This response was completely lost in STAP-trained and all mock-trained animals, but not in STAV-trained animals (Figure 4E-G). This suggests that the ASER neuron may be coding the stimulus component of the memory as well as the positive (US+) and the negative (US-) experiences of the training paradigms.”

For other neurons, we state that the observed differences are mostly between aversive and appetitive training regimes, but refrain from stating that these neurons code the valence (page 10, lines 246 to 248):

“Activity changes of other sensory neurons were more subtle: for example, ASI, AWB, and ASJ neurons appeared to be related to the differences between aversive and appetitive experience (Figure 4-supplement 2,3).”

As for the AWC^ON^ neurons in the short term paradigms, we now state (pages 10 lines 243 to 246):

“Activity changes in the AWC^ON^ neurons were hallmarked by a large increase in all treated groups (when compared to the naive group), which is indicative of coding both the aversive and appetitive US (Figure 4H-J, Figure 2-supplement 3 E-H).”

As for the long-term paradigms, there are indeed marked procedural differences, and hence, we completely refrained from making any statements regarding the valence in these paradigms. In the discussion, we now acknowledge the limitations in assigning valence to neurons (page 20 lines 491 to 499):

“Assigning valence-coding neurons was somewhat limited due to the procedural differences in aversive and appetitive training paradigms. This was particularly evident when training for the long-term paradigms which involved several key procedural differences. Nevertheless, the procedural differences between positive and negative training in the short-term paradigms were minimal (see Methods), so valence-coding neurons could be extracted with higher certainty. These analyses indicated that the ASJ and ASI neurons may be the strongest candidates for coding valence (Figure 4A, Figure 7C). Indeed, the ASI neurons were shown to integrate information regarding food availability (Gallagher et al., 2013; Hapiak et al., 2013).”

Altogether, the resulting picture is still vastly confusing; the authors frequently conclude that certain neuronal activity changes "code" for various task parameters, but this is not really shown i.e., simply showing correlates of memory traces is not sufficient to make these statements. Moreover, It will be extremely challenging for the reader to navigate through the forest of > 30 Figures. I find it unsatisfactory that the authors main response to this is, in other words, – memory components are encoded in a distributed but in a somewhat unexplainable way-. The authors should make an effort to address this. I would suggest to revise Figure 4 (and Figure 3) to include all MOCK vs MOCK and MOCK vs naïve, also the sign of the δ values should be depicted, as well the significance levels of each matrix entry should be depicted on top of each square (not just labeling the neurons yellow). This would give a better overview. Then use this data matrix to train a classifier and test whether the task parameters, CS, US ( = the 2-4 different procedures), positive- and negative valence can be indeed decoded, and along which dimensions the major coding axes reside.

We have now closely followed all these suggestions. We revised Figures 3 and 4 to include all MOCK vs MOCK and MOCK vs naïve comparisons, and we provide the sign of the δ values (figure 3A-B and Figure 4A) together with the significance levels of each matrix entry (per comment number 10). This indeed provides a much better graphical view of the data.

We attempted to train a classifier on a fraction of the activity δ matrix and then to test against the held-out data, but this resulted in overfitting and poor accuracy due to the low number of repetitions. Instead, we performed a principal component analysis on the data (provided in the new figure 7), which clearly decoded between the different task parameters (CS, US etc.). In fact, the first two components sufficed to distinguish between these task parameters (Figure 7 A-B). Consequently, this PC analysis allowed us to assign coding roles to the different neurons (Figure 7C). We now explain this in detail in the Results section, pages 14-16 lines 354 to 392:

“Next, we performed a principal component analysis (PCA) on this difference Matrix. Interestingly, the first two components of the PCA already generated distinct clusters, each cluster representing an experience component (Figure 7A). For example, both CS components (bluish colors, denoted by the difference between trained and the corresponding mock-trained animals, Figure 1—figure supplement 3) are clearly distinct from the US components (green/red colors, denoted by the activity difference between naive and mock-trained animals). Furthermore, even within the experience components, each condition is distinctly clustered…

…Together, these analyses show that experience components are distributed across various neurons that collectively form a unique population code for each of the training paradigms.”

The applied PC analysis is detailed in the methods section (page 35, lines 789 to 802) and Figure 7—figure supplement 1.

“PCA-based filtering of activity deltas Neural activities were averaged across individuals for each trial. Activity deltas of averaged activities (see Figure 1—figure supplement 3) were calculated for the CS, US, and valence differences. All activity deltas for each comparison and trial were aligned in a single vector and subjected to PCA. Using the PC scores and loads from principal components 1, 2, 3, and 5 (out of 35 principal components), we reconstructed the original activity deltas while filtering out 39.1 % of the unrelated variance (Figure 7 C, see Figure 7—figure supplement 1A-C). The filtered activity deltas were then summed and normalized by the mean amplitude of the neuron (contributions of neurons with higher response amplitudes are weighted more). The mean of the six trials for each neuron was used to indicate increase or decrease in the activity, and the standard deviation to provide an estimate of the variance (Figure 7C). To validate the method, we simulated the data and subjected it to PCA-based filtering (Figure 7—figure supplement 1 E-M), allowing us to differentiate between significant and insignificant changes using k-means-based thresholding of the activity changes (Figure 7—figure supplement 1 N).”

Moreover, the possibility that the memory is distributed throughout the chemosensory neurons is further supported by the observation that classifiers trained on portions of the sensory raw activities could confidently classify the underlying memory conditions on the held-out portion of sensory activities (Figure 4K-M, Figure 4—figure supplement 7). In particular, the fact that combining the activities of different sensory neurons increased the prediction accuracy over single sensory neurons (an observation which is not replicated in the scrambled label controls) suggests that information is distributed across the sensory neurons.

We now explain this on page 11, lines 258 to 270:

“As multiple chemosensory neurons changed activity following training in each of the paradigms, we next asked how many neurons need to be considered to accurately describe each of the training-associated states such that their combined activity can distinguish between these training paradigms. We trained several classifiers (including k-means, random-forest, and a neural net) on a fraction of the data, after which we tested the model accuracy on the remaining data (see Methods). All classifiers provided similar results (Figure 4L-M and Figure 4-supplement 7E,F): When considering single neurons only, classification accuracy (measured as F1 scores) was rather low for all sensory neurons (up to 50%), though the scores were significantly higher than randomly expected (Figure 4K and Figure 4-supplement 7). When considering sets of sensory neurons, the decoding of the underlying training paradigm was better the more neurons were added to the model. Combining activities of all chemosensory neurons together resulted in 90% decoding efficiency, irrespective of the classification algorithm used (Figure 4L-M and Figure 4-supplement 7E-F).”

We also toned down the statements in the results part relating to figures 3 and 4, and we state that activity differences may suggest the encoding of certain memory components, rather than directly claiming that they encode the US or CS.

For example, on page 10 lines 229 to 231:

“However, in trained animals (positive and negative), the AWA neurons showed no

responses, suggesting that these changes code the stimulus component.”

On page 10 lines 241 to 243:

“This suggests that the ASER neuron may be coding the stimulus component of the memory as well as the positive (US+) and the negative (US-) experiences of the training paradigms.”

And on page 12 lines 291 to 293:

“This suggests that the AIY neurons, similarly to the AWA neurons, might encode the

stimulus component of the memory.”

We also changed the discussion of the distributed code to better explain its function. We now state that for short-term memories, the sensory neurons participate in population coding, where we can also assign certain coding roles. This sensory population code is then integrated in the interneurons. We address the population code on pages 19 and 20, line 480 to 509:

“Activities of sensory neurons were broadly modulated by short-term experiences (Figure 4A and Figure 7C). The widely-distributed changes in the chemosensory layer suggest that the sensory neurons encode the experience as a population code. Considering modulation of individual neurons only precludes accurate distinction between the training paradigms. But when considering all the chemosensory inputs, the underlying experience could be decoded with sufficient accuracy (Figure 4K-M), further highlighting the notion that the experience code is distributed among the sensory neurons. Within the distributed code of the experience components (Figure 7A,B), the PC-based analysis revealed how activity changes of individual neurons contributed to encoding the CS, the US, or both (Figure 7C and Figure 9C-D). […] The observed activity changes were generally consistent with the experience logic (positive or negative). For instance, in aversively trained animals, switching from DA to BUT leads to ASER activation (Figure 4 E-G), an activation that triggers a reversal behavior (Appleby, 2012; Suzuki et al., 2008) which prevents the animals from moving towards BUT. Changes in other neurons were harder to reconcile. For example, in both negatively and positively trained animals, activity in the AWA neurons was strongly reduced in response to BUT (Figure 4 B-D). Thus, it is presumably the combined modulated activity of all sensory neurons that is being integrated into the downstream interneuron layer to express the adaptive behavioral responses.”

We describe the integration of the sensory outputs on the RIA neurons and how it may underlie the differential choice of direction in trained animals, which is consistent with the lack of changes in command neurons’ activity and reversals. We also further highlight the limitations of our approach including the bias of this study towards certain neurons (pages 23-24, lines 575 to 596):

“Indeed, RIA neurons were shown to participate in formation and forgetting of aversive experiences, suggesting that past experiences may be converged onto RIA interneurons to shape behavioral outputs (Jin et al., 2016; Liu et al., 2022). By contrast, reversal frequency and speed were not modulated. Together with the lack of changes in AVA and AVE activity patterns, these results further support the notion that animal directionality towards the target underlies the enhanced attraction following positive training. Nevertheless, while the correlated changes in behavior and RIA activity hint to a functional link, additional experiments are needed to establish a causal role.

As memories are encoded in a distributed manner, it is presumably the integrative activity of many neurons that gives rise to the modulated RIA activity and the altered choice of directionality. As an integrated code, the impact of a single neurons' activity on locomotion might be limited. The need to consider the integrated response of all components was also evident from analyses of freely-behaving animals, where a population coding, rather the activity of any of the individual neurons, was shown to be a better descriptor of locomotion (Hallinen et al., 2021). Clearly, additional interneurons which control turning rates (e.g. AIB, AIZ, and RIM) and speed (e.g., RIB, SIA, and RMG) (Garrity et al., 2010; Iino and Yoshida, 2009; Lee et al., 2019; Li et al., 2014; Wakabayashi et al., 2004), may also play roles in these memory-induced behavioral modulations, and in fact, some of these neurons had been shown to participate in either memory formation (AIB and RIM) or memory retrieval (AIY and RIA) (Jin et al., 2016). In addition, while we exclusively focused on neurons whose activity was modulated upon memory retrieval, these and other neurons may also contribute to memory formation via transcriptional changes (Freytag et al., 2017; Lakhina et al., 2015).”

2) There are many instances where the authors applied repeated stimulus sequences leading to an inflation of datapoints, which are treated by the respective statistical tests as independent measures. This is incorrect and I fear many of the seemingly statistically significant results could fade-away when properly tested for. In all cases where the authors applied repeated stimuli mean responses per animal should be calculated so that each datapoint corresponds to one animal. In all figure captions n numbers must be indicated.

We have now changed the statistical comparisons so that all of the comparisons are based on individuals rather than on trials. Consequently, some of the comparisons turned out to be insignificant, yet, our main premises remained supported: Sensory neurons still show significant changes following short-term, but not following long-term, training paradigms. In addition, both short and long-term changes are still significantly observed in the interneurons.

Animal numbers are now denoted in every figure caption. Moreover, each animal is represented by a single line in heatmaps or a single dot in the scatter plots. Thus, variance, effect size, and population size, can be unambiguously estimated from the graphs. Consequently, the statistics, heat maps and scatter plots in Figures 3-5 and Figure 2—figure supplement 3-5. Figure 3—figure supplement 1-5, Figure 4—figure supplement 1-4, Figure 5—figure supplement 1-3 had been changed accordingly.

The only exceptions are Figure 4—figure supplement 2V,W; Figure 4—figure supplement 6; Figure 5—figure supplement 3O-T; and Figure 7 —figure supplement 1GandH. These figures address properties that are related to trial-to-trial variance or activity features visible in trials only.

3) I find the results of the regression analysis quite interesting but unfortunately the authors did not control for overfitting, a frequent pitfall in regression procedures particularly when multi-parameter models are used. The authors should devise a cross-validation procedure and report R2 values for held-out test data.

We have now performed the suggested overfitting tests on all of the regression models. These analyses indicated that our conclusions regarding changes in the connections between the sensory neurons and AIY remain valid with slight differences (see details below and Figure 6 and Figure 6—figure supplement 1). In contrast, the resulting R^2^ values for AIA and RIA were too low, possibly due to the large number of regressors (as these neurons have many more inputs) and the limited number of trials. We therefore removed the regression analysis for AIA and RIA neurons from the manuscript.

More specifically, the R^2^ scores for AIY (see Figure 6—figure supplement 1 F) were low only for the STAP-Trained and STAV-Trained training paradigms. For all other tested paradigms, R^2^ values increased as more regressors were included. This prompted us to further test whether our main claim, the change in the regression coefficients, is still supported. Therefore, we plotted individual regression coefficients following re-testing in cross-validation to check if the variation in regression coefficients between conditions persists even if restricted data is used (Figure 6—figure supplement 1G). In this case, the numerical values of the regression coefficients of individual repeats were consistently clustered and their means were not far from the regression coefficient calculated using the full data. More importantly, variation of regression coefficients between memory conditions is still clearly evident when these restricted re-trials of the cross-validation are considered (Figure 6—figure supplement 1G). This supports our notion that these regression coefficients genuinely vary between the different memory paradigms. In the revised version, we raise the limitations of our model and state that the model mainly serves to identify the coefficients, reflecting synaptic communication routes, that are clearly modulated following training. Moreover, we caution that these regression coefficients are no direct metrics for synaptic weights.

We state the new results for AIY neurons, emphasizing that only the identity of the modulated connections can be inferred (page 13-14, lines 320 to 340):

“To study how the modulated activity of the sensory neurons impacts the activity of the interneurons, we considered a simple mathematical model where interneuron dynamics is dictated by a linear combination of the sensory neuron activities. For this, we averaged activities of each trial for each of the sensory neurons across all animals in the different paradigms (Figure 6—figure supplement 1I), and used a multivariate regression analysis to extract the weights that would best fit the activity of the AIY neurons (see Methods). As expected, the more sensory neurons added to the model, the better was the overall prediction of the AIY activity (Figure 6A,D). When considering the combined activity of five sensory neurons types (namely, AWA, AWC, ASE, AWB, and ASG), up to 88% of the variance in the AIY neurons activity could be explained (Figure 6A-D). However, this improved accuracy was detected for the naive and the aversively-trained animals (Figure 6A, pink arrows), while no improvement (compared to when considering AWA alone) was detected for the positively trained animals and the associated mock controls. This lack of improvement suggests that ASER, AWC, ASG, and AWB neurons contribute more to encoding the aversive experiences, while the AWA neurons were the prime contributors to the downstream AIY activity in appetitive experiences (Figure 6C,D). Model evaluation using F-statistics and cross validations indicated that different combinations of sensory neurons should be used to best explain the AIY activity in each of the training paradigms (Figure 6figure supplement 1H). Together, these analyses indicate specific and distinct synaptic routes between the chemosensory neurons and the postsynaptic AIY interneuron are modulated in a paradigm-specific manner (Figure 6E).”

And in the discussion, we highlight the limitations of the our model (page 21-22, lines 542 to 565):

“Population coding requires integration of the coding sensory neurons into downstream neurons, such as the AIY interneuron. A regression analysis revealed that AIY activity can be explained via a simple linear combination of the sensory neurons (Figure 6). This analysis provided a quantitative measure for the relative contribution of the sensory neurons to the overall activity of the AIY neurons. These relative contributions uniquely changed in an experience-dependent manner, suggesting that the identity and the specific modulation of neural communication routes dictate each of the specific memory types. For example, the AWA neurons contributed the most to the modulated activity of AIY neurons in the appetitive paradigms, while AWC, ASER, ASG, and AWB neurons dominated in dictating AIY activity in naive animals and following aversive training (Figure 6E). In that respect, the AWA neurons are associated with attractive stimuli (Bargmann et al., 1993), while the sensory neurons required for aversive conditioning are mostly associated with reversals and avoidance (Gray et al., 2005; Suzuki et al., 2008; Troemel et al., 1997).

Notably, the aim of the linear combination model was to identify modulated synaptic communication routes. The computed regression coefficients, which denote a change in the communication routes, are not a direct measure of the underlying synaptic weights. This is due to the limitations in data acquisition (i.e. frame rate, kinetics of GCaMP, and number of available data points, etc.) and the fact that we measure calcium levels which are only correlates of neural potential. Furthermore, the modulated communication routes may include, in addition to the classical chemical synapses, also electrical gap junctions that were demonstrated to be imperative to form memories in *C. elegans* (Choi et al., 2020).

Nonetheless, the model reliably revealed specific communication routes (chemical and electrical synapses combined) that changed in an experience-dependent manner.”

In the Supplement, we provide the detailed results concerning the overfitting tests and the verification of the regression coefficient scattering (page 71, lines 2055 to 2071):

“(F) Cross-validation of the multivariate regression model shown in Figure 6. Data for training and testing were randomly split 50/50. Regression coefficients were calculated on half of the data. Then R^2^ was calculated using these coefficients on the set-aside portion of the data. This procedure was repeated 10 times. Each field denotes the mean of R^2^. Note that in NAIVE and MOCK conditions the R^2^ values increase the more neurons are added to the model. In STAP, R^2^ decreases due to insignificant regressors (ASE, AWC, ASG, and AWB are not significantly contributing to AIY in STAP, see pane I and Figure 6C, and the AWA neurons account for most of the variance).

(G) Cross validation yields low R^2^ values for STAP and STAV. However, the regression coefficients show robust variation throughout conditions during re-testing. Scatter plots show regression coefficients of the AWA, ASER, ASG, and AWC^ON^ neurons as calculated based on the cross-validation shown in panel F. Single dots are regression coefficients of individual retrials. Colored bar represents the cross-validation mean and the black bar denotes the regression coefficient when using the entire data. Despite the low R^2^ values in cross-validation (F), the regression coefficients are consistently different across the various training paradigms, suggesting that the changes in the regression coefficients are genuine and generalizable.”

In the Methods section, we explained why regression analysis could not be extended to AIA and RIA neurons (page 34, line 785 to 787):

“Multivariate regression analysis could not be extended to AIA and RIA neurons because

there were too many input neurons (regressors) opposed to too few data points.”

4) AWCon and AWCoff in the multi-neuron recordings were classified based on high/low responses in MOCK, a classifier that was empirically inferred from imaging experiments performed in a marker line. This classifier seems unambiguous in STAP MOCK only, but how were AWCon and AWCoff classified in the multi-neuron experiments in all other conditions"?

There were two types of experiments, one consisting of butanone only, and the other involving butanone/diacetyl exchange.

In the latter case of butanone/diacetyl exchange, there was no ambiguity in assigning the individual AWC neurons. In all animals of the pan-sensory neuron strain, the two AWC neurons responded differently to each of the stimuli: One of the AWC neurons responded following diacetyl removal (butanone presentation) while the other neuron responded during butanone removal (diacetyl presentation). The activation dynamics in the latter case followed precisely the dynamics observed in the AWC^ON^-specific marker line, exhibiting a slow rise in activity across all conditions (compare panels F and J in Figure 2—figure supplement 3, and also evident in the heatmap shown below). Consequently, the neuron responding during butanone removal was assigned as AWC^ON^. The other AWC neuron showed a fast activation following diacetyl removal (butanone presentation). While this was not fully reproduced by the AWC^OFF^-specific marker line (Figure 2—figure supplement 3 L and H), by elimination, we designated this AWC neuron as AWC^OFF^. This issue also constitutes the activity discrepancy which we address in comment number 5 below.

When classifying the AWC neurons in the butanone only experiments, both neurons respond to butanone removal (Figure 3). While there might be ambiguity in our identification and neural assignment, we note that in our marker strains with known AWC identities, the AWC^OFF^ neuron consistently showed a stronger transient activation with a fast rise time and high amplitude followed by a slow decline towards baseline (see Figure 2—figure supplement 3 C upper line). In contrast, AWC^ON^ activation showed lower amplitudes and a slower rise time. We therefore designated the neuron with the lower response amplitude (during the 15 seconds post stimulus exchange) as AWC^ON^ and the neuron with the higher response amplitude as AWC^OFF^(compare Figure 2—figure supplement 3 A vs C).

Thus, in the majority of the response traces, the two neurons show marked differences in their activation patterns. We also agree that in a few instances, we may have wrongly assigned traces to neurons as ambiguity arises when both neurons respond with similar magnitudes (which really concerns only a few cases). However, when averaging over all traces to get the population mean, such possible erroneous assignments cancel out and the mean activity provides a good depiction for each neuron.

To corroborate this, we trained a k-means classifier on the response activities from the known reporter line (traces shown in Figure 2—figure supplement 3C). We then used this classifier to assign traces to neurons in the pan-sensory reporter line. We found almost identical assignments across all conditions, except for STAVT animals in which the AWC^ON^ neuron showed slightly higher activation. The results are provided in Author response image 2.

Nevertheless, we have now revised the manuscript and refer to these neurons as AWC^W^, for the weakly activated neuron, and AWC^S^ for the strongly activated neuron (Figure 3 and Figure 2—figure supplement 3, Figure 3—figure supplement 2 and 3). However, when referring to the population means we refer to these neurons as AWC^ON^ and AWC^OFF^, respectively.

We now explain how we assigned the neurons in the legend of Figure 2—figure supplement 3, page 55-56, lines 1595 to 1620:

“AWC neural activities in the pan-sensory strain were classified into strongly responding (AWC^S^) and weakly responding (AWC^W^) AWC neurons as in each animal the two AWC neurons showed marked differences in response amplitude. This classification is consistent with the activation pattern observed in the reporter strain with known AWC identities:

(A,B) When using the pan-sensory reporter strain (Posm-6::GCaMP), the two AWC neurons exhibited distinct activities in response to butanone removal. One neuron showed high magnitude responses in trained and mock-trained animals for both short-term positive (STAP) and short-term aversive (STAV) training paradigms (denoted as AWC^S^). The second neuron exhibited weak responses in mock-trained animals (designated as AWC^W^), but robust strong responses following STAV and STAP training. Note the differences in mean activity between the conditions in the AWC^W^ neuron (line graphs, n=9-15).

(C,D) Imaging activity of both AWC neurons where the identity of AWC^ON^ and AWC^OFF^ is known. Here, the AWC^OFF^ neuron showed strong responses to butanone removal in both trained and mock-trained animals. These dynamics correspond to the AWC high neuron shown in panel A. Accordingly, AWC^ON^ matches the dynamics corresponding to AWC low

(compare line graphs below, n=6-16).

Note that in naive and mock-trained conditions, one of the AWC neurons is inactive (or very weakly active). Hence, we termed this neuron as AWC^w^ (weak) while the other active neuron we termed AWC^S^ (strong). In the trained conditions, both neurons are activated, though there is a clear distinction between their amplitudes.

Based on the activation patterns obtained from the known AWC-identity reporter strains, activation of the AWC^OFF^ neurons is usually faster and higher than that of AWC^ON^. Hence, at the neural population dynamics of the weakly activated AWC (AWC^W^) and the stronger

activated AWC (AWC^S^), it is plausible to identify them as AWC^ON^ and AWC^OFF^, respectively.”

In the result section of the main text, where we first address AWC neurons, we reference these observations by directing the reader towards Figure 2—figure supplement 3 and introducing the AWC^W^ neuron, page 8 line 181 to 187:

“For example, while the AWC^W^ neuron (identified as AWC^ON^, see Figure 2—figure supplement 3) shows mild innate responses to BUT in naive and mock-trained animals, its response activity was significantly heightened following short-term appetitive (STAP) and short-term aversive (STAV) training paradigms (Figure 3C-F). Since the sole difference between the trained and the corresponding mock-trained animals was the presence of the CS (BUT) during the training period, the differential activity suggests that the AWC^W^ neuron may be coding the stimulus component of the memory.”

**Author response image 2. sa2fig2:** Classifying AWC neurons based on simple activity thresholds and on k-means classification yields almost identical population mean activities. In image rows labeled as ‘k-means’, AWC neurons were classified by k-means trained on activation features (such as, amplitudes,steepest ascend/descend, minimum, maximum etc) of AWC activities of known identities (data shown in Figure 2—figure supplement 3). In image rows labeled as ‘Threshold’, AWC neurons are classified by the overall activity within a 15 second window. (**A**) shows comparisons of neurons classified as the weakly activated AWC neuron (AWC^S^ which correspond to AWC^ON^). (**B**) shows comparisons of neurons classified as the weakly activated AWC neuron (AWC^S^ which correspond to AWC^ON^). When comparing ‘Threshold’ and ‘k-means’ graphs, note that the population means are extremely similar (identical in most cases). The only evident differences occurred in aversively-trained animals in short-term memory (light red curves). In some of the groups individual neurons have been classified differently by the two methods. This demonstrates that differentially classifying ambiguous cases has very little impact on the overall mean. In both classification methods, the conclusion that AWC^ON^ neurons increase activity in Short-term but not long-term is supported. Given that k-means performed with only ~88 % accuracy on the training data, we feel more confident with the activity threshold method as it is also more in line with the experimenter's experience after visually reviewing the data.

5) AWC and AWA (Figure S21, S7E-L) show different results in different imaging lines. I think the authors do not provide a convincing explanation for this.

We believe the variable response is due to the transgenic lines where over-expression of reporter proteins, and presumably more so of a calcium indicator, may affect and divert responses of specific neurons under specific conditions. Specifically, GCaMP expression is associated with modulated synaptic and circuit properties (Singh et al. 2018; Steinmetz et al. 2017). Consequently, in some cases, reporter strains with different expression levels may show variable activities. In particular, such cases may appear when assaying multiple neurons across many different conditions. We note that the vast majority of the data shows high consistency. For example, at high butanone concentrations, responses of the AWC neurons in the multisensory line are consistent with the neuron-specific marker line (Figure 2—figure supplement 3A-D). Similarly, at higher butanone concentrations, there is no difference in the responses of the AWA neurons between the multi-sensory reporter line and the neuron-specific line.

Furthermore, we note that for the AWA neurons, both reporter lines encode the memory yielding the same valence logic: in the pan-sensory line, aversively-trained animals showed an increased response upon switching from butanone to DA. In the AWA reporter line, appetitively-trained animals showed a heightened response upon switching from diacetyl to butanone (Figure 4—figure supplement 4 M-N). Thus, in these opposite-valence trained animals, the outputs maintain the same functional logic.

Also, to verify that the multi-sensory reporter line is not defective, we analyzed its training-induced behavioral outputs and found that they were consistent with the behavioral outputs of WT animals (Figure 4—figure supplement 5).

In the revised version, we raise the variable activity and suggest that this difference might be due to variable GCaMP expression loads.

In the caption of Figure 2—figure supplement 3 and Figure 4—figure supplement 4, pages 54 and 64, respectively:

Page 56, lines 1630-1635:

“Note that AWC^OFF^ presumably got sensitized to diacetyl removal while in the naive state still maintaining vestiges of butanone sensitivity (see H). As the two reporter strains have vastly different degrees of GCaMP expression and GCaMP expression has been shown to alter synaptic transmission and firing properties of neural networks (Singh et al., 2018; Steinmetz et al., 2017), it is likely that these variable activities are due to differences in GCaMP expression.”

Page 66, lines 1877 to 1879:

“We believe these variable activity patterns to be the result of variable GCaMP3-expression load in our reporter lines.“

[Editors’ note: what follows is the authors’ response to the third round of review.]

The manuscript has been improved and all reviewers are very enthusiastic about having the revised paper published in eLife, but there are some remaining issues that need to be addressed, as outlined below:Please consider suggestions by Reviewer #1 on AIY activity and similar changes in AP and AV paradigms. Also, please consider suggestions (1) – (4) by Reviewer #3. We look forward to the revised version rather soon.Reviewer #1 (Recommendations for the authors):The revised manuscript has further improved by addressing concerns raised in previous reviews.More specifically, the authors have further analyzed their data by focusing on paradigm-specific activity of sensory neurons in response to BUT – DA stimulation patterns. This focus allowed them to see more robust effects generated by different types of experience to better support the data interpretation. Their interpretation also improved and are now more supported by the results.Their focus on using the linear combination of presynaptic sensory activities to explain AIY activity is appropriate given the sample size and variabilities of their imaging results across animals and across trials. It is informative to see that AIY activity can be explained by those of anatomically connected sensory neurons. It is worth noting that AIY activity is best explained by the activities of sensory neurons under mock conditions. This might suggest the role of these neurons in responding to the US experience, not necessarily the association of CS with US.

Agreed, and in the results part, where we address AIY regression, we have now included this observation (Page 14, lines 337-342):

“Overall, the highest portion of variance that explains AIY activity was obtained in the mock-trained animals, suggesting that the US experience alone may dominate the sensory-to-AIY input weights. This is particularly evident for the appetitive conditioning where the presence of food alone yielded a better fit to AIY activity than the training regime consisting of food and BUT combined.”

The principal component analysis in Figure 7 and supplementary figures is an improvement on data analysis and useful in dissecting complex neural responses in a large number of different imaging conditions implemented in this study.Another clear feature of the neurons characterized here is that they seem to show similar patterns of activity changes in AP and AV paradigms (for example, as shown in Figure 5 for sensory neurons AWA, interneuron AIY and AIA; for AWC in Figure 3; for RIA in response to DA-BUT stimulation pattern). Although AP and AV paradigms showed different effects on RIA in response to control-BUT stimulation, it is not clear how the changes relate to chemotaxis behavior. The authors speculate this feature as one indicator for combinatorial effects. This is possible but remains to be tested in future studies.

We agree and we now restate that RIA is one of several possible neurons that mediate the sensory changes to the observed modulated behavior (pages 23-24, lines 593-607).

“As the animals’ reversal frequency and speed were not modulated (and no changes in AVA and AVE activity patterns were observed either), RIA is a possible candidate that could underlie the observed change in directionality. However, it is impossible to tell whether animals changed directionality due to RIA-induced head swing bias, or alternatively, due to reorientation mediated by command neurons (e.g., AVA, AVE) since we tracked animals as center mass points. Also, activity changes in RIA neurons were significant though small, thus raising the question whether these changes are large enough to mediate the observed trained phenotype. Moreover, additional interneurons that control turning rates (e.g. AIB, AIZ, and RIM) and speed (e.g., RIB, SIA, and RMG) may also play roles in these memory-induced behavioral modulations (Garrity et al., 2010; Iino and Yoshida, 2009; Lee et al., 2019; Li et al., 2014; Wakabayashi et al., 2004). In fact, some of these neurons had been shown to participate in either memory formation (AIB and RIM) or memory retrieval (AIY and RIA) (Jin et al., 2016). Thus, additional experiments are required to support the functional role of RIA plasticity in shaping the learning-dependent behavioral outputs.”

When analyzing locomotion in chemotaxis, the revised study removed the results on long-term training and focused on short-term training. The results are easier to be understood than before. The short-term training paradigms modulated the directionality toward the CS, but not the speed and reversal frequency.As a whole, the study produced a large collection of activity recordings in a handful of sensory neurons and a few interneurons with the goal to explain chemotaxis behaviors under 4 learning paradigms. Although more work is needed to explain how these activity patterns relate to behavior, the collection of data provides hypotheses for future studies on function of the neurons implicated in the learning paradigms based on calcium imaging results and provide useful references for similar studies in the field.Reviewer #3 (Recommendations for the authors):Upon revision, authors made a good job to improve the quality and readability of the manuscript. Additional evaluation of data was performed such as cross validation of linear regression and classification models, which adequately evaluates authors' assertions and strengthen them.The concern by reviewer #3 was appropriately addressed, with the exception of the following.This reviewer recommends publication of the paper because it is a comprehensive analysis which adds precious information to the research community, provided that the presentation flaws as listed below are corrected.1) As previously pointed out, definition of "trials" needs to be explicitly provided. It is understandable that reflecting the revision history, there are several different protocols for the imaging experiments. For this reason, authors need to carefully try to avoid confusion for the readers. In the current version, the figure depicting "trials" appear in Figure 6—figure supplement 1I, while the word first appears in the text in line 255 where authors say "most animals (or trials) showed homogeneous responses", and here, readers may be puzzled. In Methods, there is also no such word, though it is described that repeated stimuli were applied in experiments in Figure 2B and Figures4-5.

Indeed, and to avoid possible confusions, we now explicitly explain what we mean by ‘trial’ early on in the in Results section (Page 11, lines 253-255):

“(in the following, the term ‘trial’ refers to individual BUT/DA exchanges, where animals underwent six cycles of such trials (exchanges), Figure 4-supplement 6A).”

We also added the definition of ‘trials’ to the Methods section (Page 29, lines 704-705):

“Each BUT presentation/removal is referred to as a ‘trial’.”

And on line 710:

“Note that in Figures 4-7, the BUT/DA exchanges are referred to as trials.”

2) p23, page 581: related to the previous comment by reviewer #1 (third in Major point), there is not enough basis provided in this paper that leads to the conclusion that head swing control (klinotaxis which RIA has been implicated) is the major mechanism that causes learning-dependent modulation of chemotaxis. The other major mechanism known, klinokinesis, involves changes in the turning frequency DEPENDING ON chemical concentration changes, where animals backup and turn more frequently when facing away from the odor source and odor concentration decreases upon its locomotion. It would also cause the deviation angle smaller.

Indeed, and as also explained above in response to reviewer #1, we have now restated and toned down our conclusion regarding RIA (Pages 23-24, lines 596-607):

“However, it is impossible to tell whether animals changed directionality due to RIA-induced head swing bias, or alternatively, due to reorientation mediated by command neurons (e.g., AVA, AVE) since we tracked animals as center mass points. Also, activity changes in RIA neurons were significant though small, thus raising the question whether these changes are large enough to mediate the observed trained phenotype. Moreover, additional interneurons that control turning rates (e.g. AIB, AIZ, and RIM) and speed (e.g., RIB, SIA, and RMG) may also play roles in these memory-induced behavioral modulations (Garrity et al., 2010; Iino and Yoshida, 2009; Lee et al., 2019; Li et al., 2014; Wakabayashi et al., 2004). In fact, some of these neurons had been shown to participate in either memory formation (AIB and RIM) or memory retrieval (AIY and RIA) (Jin et al., 2016). Thus, additional experiments are required to support the functional role of RIA plasticity in shaping the learning-dependent behavioral outputs.”

3) p19, line466 and later: It is convincing that AIY changes its activity after LTAV training in DA-BUT imaging experiments. On the other hand, changes in RIA are not convincing. Figure 5G,H, RIA response is suggested to be increased, but it looks like the conclusion is based on two outlier animals. Also in Figure 3—figure supplement 4K/L, RIA activity looks increased even without BUT stimulus. Therefore, long-term changes in RIA activity needs to be toned down, at least in the discussion (where RIA is so heavily discussed in the current version). As authors depicted in Figure 9B, it is clear through the extensive data by the authors on sensory neurons that there are virtually no changes in the sensory neuron responses after long-term memory, but the behaviors change. As a logical consequence, interneuron activities need to be changing. However, observed changes are limited to AIY LTAV so far, and therefore hitherto unobserved changes may underlie the behavioral changes, for example that of AIB or any other neurons in the circuit.

We agree, and we have now toned down our conclusions for RIA. We refrain from presenting RIA as the functional link between modulated sensory activities and the modulated behavioural output, and instead, we state that RIA is one of possible interneurons (not included in this study) that could exert this function (see point above).

Also in the discussion, we note that we refer to the short term memory only (and not the long term) and how it is reflected in the AIY and RIA activities (page 23, lines 585-588):

“Behavioral assays following short-term training revealed that positively-trained animals were more directed towards the CS, while negatively-trained animals were significantly less directed (Figure 8C-D). Interestingly, aversive training also inflicted significant changes in sensory-specific responses of the RIA neurons (Figure 5G,H). As the RIA neurons …”

And following this paragraph, we state that it is probably the integrative activity that underlies the directionality bias (lines 608-609):

“As memories are encoded in a distributed manner, it is presumably the integrative activity of many neurons that gives rise to the altered choice of directionality.”

4) p18 line 450: It is misleading to suggest sensory neurons are not involved in long-term memory. Authors only looked at calcium level in sensory neurons. However, there are multiple cases known in *C. elegans* and other animals where synaptic output machinery is modulated by learning, which is not reflected in calcium imaging of the sensory neurons; rather, it will be reflected in the response of downstream interneurons.

We agree and we have now stated accordingly that sensory neurons can still code long-term memory via synapses (Page 18 lines 462-464):

**“**Moreover, sensory neurons may still be coding long-term memories via modulation of the synaptic output, which is not reflected by soma calcium activity (Oda et al., 2011).**”**

To avoid confusion in that section between these separate neural functions we also replaced the term “sensory neurons” by the more explicit “sensory neuron’s activity”.

5) It is a bit of a concern that in cross validation in Figure 4—figure supplement 7A, authors used all individual animals for training the model, especially given that any single neuron can be a predictor, which suggests overfitting. However, revisions are not requested on this as per the principle of the journal.

We agree and we have now repeated the analysis when splitting by animals rather than by trials. We have now added this analysis to the supplementary figure (Figure 4 and Figure4-supplement 7F):

“(F) When trials per animal are averaged and animals are partitioned in training and test data using 80/20 split, the classification error increases. All classifiers still perform better than scrambled label controls. In particular, the neural net performs three times better than expectation and clearly shows an increase in accuracy as neurons are added to the model. The neural net was composed of three layers with 50, 40, and 70 neurons, respectively. K-means nearest-neighbor used the distances to the next nine data points. The random forest consisted of 350 decision trees.”

We also added to the methods section a paragraph that describes this newly added analysis (page 34lines 786-790):

“When cross-validating the results after splitting the data into training and test data based on animal repeats, we obtained similar results, though with higher classification error.

Thus, classification accuracy could benefit from adding more animals per training group (see ‘between individuals’, Figure 4-supplement 7F).”